# Computational single-neuron mechanisms of visual object coding in the human temporal lobe

Runnan Cao [1] ✉, Jie Zhang [1], Jie Zheng [2], Yue Wang[1], Peter Brunner [3], Jon T. Willie [3,4] & Shuo Wang [1,3] ✉

Understanding how the human brain encodes visual objects involves deciphering the neural computations and circuits in the temporal lobe. Here, we recorded intracranial EEG from the human ventral temporal cortex (VTC) and medial temporal lobe (MTL), as well as single-neuron activity in the MTL, to investigate the computational mechanisms of neural object coding. The VTC exhibited axis-based feature coding, and a neural feature space could be constructed using VTC neural axes, within which visual objects clustered according to high-level categorical relationships. Importantly, MTL neurons encoded receptive fields within this VTC neural feature space, exhibiting selective responses to objects that shared perceptual and conceptual similarities. This computational framework, therefore, explains how dense, feature-based representations in the VTC are transformed into sparse, high-level representations in the MTL. We further validated our findings using an additional dataset with different stimuli. Notably, we uncovered the physiological basis of this computational framework by demonstrating VTC-MTL interactions at multiple levels. Together, our neural computational framework provides a mechanistic understanding of the neural processes underlying object recognition.

Humans effortlessly recognize everyday objects within a fraction of a second, despite tremendous visual variations in appearance[1,2]. The ability to rapidly extract object identity is fundamental to various high-level cognitive functions essential for survival[3]. Understanding how the human brain achieves object recognition has garnered significant attention over the past three decades due to its importance[2,4–7]. However, we still lack a mechanistic understanding of this process, which underlies a complex computational feat[8–10].

Object recognition is primarily supported by the ventral visual pathway, which spans the ventral temporal cortex (VTC) and extends into the medial temporal lobe (MTL), including the amygdala and hippocampus[11,12]. Within this pathway, neurons encode objects at different levels of abstraction, transitioning from lower-level, feature-

based representations in the VTC to higher-level, conceptual representations in the MTL. Specifically, the VTC represents objects using axis-based coding, where neural responses are organized along key feature dimensions such as shape, texture, and curvature[13,14]. Each neural axis represents a fundamental direction in this high-dimensional feature space, capturing variations in object appearance. Non-human primate studies[15–18] have shown that axis-based coding enables efficient and flexible object representation, allowing different objects to be distinguished based on their position along multiple neural axes. Axis-based coding in the VTC results in dense coding, meaning many neurons participate in encoding the fine-grained details of an object's visual features. This requires a broad and distributed population of neurons[19–22], with each neuron responding to

[1]Department of Radiology, Washington University in St. Louis, St. Louis, MO, USA. [2]Department of Biomedical Engineering, University of California Davis, Davis, CA, USA. [3]Department of Neurosurgery, Washington University in St. Louis, St. Louis, MO, USA. [4]Department of Neurosurgery, University of Texas at Austin, Austin, TX, USA. ✉e-mail: r.cao@wustl.edu; shuowang@wustl.edu

multiple objects that share specific visual features. In contrast, the MTL employs sparse coding, where only a small subset of neurons respond selectively to specific object categories[23–26], abstracting away lower-level visual details[25,26]. The transformation from dense, feature-based coding in the VTC to sparse, category-based coding in the MTL is critical for efficient memory storage and retrieval. However, the precise neural computational mechanisms underlying this transformation remain unclear.

To address this question, our recent work suggests that MTL neurons encode "receptive fields" (i.e., coding regions) within a visual feature space, which is constructed using deep neural network (DNN) features (rather than neural responses), for faces[27] and objects[28]—a phenomenon we refer to as region-based coding. Rather than tracking individual visual features, these neurons selectively respond to specific regions (i.e., receptive fields) within the visual feature space that contain objects with similar visual features. Critically, region-based coding can provide a potential link between visual feature processing in the VTC and conceptual representations in the MTL (Fig. 1). A neural feature space can be constructed from VTC neural axes using principal component analysis (PCA). Notably, the dimensions of this space reflect principal variations in object categories (e.g., ranging from natural to artificial objects or from animate to inanimate objects; see Methods)[18,29]; and within this space, objects with similar high-level abstract features cluster together, demonstrating clustering that reflects their perceptual and conceptual relationships. If MTL neurons exhibit receptive fields (i.e., coding regions) within this neural feature space and respond to stimuli that fall into these regions and share similar high-level properties, they will exhibit category-selective responses. Therefore, this neural computational framework can explain how the brain transforms feature-based representations in the VTC into more abstract and integrative representations in the MTL.

To test this hypothesis, we simultaneously recorded microscopic single-neuron activity in the human MTL and mesoscopic intracranial electroencephalography (iEEG) in both the VTC and MTL while neurosurgical patients viewed 500 naturalistic object images spanning diverse categories. Specifically, we hypothesized that (1) the human VTC exhibits axis coding of visual features similar to that observed in non-human primates[18,30] (i.e., iEEG responses parametrically vary as a function of DNN features; Fig. 1 bottom left), (2) VTC neural axes collectively form a neural feature space after dimensionality reduction using PCA (Fig. 1 bottom middle), and (3) the human MTL exhibits region coding within the VTC neural feature space (i.e., MTL neurons selectively respond to objects that occupy similar positions within this space; Fig. 1 bottom right). To further support these hypotheses, we investigated inter-areal functional connectivity between the VTC and MTL at multiple levels, including iEEG phase-locking, Granger causality, and spike–field phase consistency. Additionally, we validated and generalized our findings using an additional dataset with different stimuli. Together, this study provides critical insights into the neural computational mechanisms of how the brain transforms detailed visual feature information into abstract, category-specific representations of visual objects.

## Results

### Neural response to visual objects in the VTC and MTL

Fourteen neurosurgical patients (9 females; Supplementary Table 1) implanted with depth electrodes participated in this study. Participants viewed 500 naturalistic ImageNet object images from 50 different categories[31] while performing a one-back task (Fig. 2a). They performed well in the task, with an average accuracy of $86.38\% \pm 11.57\%$ (mean ± SD), suggesting that they were paying attention to the task. This task has been shown to effectively maintain participants' attention while efficiently presenting a large number of images in a single session[27,32]. We recorded iEEG from 1074 channels: 692 in the VTC ($n = 278$ for the fusiform gyrus [FG] and $n = 414$ for the inferior

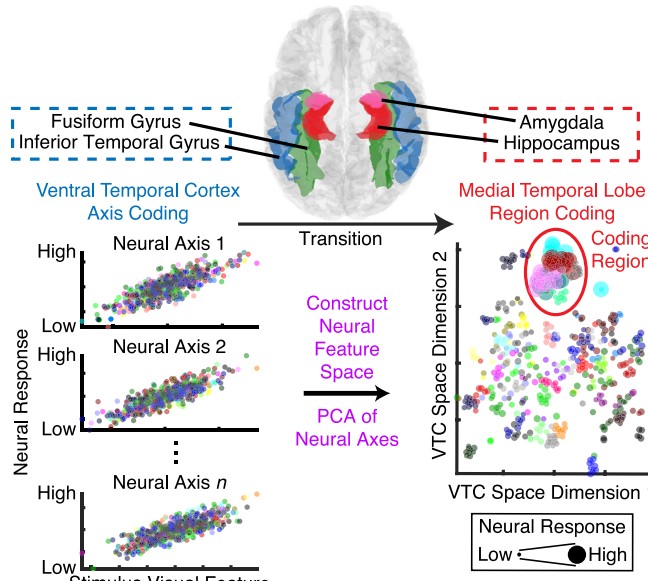

**Fig. 1 | A neural computational framework illustrating the transition from axis-based coding in the VTC to region-based coding in the MTL.** Neurons and iEEG channels in the VTC (note that here neural response can refer to iEEG HGP, as in the present study, or to single-neuron activity, as in classical studies of axis coding in the VTC[15,18]) encode visual features (e.g., DNN features) using an axis-based code, where neural coding axes are organized along visual feature dimensions (i.e., parametrically varying as a function of visual features). Axis-based coding in the VTC results in dense coding, requiring a broad and distributed population of neurons. A neural feature space can be constructed using PCA of all neural axes from a brain area (e.g., VTC), where the dimensions of this space represent principal variations in object categories (e.g., ranging from natural to artificial objects or from animate to inanimate objects). Notably, within this space, objects with similar high-level abstract features cluster together, demonstrating clustering that reflects their perceptual and conceptual relationships. MTL neurons receive processed visual input from the VTC and encode receptive fields (i.e., coding regions) within this neural feature space, exhibiting region-based coding. Traditionally, a receptive field refers to the portion of the visual field that a neuron responds to. Here, we extend this concept by defining a receptive field within the neural feature space—representing the range of visual feature values that elicit a response from an MTL neuron. Instead of responding to a specific location in the physical visual field, these neurons respond to objects that occupy similar positions in the neural feature space. By responding to stimuli that fall within these regions, MTL neurons exhibit selective responses to objects that share similar perceptual and conceptual features, thereby providing a crucial link between feature-based visual processing in the VTC and high-level representations in the MTL. Together, this computational framework explains how the brain transforms detailed visual feature representations into abstract, concept-driven encoding. Each color represents a different object category.

temporal gyrus [ITG]; Fig. 2b left panel) and 382 in the MTL ($n = 100$ for the posterior hippocampus [PH], $n = 105$ for the anterior hippocampus [AH], and $n = 177$ for the amygdala; Fig. 2b right panel). We used high-gamma power (HGP; 70–170 Hz) from the iEEG signals as the response, which reflects the average neuronal firing of local neural populations[33–35].

We first examined the visual responsiveness of the iEEG channels across different brain areas (Fig. 2c, e). A significant percentage of visually responsive channels was observed in each subregion of both the VTC (FG: $n = 201$, 72.30%, binomial test against 5% chance level: $P < 10^{-20}$; ITG: $n = 141$, 34.06%, $P < 10^{-20}$) and MTL (PH: $n = 34$, 28.57%, $P < 10^{-20}$; AH: $n = 47$, 32.87%, $P < 10^{-20}$; amygdala: $n = 78$, 42.86%, $P < 10^{-20}$), with the FG containing a significantly higher proportion of visually responsive channels compared to all other regions (Fig. 2c; $\chi^2$ test: $Ps < 0.0001$ for all comparisons between the FG and other regions). The results remained consistent using a linear mixed-effects

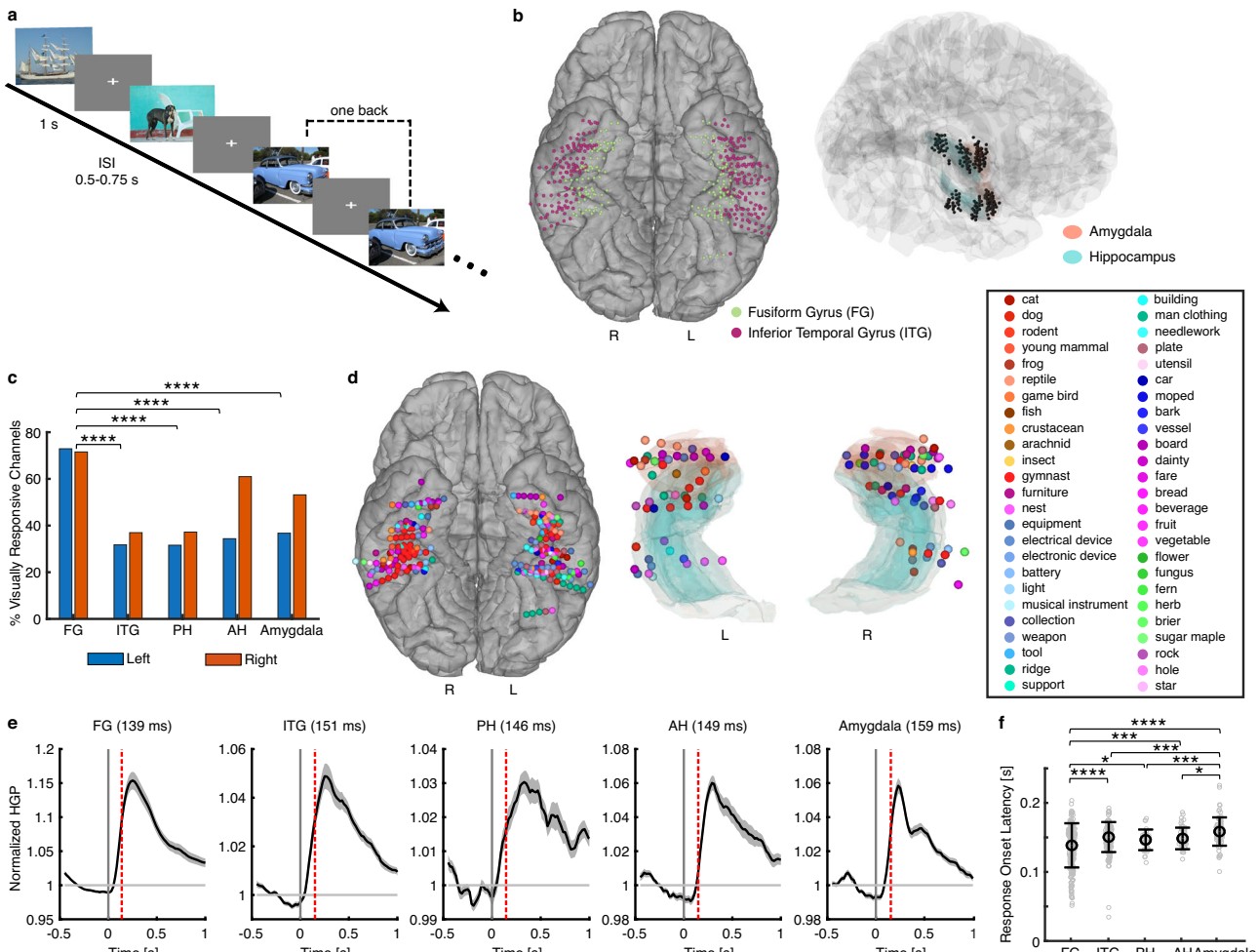

**Fig. 2 | Neural response to visual objects in the VTC and MTL. a** Task. We used a one-back task in which participants were asked to respond when an identical image was repeated consecutively. Each image was displayed for 1 second, followed by a jittered inter-stimulus interval (ISI) of 0.5 to 0.75 seconds. Images were obtained from Deng J, Dong W, Socher R, Li L-J, Li K, Fei-Fei L (2009). IEEE Conference on Computer Vision and Pattern Recognition 2009: 248–55. IEEE. **b** Visualization of recorded channels in the MNI brain space. The channels are pooled from all participants with iEEG recordings. Colors represent different regions of interest (ROIs). L: left. R: right. **c**–**f** Spatial-temporal characteristics of visually responsive channels. **c** Percentage of visually responsive channels for each ROI in the left (blue) and right (red) hemispheres. FG fusiform gyrus, ITG inferior temporal gyrus, AH anterior hippocampus, PH posterior hippocampus. Asterisks indicate a significant difference in the percentage using a $\chi^2$ test (uncorrected). ****$P < 0.0001$.

FG versus ITG: $P < 10^{-20}$. FG versus PH: $P = 1.26\times10^{-11}$. FG versus AH: $P = 4.84\times10^{-7}$. FG versus amygdala: $P = 1.65\times10^{-9}$. **d** Object category eliciting the maximal response for each visually responsive channel. Color coding indicates object categories. **e** Time course of HGP averaged across all visually responsive channels for each ROI. HGP was normalized to the pre-trial baseline. The red dashed lines indicate the median response onset latency (numbers shown in the title). Error shades denote ±SEM across channels. **f** Response onset latency of visually responsive channels for each ROI (FG: $n = 201$; ITG: $n = 141$; PH: $n = 34$; AH: $n = 47$; amygdala: $n = 78$). Channels in the left and right hemispheres are combined. Each gray circle represents an individual channel. The black circle represents the median, and error bars denote ±SD across channels. Asterisks indicate a significant difference between ROIs using a two-tailed two-sample t-test. *$P < 0.05$, **$P < 0.01$, ***$P < 0.001$, and ****$P < 0.0001$. Source data are provided as a Source Data file.

model where patient index was included as a random variable (Percentage ~ Region + 1|Patient; $P$s < 0.05 for all comparisons between the FG and other regions). Comparable percentages of visually responsive channels were observed in both the left and right hemispheres for each region of interest (ROI; Fig. 2c).

We next analyzed the object category that elicited the greatest HGP for each visually responsive channel (Fig. 2d). We found that the right FG primarily encoded animal/face stimuli (e.g., cat, dog, young mammal, and gymnast [people]; shown in red colors in Fig. 2d) whereas the left FG and the bilateral ITG and MTL regions exhibited more heterogeneous response profiles, with maximal responses to various categories across different channels (Fig. 2d). Specifically, compared with the left FG ($n = 29$, 25.22%), visually responsive channels in the right FG ($n = 44$, 51.16%; $\chi^2$ test: $P = 0.0002$) had a significantly higher proportion of maximal responses to animal stimuli;

and this was particularly the case for people/human faces (Fig. 2d; right FG: $n = 33$, 38.37%; left FG: $n = 10$, 8.70%; $P = 3.85 \times 10^{-7}$), consistent with findings from neuroimaging and lesion studies[36–38] (see Supplementary Results and Supplementary Fig. 1 for detailed analyses of object category selectivity).

To investigate the temporal dynamics of object encoding across the VTC and MTL, we estimated the response onset latency for each ROI using a trial-by-trial analysis (see Methods). The response onset latency in the FG ($138.5 \pm 32.0$ ms [median ± SD]) was significantly shorter than in all other regions (ITG: $150.5 \pm 21.7$ ms; PH: $146.5 \pm 14.9$ ms; AH: $148.5 \pm 15.6$ ms; all $P$s < 0.05; Fig. 2e, f), while the response onset latency in the amygdala ($158.5 \pm 20.5$ ms) was significantly longer than in all other regions (all $P$s < 0.05; Fig. 2e, f). The results remained consistent when using a linear mixed-effects model with patient index included as a random variable (Latency ~ ROI + 1|

Patient; all $P$s < 0.05). Similar results were also derived using response peak latency. Interestingly, given comparable latency, the ITG (primarily anterior ITG in our case) may be involved in visual object processing in parallel with the MTL, despite its anatomical proximity to the FG.

Together, our results delineate the encoding of naturalistic visual objects along the ventral visual pathway.

## Axis-based feature coding in the VTC

Neurons in the macaque inferotemporal cortex have been shown to represent visual faces or objects by parametrically encoding specific visual features[15,17,18], exhibiting axis-based feature coding. To determine whether the human brain also exhibits axis coding, we extracted visual features from each stimulus image using a pre-trained deep neural network (DNN) and modeled the neural responses of each visually responsive channel to these features using partial least squares (PLS) regression[30,32,39] (see Methods for details; see Supplementary Fig. 2a, b for selection and control analyses of different DNN layers).

Indeed, we observed strong axis coding in the bilateral FG (left: $n = 47$, 40.86%, binomial test: $P < 10^{-20}$; right: $n = 38$, 44.19%, $P < 10^{-20}$; see Fig. 3a and Supplementary Fig. 2c for representative channels) and left ITG ($n = 18$, 24.66%, $P < 10^{-20}$; Fig. 3b). The preferred axes of these channels were highly interpretable. For instance, a channel in the FG encoded changes from natural to artificial objects (Fig. 3a), while a channel in the ITG encoded changes along the animacy axis (Fig. 3b; see Methods for the distinction of these axes). Interestingly, the left ITG had a significantly higher percentage of axis-coding channels than the right ITG (Fig. 3c; $\chi^2$ test: $P = 0.002$), indicating laterality in axis coding. Notably, we observed that axis-coding channels were predominantly distributed in the medial VTC rather than the lateral VTC (Fig. 3d). On the other hand, the MTL exhibited minimal axis coding (all binomial $P$s > 0.05, except for the left AH [$n = 4$, 18.18%, $P = 0.004$] and right AH [$n = 4$, 16.00%, $P = 0.007$]; Fig. 3c, d), consistent with results from human single-neuron recordings that showed an absence of axis coding in the MTL[27,28]. All of the above results were further confirmed using quantitative measures of axis-coding strength (Fig. 3d, e; see Methods). Notably, across all visually responsive channels, the axis model explained significantly more variance compared to the category model in both the FG (axis: $0.63 \pm 0.038$ [mean $\pm$ SD]; category: $0.17 \pm 0.087$; two-tailed paired $t$-test: $t(200) = 91.94$, $P < 10^{-20}$, $d = 6.75$, 95% CI = [0.44, 0.46]) and ITG (axis: $0.62 \pm 0.030$; category: $0.12 \pm 0.046$; $t(73) = 63.34$, $P < 10^{-20}$, $d = 12.86$, 95% CI = [0.49, 0.50]; Supplementary Fig. 2d; similar results were obtained when controlling for the input dimensionality of the axis model and the number of object categories used in the category model).

To further characterize the encoded feature axes of the axis-coding channels and compare them with known object dimensions from previous studies[18], we performed PCA on the object visual features to derive orthogonal feature dimensions (principal components [PCs]) from our stimuli. Visualization of the PCs revealed the well-known natural–artificial, spiky–stubby, and inanimate–animate dimensions (i.e., the first three PCs; Supplementary Fig. 3a), consistent with prior studies (see also Methods for the distinction of these dimensions)[18,29]. We then examined how the tuning axis of each axis-coding channel aligned with these feature dimensions using Pearson correlation. We found that across axis-coding brain areas (i.e., FG, ITG, and AH), the vast majority of the tuning axes were correlated with the first three feature dimensions (Supplementary Fig. 3b). In particular, the first PC, representing the natural–artificial dimension, was most frequently (FG: 60.00%; ITG: 56.52%; AH: 75.00%; Supplementary Fig. 3b) and most strongly (Supplementary Fig. 3c) aligned with the extracted neural axes (see Supplementary Fig. 3d for individual channels), followed by the third PC, representing animacy (FG: 28.24%; ITG: 21.74%; AH: 25.00%). Unlike findings from the monkey IT cortex[18], axis-coding channels in the human VTC were less likely—and less strongly—

to encode the spiky–stubby dimension (the second PC; FG: 11.76%; ITG: 21.74%; AH: 0%; Supplementary Fig. 3b, c). These results were further replicated using a separate dataset (Supplementary Fig. 4; see below).

To investigate the temporal dynamics of visual feature encoding along the ventral visual pathway, we measured the response latency of each axis-coding channel. Consistent with visual responsiveness, axis coding began earliest in the FG (151 ms; see Fig. 3f for each ROI in the left hemisphere and Supplementary Fig. 2e for the right hemisphere), followed by the ITG (231 ms) and AH (491 ms). Notably, the onset latency of axis coding varied as a function of the $y$-coordinate in the MNI space within the FG (Fig. 3g), suggesting a local hierarchical processing of visual objects. To control for differences in axis-coding strength across channels (as stronger axis coding may lead to earlier onset), we normalized each channel's axis-coding strength to its maximum before calculating the onset latency of axis coding and obtained similar results.

Together, our results reveal axis-based feature coding in the human VTC for visual object processing, aligning with findings from non-human primates. Latency analysis further suggests a hierarchical progression of axis coding, starting in the FG and moving to the ITG and AH. Importantly, representational similarity analysis (RSA) of the neural pairwise distance of visual objects (see Methods) showed that visual objects were similarly represented in axis-coding channels in the VTC and category-coding channels in the MTL (Fig. 3h; see Supplementary Fig. 5 for individual dissimilarity matrices), indicating information flow between these brain areas. We next focused on the mechanisms underlying the transition of visual object coding from the VTC to the MTL.

## Single neurons in the human MTL are tuned to regions in the VTC neural feature space

We have demonstrated that single neurons in the human amygdala and hippocampus exhibit region-based feature coding. Specifically, these neurons respond to faces or objects that occupy a particular region of the visual feature space, constructed using DNN unit activations[27,28]. DNN units represent linear combinations of visual features, analogous to the neural axes in the VTC (Fig. 3). Therefore, we hypothesized that MTL neurons encode a tuning region / receptive field (i.e., they become selective to stimuli within this tuning region / receptive field) in the VTC neural feature space, with the axes of this space encoded by VTC neurons (Fig. 1). This provides a computational framework linking VTC and MTL representations, allowing us to understand how perceptual representations in the VTC are translated into abstract conceptual representations in the MTL.

To test this hypothesis, we first constructed the VTC neural feature space using axis-coding channels from the VTC (see Methods). We extracted the most preferred tuning axis for each axis-coding channel (i.e., the first PLS component; see Fig. 3a, b for examples) and then pooled these axes across channels (Fig. 1). PCA was applied to these axes to create a two-dimensional neural feature space, representing the primary visual information encoded in the VTC (Fig. 4a; see also Supplementary Fig. 6). Indeed, the VTC neural feature space exhibited an organized structure (Fig. 4a; Supplementary Fig. 6), where the horizontal dimension represented the transition from natural objects to human-made artificial objects, and the vertical dimension reflected changes in animacy. Images from the same object category were clustered together.

To identify MTL region-coding neurons in the VTC neural feature space, we projected an MTL neuron's response to each object onto the VTC neural feature space (i.e., multiplying the firing rate for each object to its position in the VTC neural feature space to create a response-weighted 2D feature map; Fig. 4b, c middle). This revealed that a subset of MTL neurons was selective for objects clustered in specific regions of the VTC neural feature space (Fig. 4b, c middle). We refer to these MTL neurons as exhibiting region-based feature coding

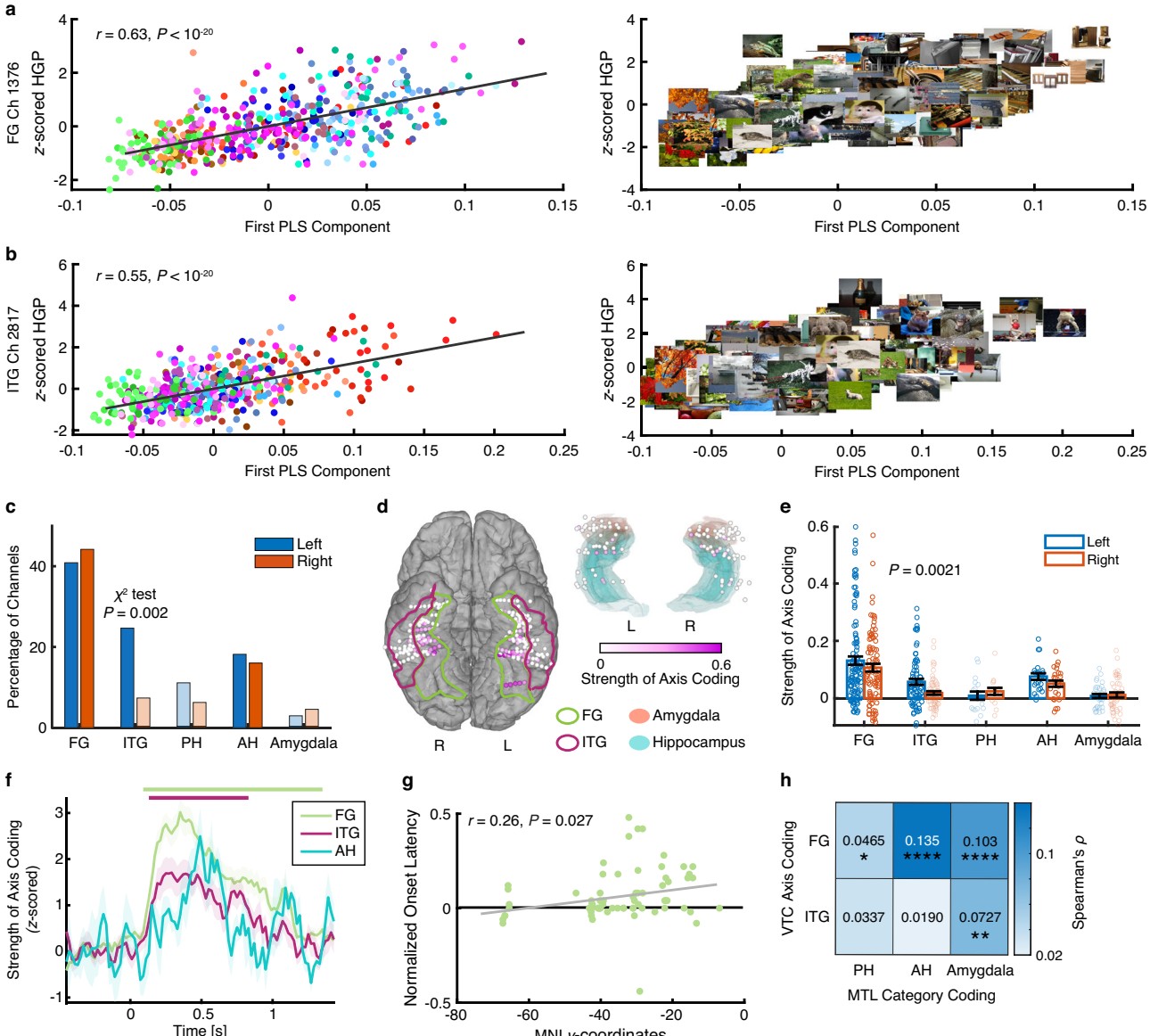

**Fig. 3 | Axis-based feature coding in the VTC. a, b** Example channels exhibiting axis coding. **a** FG. **b** ITG. (left) Pearson correlation between $z$-scored HGP and the first PLS component of the feature map (uncorrected). Each dot represents an object image and color coding denotes the object category. The gray line represents the linear fit. (right) Visualization of the encoded feature axis. Note that both channels showed a significant relationship with the feature map (PLS regression, permutation $P < 0.001$). Images were obtained from Deng J, Dong W, Socher R, Li L-J, Li K, Fei-Fei L (2009). IEEE Conference on Computer Vision and Pattern Recognition 2009: 248–55. IEEE. **c** Percentage of channels exhibiting axis coding. Dark colors represent an above-chance number of selected channels in the corresponding ROIs (binomial test: $P < 0.05$; Bonferroni correction), while light colors indicate chance-level selection. **d** Distribution of axis-coding channels in the VTC and MTL. Color coding shows the strength of axis coding (see Methods). **e** Strength of axis coding averaged across all visually responsive channels in each ROI (FG: $n = 201$; ITG: $n = 141$; PH: $n = 34$; AH: $n = 47$; amygdala: $n = 78$). Dark colors represent

significant ROIs (right-tailed one-sample $t$-test against 0: $P < 0.05$). A two-tailed two-sample $t$-test was performed to compare the strength of axis coding between the left and right hemispheres. **f** Temporal profile of axis coding among the significant ROIs from the left hemisphere. Shaded area denotes ±SEM across axis-coding channels. Top bars indicate time points showing significant axis coding (two-tailed one-sample $t$-test against the mean pre-trial baseline: $P < 0.05$, corrected by FDR[91]) for each ROI. **g** Pearson correlation between the response onset latency and MNI $y$-coordinates of axis-coding channels in the FG ($n = 85$). Each dot represents a channel, and the gray line represents the linear fit. To control for individual differences in response latency, the onset latency for each channel was normalized by subtracting the latency of the channel with the highest axis-coding strength in each session. **h** Representational similarity between axis-coding channels in the VTC and category-selective channels in the MTL. Statistical difference was determined using a permutation test (see Methods). *$P < 0.05$, **$P < 0.01$, ***$P < 0.001$, and ****$P < 0.0001$. Source data are provided as a Source Data file.

in the VTC neural feature space. To formally quantify this tuning (see Methods), we estimated a continuous spike density map within the 2D neural feature space (Fig. 4b, c upper right) by smoothing the discrete firing rate map (Fig. 4b, c middle) using a 2D Gaussian kernel and applied a permutation test (1000 runs; Fig. 4b, c lower right) to identify regions with significantly higher spike densities than expected by chance (outlined and visualized in Fig. 4b, c middle panels; significant

pixels were selected with a permutation $P < 0.01$ and a cluster size threshold). These regions indicate the parts of the VTC neural feature space to which an MTL neuron was tuned.

We recorded from 928 single neurons in the MTL of 20 patients (34 sessions) during the same task as the VTC recordings. We found that 106 out of 928 MTL neurons (11.42%, binomial $P = 2.55 \times 10^{-15}$) exhibited region coding in the VTC neural feature space (see Fig. 4d for

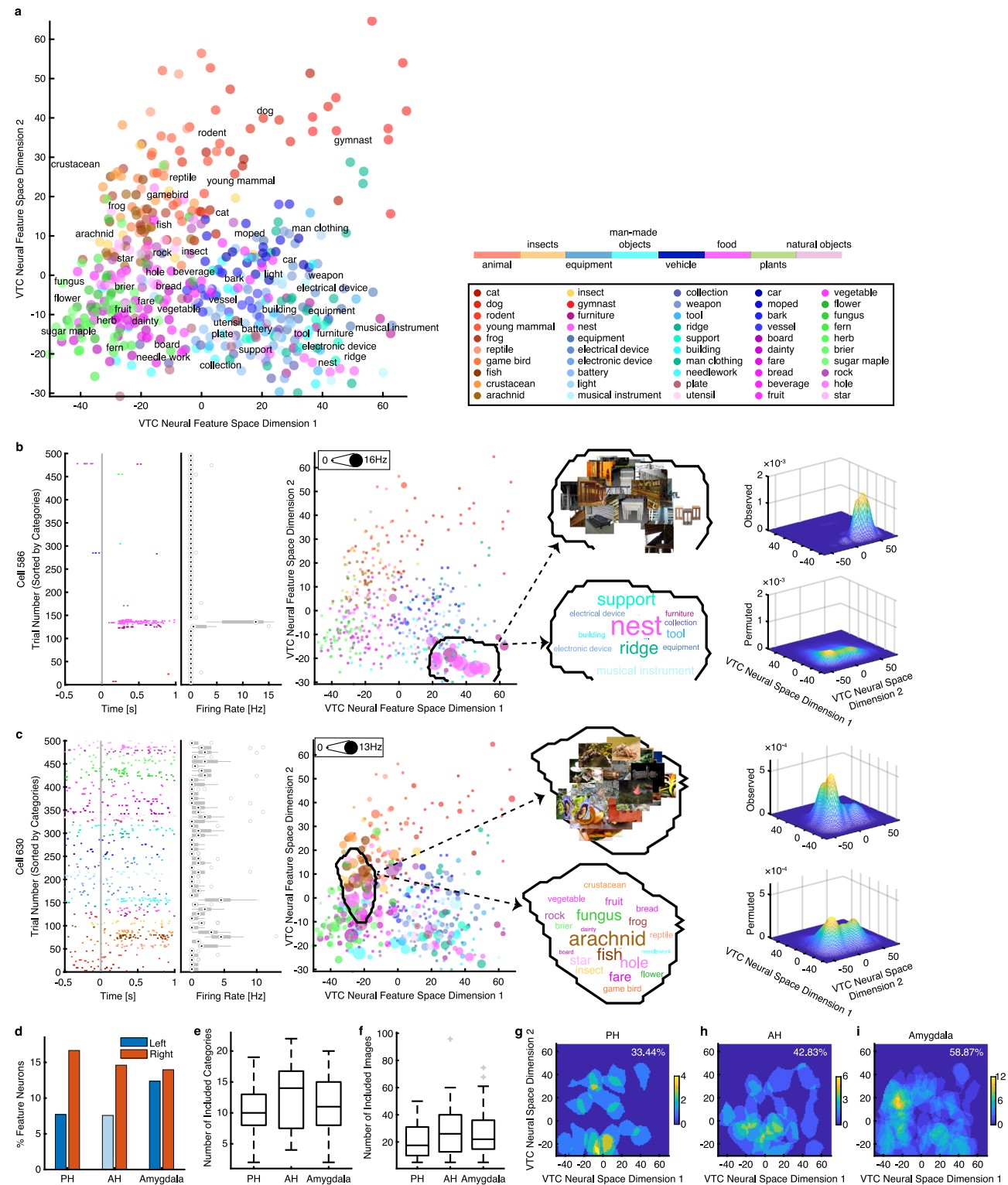

the percentage of selected neurons in each MTL ROI). The number of object categories (Fig. 4e) and object images (Fig. 4f) covered by each tuning region indicates the size of the receptive field of a region-coding neuron within the VTC neural feature space. On average, these regions encompassed 10 to 15 categories and 20 to 30 images, suggesting that some tuning regions may contain only a subset of images from different categories. This image-level, rather than category-level, encoding also indicates that some region-coding neurons are tuned to continuous visual features shared across images. The tuning region of individual neurons spanned approximately 1.73%–7.12% of the neural

feature space (Fig. 4g–i), while the total neural population we sampled covered around 33.44%–58.87% of the neural feature space. Additionally, we tested region coding using signals from iEEG channels in the MTL. However, fewer region-coding channels (16/511, 3.31%) were detected than expected by chance, indicating that region coding did not occur at the mesoscopic level.

Together, our results reveal region-based feature coding of MTL neurons within the VTC neural feature space, providing a computational framework for translating perceptual processing in the VTC into conceptual processing in the MTL.

**Fig. 4 | Region-based feature coding of MTL neurons in the VTC neural feature space. a** VTC neural feature space constructed using all axis-coding channels in the FG. All stimuli are shown in this space. **b**, **c** Two example MTL neurons that encoded a region in the VTC neural feature space. (left) Neuronal responses to 500 objects (50 object categories, with 10 images per category). Trials are aligned to stimulus onset (gray line) and are grouped by individual object category. Boxes indicate the 25th–75th percentiles with the median line, whiskers show non-outlier ranges, and circles mark outliers. (middle) Projection of firing rates onto the feature space, with each color representing a different object category. The size of the dot indicates firing rate. The coding regions are delineated with encompassed object images and categorical labels in the insets, with text size proportional to the number of encoded stimuli within each category. (right) Estimate of the spike density in the feature space. By comparing observed (upper) versus permuted (lower) responses, we could identify a region (black contour in the

middle panel) where the observed neuronal response was significantly higher in the feature space. This region was defined as the tuning region of a neuron. Images were obtained from Deng J, Dong W, Socher R, Li L-J, Li K, Fei-Fei L (2009). IEEE Conference on Computer Vision and Pattern Recognition 2009: 248–55. IEEE. **d**–**i** Population summary of MTL region-coding neurons. **d** Percentage of region-coding neurons in each MTL ROI. **e** The number of categories encoded by region-coding neurons. **f** The number of objects encoded by region-coding neurons (i.e., the number of object images that fell within the tuning region of a region-coding neuron). PH: $n = 18$. AH: $n = 23$. amygdala: $n = 65$. Legend conventions for box plots as in (**b**, **c**). **g**–**i** The population aggregated tuning region in each MTL ROI. Color bars show the counts of overlap between individual tuning regions. Numbers in the density maps show the percentage of the VTC neural feature space covered by the tuning regions of the total observed region-coding neurons. Source data are provided as a Source Data file.

## Validation with an additional dataset

We further validated our findings using an additional dataset comprising Microsoft COCO images[40]. Here, we included 10 object categories but 50 images per category to better examine the neural coding of objects within a category. First, we confirmed axis-based coding in the VTC (including both the FG and ITG; see Fig. 5a for an example and Supplementary Fig. 7 for group results), consistent with our observations from the ImageNet dataset. Notably, the COCO dataset revealed similar feature axes encoded by the axis-coding channels (Fig. 5a right), suggesting that the VTC represents a general visual feature space that does not depend on specific image characteristics.

Importantly, in a subset of 8 patients (13 sessions), we recorded iEEG signals for the COCO images immediately after the ImageNet images, allowing for cross-validation between the datasets. We trained a PLS regression model using the ImageNet dataset and predicted neural responses to the COCO images (see Methods for details). Indeed, we observed a significant number of channels in the bilateral FG (left: $n = 32$, 34.78%, binomial test: $P < 10^{-20}$; right: $n = 23$, 50%, $P < 10^{-20}$; Fig. 5b, c), ITG (left: $n = 15$, 26.79%, $P < 10^{-20}$; right: $n = 13$, 23.64%, $P < 10^{-20}$), and left AH ($n = 5$, 26.32%, $P = 0.0002$), whose responses could be significantly predicted across datasets. Therefore, axis coding in the VTC was generalizable across datasets.

We next constructed the VTC neural feature space (Fig. 5d) using axis-coding channels selected from the above cross-validation procedure (Fig. 5b, c) and tested region coding in MTL neurons ($n = 487$) using the COCO dataset (Fig. 5e–g; similar results were obtained when using neural axes derived solely from the COCO dataset [Supplementary Fig. 7]). Once again, we observed robust region coding. Interestingly, some MTL neurons encoded specific regions containing a subset of images from one or a few categories (e.g., Fig. 5e), indicating that region coding was not dependent on object categories but rather on VTC neural features. At the population level, we found a significant percentage of region-coding neurons in all three subregions of the MTL (Fig. 5f), with our sampled neurons covering 29.14% to 60.73% of the VTC neural feature space (Fig. 5g).

Together, we validated our findings on axis coding in the human VTC and region coding of MTL neurons within the VTC neural feature space using an additional dataset with different stimuli.

## Functional connectivity between the VTC and MTL supports the coding transition mechanism

What is the physiological basis of the above computational framework? To address this question, we conducted a series of functional connectivity analyses between the VTC and MTL at both the neural network and circuit levels.

First, using phase-locking value (PLV) on bipolar re-referenced channel pairs between the VTC (FG and ITG) and MTL (PH, AH, and amygdala; see Methods for details), we observed significant synchronization (primarily in the lower frequency band [<20 Hz]) between VTC

visually responsive channels and MTL category-selective channels in all pairs of VTC-MTL regions (Fig. 6a). A subset of VTC axis-coding channels (59.26%) exhibited significant phase-locking with MTL category-selective channels (Supplementary Fig. 8a). These results suggest that the VTC is functionally connected with the MTL for visual category coding.

In particular, while both VTC axis-coding and non-axis-coding channels increased synchronization with MTL channels in the low-frequency range after stimulus onset, VTC axis-coding channels exhibited significantly stronger synchronization compared to VTC non-axis-coding channels (Fig. 6b), suggesting that they engage the MTL more during object processing. The difference in PLV between the VTC and PH was primarily in the theta (3–8 Hz; two-tailed two-sample $t$-test across channel pairs: $t(115) = 2.22$, $P = 0.029$, $d = 0.48$, 95% CI = [0.01, 0.24]; Fig. 6b) and alpha (7–13 Hz; $t(115) = 2.24$, $P = 0.027$, $d = 0.49$, 95% CI = [0.017, 0.28]; Fig. 6b) frequency ranges, the difference in PLV between the VTC and AH was primarily in the theta frequency range ($t(432) = 4.56$, $P = 6.62 \times 10^{-6}$, $d = 0.45$, 95% CI = [0.057, 0.14]; Fig. 6b), and the difference in PLV between the VTC and amygdala was primarily in the delta frequency range (1–3.5 Hz; $t(253) = 6.15$, $P = 3.05 \times 10^{-9}$, $d = 0.79$, 95% CI = [0.098, 0.19]; Fig. 6b). Furthermore, by separately analyzing each pair of VTC and MTL subregions (Supplementary Fig. 9), we found that the results were largely driven by the FG, consistent with the findings shown in Fig. 3h.

Importantly, we further demonstrated that the strength of axis coding was significantly correlated with the strength of VTC-MTL connectivity specifically for axis-coding channels (Fig. 6c; VTC-AH: $r(160) = 0.32$, $P = 4.32 \times 10^{-5}$; VTC-amygdala: $r(96) = 0.26$, $P = 0.010$), but this was not the case for non-axis-coding channels (Fig. 6c; VTC-AH: $r(270) = -0.094$, $P = 0.12$; VTC-amygdala: $r(155) = 0.035$, $P = 0.66$). Therefore, the stronger interaction between axis-coding channels and the MTL during visual processing, as well as the parametric relationship between axis coding and VTC-MTL connectivity, support the neural mechanism by which the VTC delivers visual feature information to the MTL for its subsequent processing. In addition, we obtained similar results when using ipsilateral channels only (Supplementary Figs. 8b, c and 9b) and when using an equal number of channels.

Second, feedforward and feedback processing may rely on neural oscillations at different frequencies[41,42]. To determine the direction of information flow between the VTC and MTL during visual object processing, we calculated Granger causality (GC) for each VTC (visually responsive)-MTL (category-selective) channel pair. Consistent with the results from the PLV analysis, we observed significant GC in all pairs of VTC-MTL regions (Fig. 6d; see Supplementary Fig. 8d for analysis using ipsilateral channels only). Bidirectional interactions between the VTC and MTL were observed, suggesting that object coding requires both feedforward and feedback processing. Feedforward communication (VTC to MTL) peaked at a lower frequency, whereas feedback communication (MTL to VTC) peaked at a higher frequency (Fig. 6d),

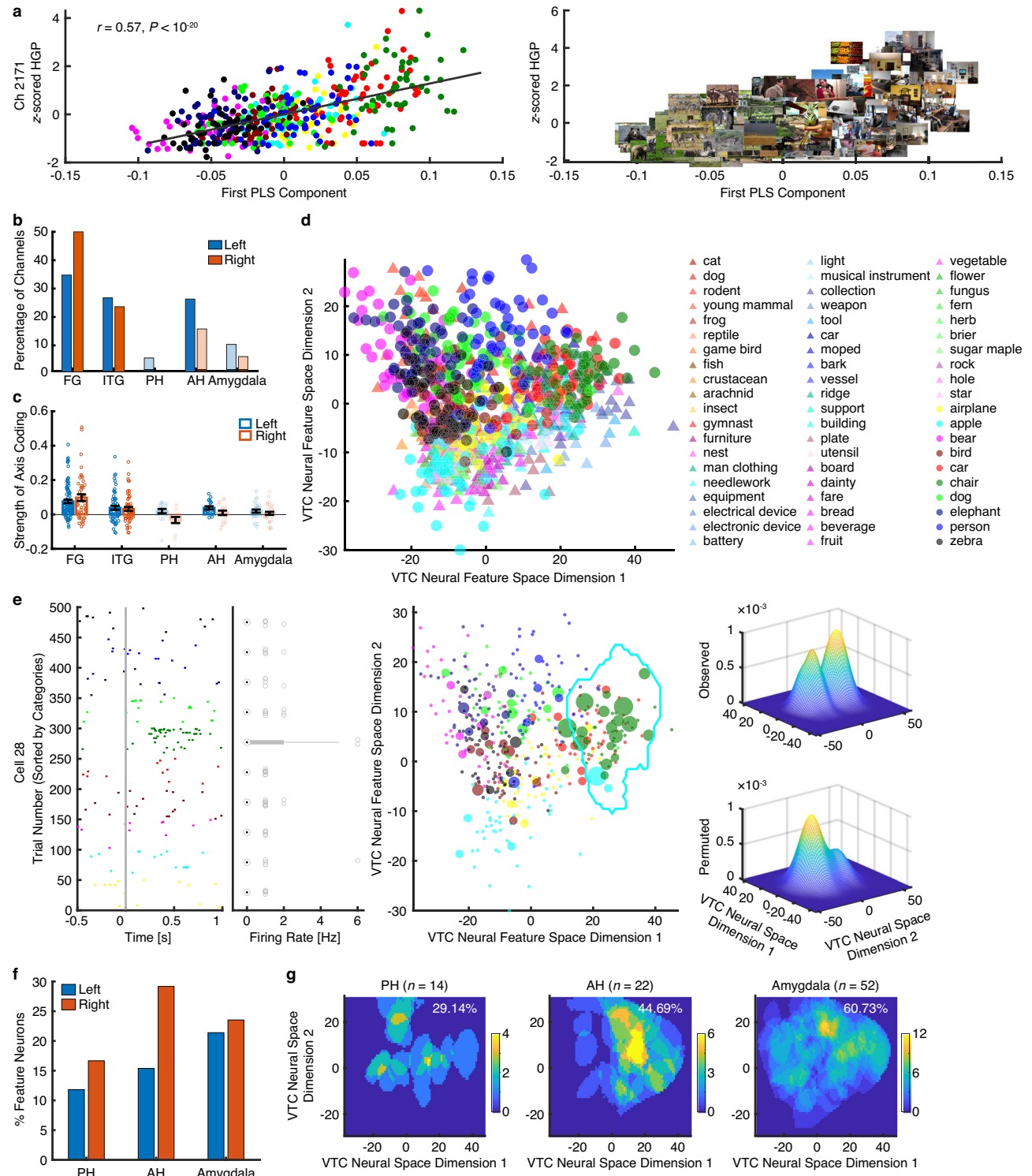

**Fig. 5 | Validation of axis and region coding using the Microsoft COCO stimuli.** **a** An example axis-coding channel (Pearson correlation; uncorrected). Images were obtained from Lin T-Y et al.[40]. **b, c** Generalization of axis coding across datasets. We trained axis-coding models using the ImageNet dataset and predicted neural responses to the COCO images. **b** Percentage of axis-coding channels selected with the cross-validation procedure. **c** Strength of axis coding (Pearson's $r$) averaged across all visually responsive channels in each ROI (FG: $n = 148$; ITG: $n = 107$; PH: $n = 30$; AH: $n = 29$; amygdala: $n = 65$). Error bars denote ±SEM across channels. Legend conventions as in Fig. 3. **d** VTC neural feature space constructed using axis-

coding channels selected from the cross-validation procedure. Triangle: ImageNet stimuli. Circle: COCO stimuli. **e** An example MTL neuron that encoded a region in the VTC neural feature space ($n = 50$ images per object category). Boxes indicate the 25th–75th percentiles with the median line, whiskers show non-outlier ranges, and circles mark outliers. **f** Percentage of region-coding neurons in each MTL ROI. **g** The population aggregated tuning region in each MTL ROI. Note that the same scale is used for the same ROI as in Fig. 4 g–i. Legend conventions as in Fig. 4. Source data are provided as a Source Data file.

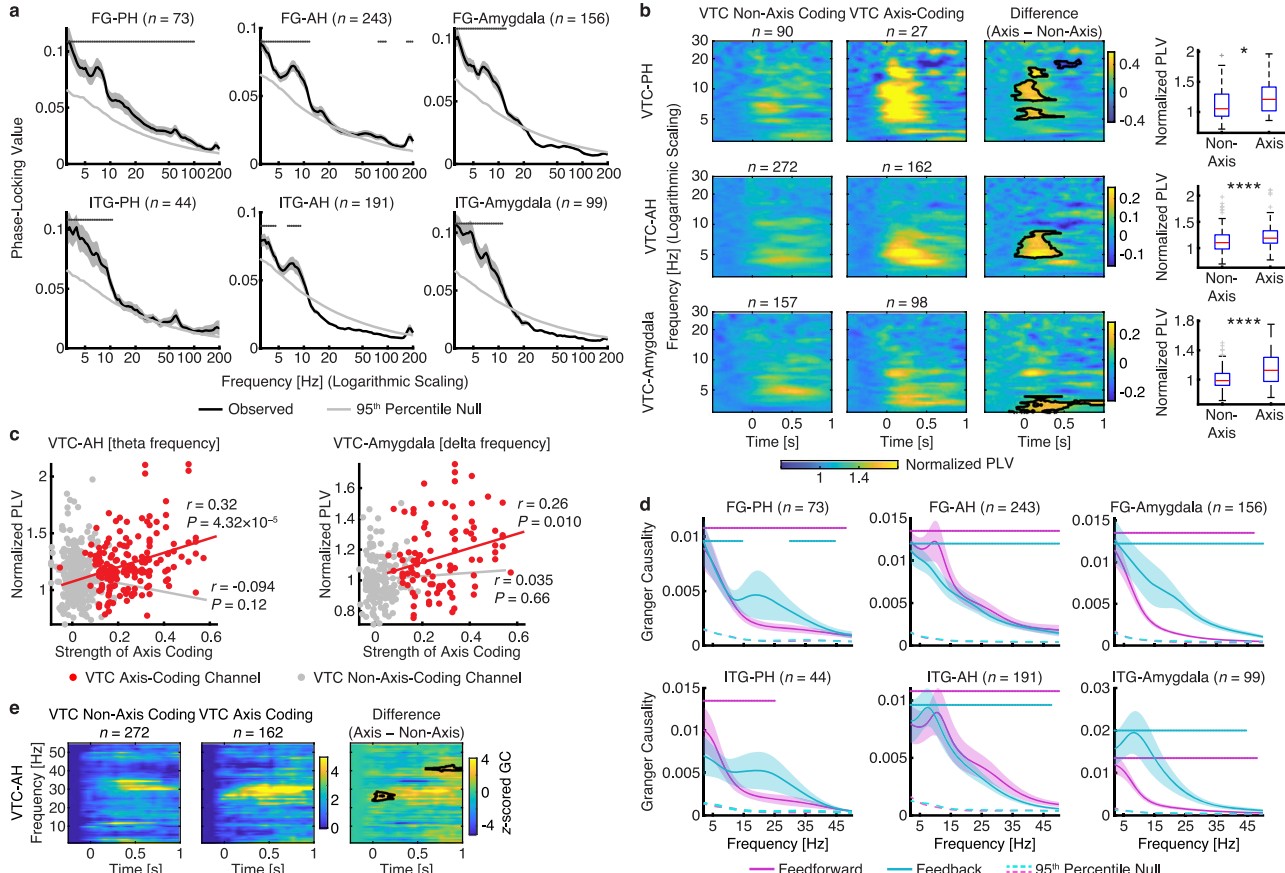

**Fig. 6 | Functional connectivity between the VTC and MTL. a** PLV shows increased VTC-MTL coherence in the lower frequency range. Shaded error bars denote ±SEM across channel pairs. The light gray line indicates the 95th percentile of the null distribution estimated through permutation. The numbers indicate the pairs of channels used. PLV across channel pairs was tested against the 95th percentile of the null distribution using a right-tailed one-sample $t$-test; and dots indicate significant PLV ($P < 0.05$ across 5 consecutive frequencies after FDR correction[91]). **b** Time-resolved PLV shows greater synchronization between axis-coding VTC channels and MTL channels compared to non-axis-coding VTC channels. The PLV was normalized to the pre-trial baseline. The contours indicate areas with a significant difference (two-tailed two-sample $t$-test: $P < 0.05$, uncorrected). The number displayed in each plot indicates the total number of channel pairs. Boxes indicate the 25th–75th percentiles with the median line, whiskers show non-outlier ranges, and crosses mark outliers. Asterisks in the box plots indicate a significant difference between axis-coding and non-axis-coding channel pairs using a two-tailed two-sample $t$-test. *$P < 0.05$, and ****$P < 0.0001$. **c** Pearson correlation between the strength of axis coding and the strength of VTC-MTL connectivity (uncorrected). We used the mean baseline-normalized PLV in the time window from 0 to 0.6 seconds after stimulus onset, in the theta frequency range (3–8 Hz) for VTC-AH channel pairs, and in the delta frequency range (1–3.5 Hz) for VTC-amygdala pairs. Each dot represents a channel pair and the lines represent the linear fit. **d** Bidirectional GC between the VTC and MTL. Shaded area denotes ±SEM across channel pairs. The dashed lines indicate the 95th percentile of the null distribution estimated through permutation. Purple: feedforward (from the VTC to the MTL) GC. Cyan: feedback (from the MTL to the VTC) GC. Dots on the top indicate significant GC at that frequency ($P < 0.05$ across 5 consecutive frequencies after FDR correction[91]). **e** Time-resolved feedforward GC from the VTC to the AH for axis-coding versus non-axis-coding channels. Legend conventions as in (**b**). Source data are provided as a Source Data file.

consistent with previous studies[42]. Importantly, we further showed that axis-coding channels had a stronger feedforward GC from the VTC to the AH in the theta frequency range compared to non-axis-coding channels (Fig. 6e), confirming the feedforward information flow (i.e., providing structured feature information from the VTC to support higher-level processing in the MTL) as shown by the PLV analysis.

Lastly, we investigated whether region coding in the MTL is related to visual processing in the VTC. In 9 patients (13 sessions), we recorded iEEG in the VTC and single neurons ($n = 189$) in the MTL simultaneously, which allowed us to directly test the functional connectivity between MTL region-coding neurons and VTC channels. Specifically, we calculated the pairwise phase coherence (PPC; see Methods) between region-coding neurons ($n = 35$) in the MTL and their corresponding VTC channels ($n = 48$), resulting in a total of 912 pairs. Indeed, we found that MTL region-coding neurons fired spikes that were phase-locked to gamma oscillations in the VTC (see Fig. 7a, b for examples; 12/35 phase-locked to the FG and 22/35 phase-locked to the ITG; for the entire population: 87/189 phase-locked to the FG and 118/

189 phase-locked to the ITG; see Supplementary Fig. 10a–d for analysis in the theta frequency range). Similar to the entire population, the majority of MTL region-coding neurons fired close to the flanks of the gamma frequency oscillation (i.e., 90° or 270°; Fig. 7c) in both the FG and ITG.

Notably, AH region-coding neurons exhibited significantly stronger phase-locking to VTC axis-coding channels than to VTC non-axis-coding channels (Fig. 7d), suggesting that the AH was more tuned to input from the axis-coding channels in the VTC. We further demonstrated that this effect was primarily driven by the FG (Supplementary Fig. 10e), which not only aligns well with the VTC-MTL representational similarity profiles (Fig. 3h) but also further supports a hierarchical processing structure within the VTC-MTL pathway. Furthermore, the PPC of the MTL spike–ITG iEEG pairs was significantly greater for in-region stimuli compared to out-region stimuli in the gamma frequency range (Fig. 7e; two-tailed paired $t$-test across spike–iEEG pairs: $t(503) = 2.01$, $P = 0.045$, $d = 0.09$, 95% CI = [0.0001, 0.01]; a similar trend was observed for the MTL spike–FG iEEG pairs), but not in the

theta frequency range (Supplementary Fig. 10d; $t(911) = 1.73$, $P = 0.083$, $d = 0.06$, 95% CI = [−0.0004, 0.0062]), suggesting that objects encoded by MTL region-coding neurons had more synchronized activity, specifically with gamma oscillations in the VTC.

To illustrate the correspondence between region coding and axis coding, we visualized the in-region stimuli of MTL region-coding neurons along the most-preferred axis of their corresponding phase-locked VTC axis-coding channels (Fig. 7f, g). Interestingly, the in-region stimuli tended to cluster at a specific point along the tuning axis of the phase-locked iEEG channels. This finding suggests that MTL region-coding neurons are tuned to visual features that are parametrically encoded by phase-locked VTC feature-coding channels, providing a physiological basis for functional coupling and supporting our proposed computational framework.

Together, our results revealed robust and bidirectional VTC-MTL interactions during object coding. In particular, VTC axis-coding channels exhibited enhanced synchronization with MTL channels, and notably, this synchronization was parametrically modulated by the strength of axis coding, suggesting that VTC axis-coding channels provide processed visual feature information (i.e., neural axes) to the MTL. On the other hand, MTL region-coding neurons displayed phase-locking to gamma oscillations in the VTC, with stronger phase-locking observed for axis-coding channels and encoded objects. These results establish the physiological basis of our computational framework of neural object coding.

## Discussion

Object recognition is fundamental to our ability to interpret and interact with the world around us. The underlying neural circuits and pathways involve a critical progression of information processing from the VTC to the MTL, where complex visual features are extracted and transformed into meaningful category-specific representations, allowing us to recognize objects regardless of changes in viewpoint, size, or context. To understand the neural computational mechanisms, we conducted a comprehensive investigation into the neural coding of naturalistic visual objects across two datasets using iEEG and single-neuron recordings in the human VTC and MTL. First, we characterized the spatiotemporal patterns of object representations in the VTC, which notably exhibited axis-based feature coding, similar to that observed in the macaque inferotemporal cortex[15–18]. Second, we showed that MTL neurons encoded a receptive field (i.e., coding region) within the VTC neural feature space, which, in turn, accounts for the sparse coding properties in the MTL[28]. Importantly, this result led to a computational framework linking visual processing to conceptual encoding in the brain. Third, we validated our findings using an additional dataset with different stimuli. Lastly, we found robust interactions between the VTC and MTL during object coding, reinforcing the notion of coordinated neural processing between these regions. In particular, supporting the physiological basis of the computational framework, VTC axis-coding channels were more strongly connected with the MTL (than VTC non-axis-coding channels) to provide visual feature information, and MTL region-coding neurons exhibited synchronization with gamma oscillations in the VTC. Together, our study reveals a computational framework that explains the transition of visual coding from dense feature-based to sparse concept-based representations, providing a mechanistic understanding of the neural processes underlying object recognition.

Previous human iEEG studies on object coding have mostly focused on visual category selectivity in the VTC[43–46]. In this study, we first demonstrated the topographical organization of visual representations for a large set of object categories at a fine-grained scale (Fig. 2b–d), as illustrated in primate physiology[47] and human fMRI studies[48,49]. Importantly, as one of the first studies, we revealed axis-based feature coding of objects in the human VTC (including both the

FG and ITG), as previously reported in non-human primates[15–18]. Notably, this robust axis coding was observed at the mesoscopic neural population level (in contrast to the microscopic neuronal level shown in primate studies[18]), indicating that the representation of visual features may involve the coordinated activity of larger groups of neurons. The preferred dimensions of these neural populations represented visual changes related to variations in animacy and artificiality (Supplementary Fig. 3; Fig. 4a; see also Methods), consistent with primate studies[18,29]. However, we observed weaker encoding of the stubby–spiky dimension compared to what was reported in ref. 18 (Supplementary Figs. 3, 4; Fig. 4a), a discrepancy likely due to differences in the stimuli used across studies (see also Supplementary Fig. 3 versus Supplementary Fig. 4). Specifically, the primate study used simple, isolated objects, whereas our study examined natural scene objects with rich backgrounds. It is worth noting that DNNs were employed for feature extraction and PCA was used for dimensionality reduction in both cases (ref. 18 used PCA for DNN features, while we used it for neural axes). Furthermore, extending findings from primate studies[18], our time-resolved feature coding analysis revealed the temporal dynamics of neural coding across different brain areas. It also demonstrated the propagation of visual feature encoding along the ventral pathway, consistent with the hierarchical processing of visual objects[50,51]. Additionally, our recordings covered the anterior FG and ITG—areas that have been less discussed due to challenges in obtaining reliable signals[52]. Lastly, it is worth noting that axis-coding channels from different brain areas may encode distinct visual features, as primate neurons tuned to different axes can be localized in distinct regions of the inferotemporal cortex[18].

The human MTL plays a critical role in recognition memory[53–58]. A key aspect of the MTL's ability to support this function is the formation of a highly sparse code at the level of single neurons[23–26]. Such neurons have two prominent properties: (1) their representations are invariant to visual features[25,26], and (2) they encode conceptually related rather than visually related stimuli[59,60]. However, it remains largely unknown how these properties emerge. The MTL is only a few synapses downstream of the higher visual cortex, where feature-based coding (i.e., axis coding), very different from that in the MTL, has been shown in many primate studies[15–18]. To address this question, our recent studies[27,28] uncovered a novel region-based coding mechanism in the human MTL that could explain the transition from dense feature-based representations in the higher visual cortex to sparse concept-based representations in the MTL. Specifically, MTL neurons receive processed visual input from the VTC and encode receptive fields within the high-level feature space, allowing them to selectively respond to stimuli that fall within these receptive fields. Our present study provides direct evidence to support this hypothesis.

While extensive studies have highlighted the critical role of the VTC in object perception[5,12,18], emerging evidence supports the involvement of the MTL in this process[61,62]. Nevertheless, studies investigating how these brain areas communicate and coordinate during object recognition remain scarce. Our study is among the first to examine the interactions between the VTC and MTL during object processing, guided by a computational framework of the neural circuit between these brain areas. Notably, our comprehensive functional connectivity analyses at both the neural network and circuit levels yielded results that strongly corroborated each other. Specifically, we demonstrated not only the information flow from the VTC to the MTL but also that VTC-MTL synchronization was parametrically modulated by the strength of axis coding, suggesting that the VTC projects visual feature information into the MTL. Furthermore, the VTC-MTL interaction was enhanced for in-region stimuli encoded by region-coding neurons compared to out-region stimuli, suggesting that the MTL encodes visual objects through enhanced synchronization with the VTC to receive visual information. Therefore, these results provide a mechanistic understanding of the neural network and circuits

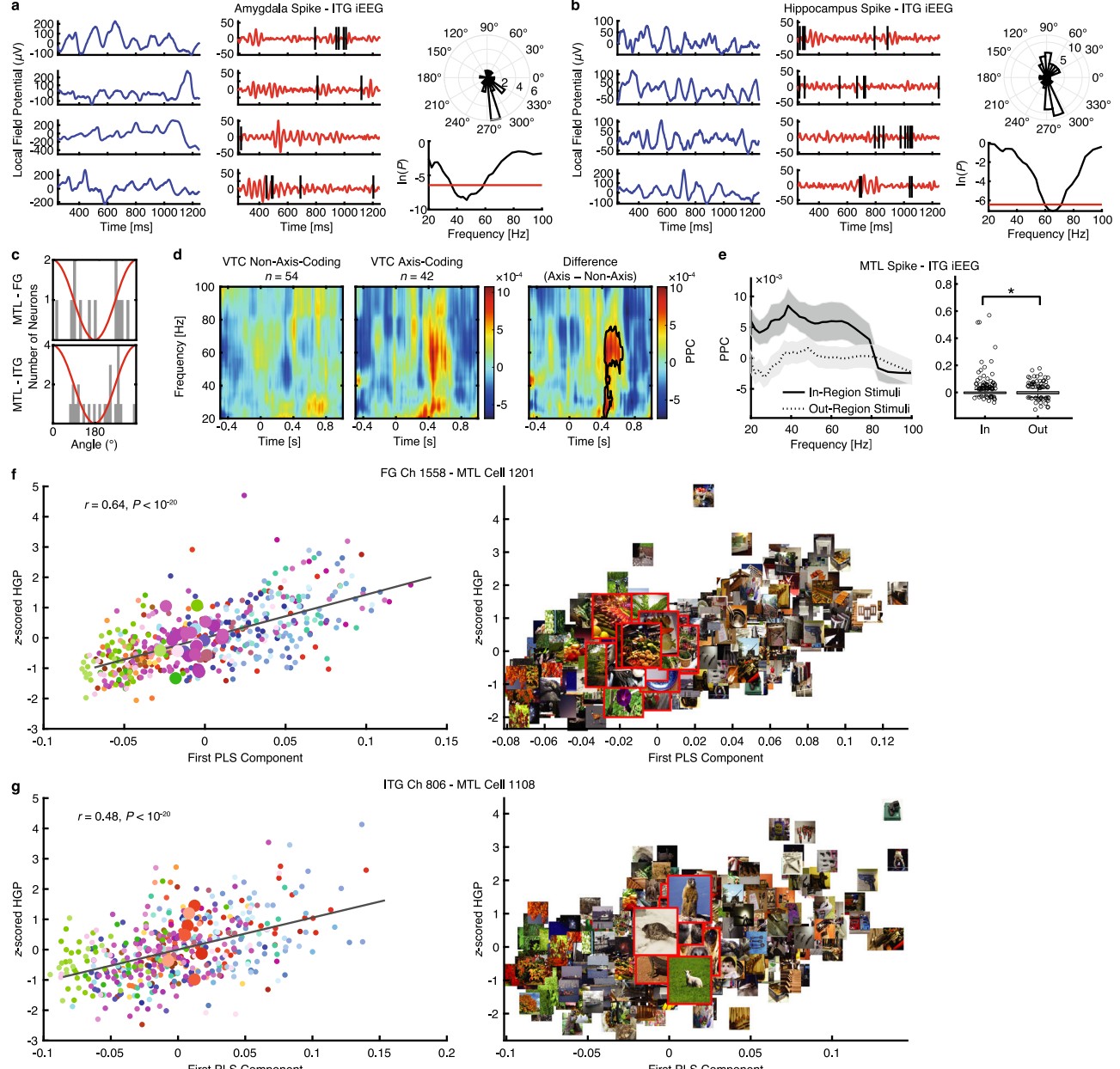

**Fig. 7 | Region coding and gamma phase-locking (30–80 Hz) between the VTC and MTL. a, b** Two example region-coding neurons from the MTL exhibiting phase-locking to ITG gamma oscillations. **a** Amygdala neuron. **b** Hippocampal neuron. (left) Raw iEEG signals. (middle) Spikes (vertical lines) on the gamma component of the iEEG signal. Each row represents a trial segment (time 0 is stimulus onset). (upper right) Distribution of spike phase for the averaged gamma oscillation. (lower right) Significance of phase-locking (right-tailed Rayleigh test) as a function of frequency (20–100 Hz). The threshold (red line) for significant phase-locking was set to $P = 0.0016$ (0.05/31, Bonferroni-corrected). **c** Distribution of the preferred phase of MTL region-coding neurons that were phase-locked to VTC gamma oscillations (MTL-FG: $n = 12$; MTL-ITG: $n = 22$). The red line indicates the phase notation used. **d** Time-resolved pairwise phase consistency (PPC) for axis-coding versus non-axis-coding channels. PPC was calculated using spikes from AH region-coding neurons and VTC iEEG. Legend conventions as in Fig. 6b. **e** PPC of the MTL spike–ITG iEEG pairs ($n = 504$). Spikes were from MTL region-coding neurons. Error

shades denote ±SEM across spike–iEEG pairs. Box plots show average PPC across the gamma frequency range (30–80 Hz). Boxes indicate the 25th–75th percentiles with the median line, whiskers show non-outlier ranges, and circles mark outliers. Asterisks indicate a significant difference between in-region stimuli and out-region stimuli using a two-tailed paired $t$-test. *$P < 0.05$. **f, g** Illustration of two representative MTL region-coding neurons and their corresponding phase-locked VTC axis-coding channels. (left) The $z$-scored HGP changed as a linear function of the first PLS component of the feature map. Each dot represents an object image and color coding denotes the object category. The gray line represents the linear fit. Larger dots correspond to images encoded by the example region-coding neuron. (right) Visualization of the encoded feature axis. Images encoded by the example region-coding neuron are highlighted with a red frame and shown at a larger size. Images were obtained from Deng J, Dong W, Socher R, Li L-J, Li K, Fei-Fei L (2009). IEEE Conference on Computer Vision and Pattern Recognition 2009: 248–55. IEEE. Source data are provided as a Source Data file.

underlying object processing. It is worth noting that region-based coding operates on a low-dimensional space, likely constructed by reducing the dimensions of the visual feature space represented in the VTC. This may be a key intermediate mechanism that solves the many-to-one problem in object recognition[63]: assigning a specific label to

many different visual forms of an object, such as leaves in different colors and shapes. It is also worth noting that while we focused on the amygdala and hippocampus in this study, other subregions of the MTL may exhibit gradual changes in coding schemes within the neural pathway for object processing[64].

Our findings align with the known anatomical connections between the VTC and MTL and the neural pathway for object processing[11], suggesting the neural basis of the computational framework for translating perceptual information into conceptual representations. Interestingly, feedforward and feedback processing may engage neural oscillations at different frequencies[41,42]. Specifically, our results align well with a previous study[42] showing that in the primate visual system, feedforward influences are carried by theta-band (~4 Hz) and gamma-band (~60–80 Hz) synchronization, while feedback influences are carried by beta-band (~14–18 Hz) synchronization. In our study, we found VTC-MTL iEEG phase-locking (Fig. 6a–c) and Granger causality (Fig. 6d) in the theta frequency range and spike–iEEG phase-locking in the gamma frequency range (Fig. 7) when the VTC fed forward information to the MTL, whereas feedback Granger causality from the MTL to the VTC was found in the beta frequency range (Fig. 6d; Supplementary Fig. 8d). Our results also suggest a multi-frequency communication mechanism between the VTC and MTL, with gamma frequencies supporting feature encoding and theta frequencies facilitating integration into memory networks[65,66]. While MTL neurons exhibited phase-locking with VTC iEEG signals in the theta frequency range (Supplementary Fig. 10a–c), the PPC of MTL spike–VTC iEEG pairs was not significantly greater for in-region stimuli compared to out-region stimuli (Supplementary Fig. 10d). This further supports the idea that gamma oscillations may play a dominant role in MTL neurons during the feature coding of visual objects. Future studies are needed to better understand this object processing network[67], particularly at the neural circuit level and with regard to the feedback influence of the MTL on the VTC, which may be related to processes such as stimulus imagination[68].

In conclusion, the transition of visual coding from the VTC to the MTL is a crucial process for transforming detailed visual information into higher-order, abstract representations. In the VTC, visual features are densely encoded, capturing the fine-grained details necessary for object identification. As this information is relayed to the MTL, it becomes more abstract and sparsely represented, enabling the brain to recognize objects regardless of variations in appearance[25,26]. The nature of this transition is such that MTL neurons encode a receptive field within the VTC neural feature space, making them selective to stimuli that fall within this receptive field. Our computational framework thus serves as a bridge between VTC and MTL representations, allowing us to decipher how perceptual representations in the VTC are translated into conceptual representations in the MTL. Understanding this transition sheds light on the fundamental mechanisms of object recognition and provides insights into how our brains make sense of complex visual environments. Future studies are needed to explore how this neural computational framework supports memory formation and decision making.

## Methods

### Participants

We recorded intracranial electroencephalography (iEEG) and/or single-neuron activity from neurosurgical patients with pharmacologically intractable epilepsy. All participants provided written informed consent under procedures approved by the Institutional Review Boards of Washington University in St. Louis (WUSTL) and West Virginia University (WVU).

For the ImageNet task, we recorded iEEG from 14 patients (9 females; 21 sessions in total; Supplementary Table 1). In 19 of these sessions, we simultaneously recorded single-neuron activity in the amygdala and hippocampus along with iEEG. In an additional 7 patients (15 sessions), we recorded only single-neuron activity in the amygdala and hippocampus.

For the Microsoft COCO task, we recorded iEEG from 11 patients (7 females; 16 sessions in total). In all sessions, we simultaneously recorded single-neuron activity along with iEEG. In an additional 5

patients (5 sessions), we recorded only single-neuron activity in the amygdala and hippocampus. Notably, 13 iEEG sessions from 9 patients were recorded immediately following the ImageNet task.

### Data acquisition and preprocessing

We recorded iEEG signals using clinical depth macro-electrodes (Ad-Tech, PMT, or DIXI), with each electrode containing 4–18 channels, and used a Nihon Kohden recording system. Data were analog-filtered above 0.01 Hz and digitally sampled at 2000 Hz during acquisition. Preprocessing was performed using EEGLAB[69]. First, the data were visually inspected, and channels contaminated by epileptic or artifactual activity were excluded from further analysis. Second, a 0.5 Hz high-pass filter and a common average reference were applied, followed by a notch filter to remove line noise (60 Hz and its harmonics at 120 Hz and 180 Hz). High-amplitude noise events, as well as interictal discharges, were identified on a trial-by-trial basis using a threshold set according to the procedure described in ref. 70. We then extracted high-gamma power (HGP) by applying a bandpass filter (70–170 Hz) and taking the absolute value of the Hilbert transform of the resulting signal. For connectivity analysis between iEEG channels, we used bipolar re-referencing instead of a common average reference to avoid artifacts introduced by the common reference. All other preprocessing procedures remained the same for all analyses. For further analysis, the data were segmented into event-related epochs around stimulus onset (−0.5 to 1.5 seconds relative to stimulus onset), and we extracted the average neural response for each object image from 0.1 to 0.6 seconds after stimulus onset.

We recorded single-neuron signals using microwires embedded in the hybrid depth macro-electrodes (Ad-Tech Behnke-Fried electrodes) implanted in the amygdala and hippocampus. From each microwire, we recorded the broadband extracellular signal (0.1–9000 Hz) with a sampling rate of 32 kHz, and the data were stored continuously for offline analysis using either a Blackrock (WUSTL) or Neuralynx (WVU) system[71]. For sessions with concurrent recordings, the iEEG and single-neuron signals were time-synchronized using a photodiode patch attached to the stimulus screen. The raw single-neuron data were filtered with a zero-phase lag 300–3000 Hz bandpass filter, and spikes were sorted using a semi-automatic template matching algorithm as previously described[72]. For each neuron, the mean firing rate within 0.25 to 1.25 seconds after stimulus onset was extracted for each object for further analysis.

### Electrode localization

We estimated electrode locations based on pre-operative high-resolution T1-weighted MRI scans and post-operative CT scans. The CT images were co-registered with the MRI scans for patient-specific electrode localization using the VERA software suite (https://github.com/neurotechcenter/VERA). Each channel was labeled with a specific anatomical area derived from Freesurfer's automatic segmentation[73]. Electrode locations were then normalized to the MNI space through nonlinear co-registration for visualization.

It is worth noting that the channels in the current study covered the anterior FG and ITG, brain areas that are under-explored due to severe BOLD signal distortion in fMRI studies[52,74]. Therefore, our results complement neuroimaging studies by revealing the functional organization of the anterior FG, ITG, and MTL areas.

### Stimuli

We used two sets of stimuli. For each set of stimuli, we used the same images for all participants. The choice of object categories was guided by the need for a diverse and representative set encompassing a broad range of categorical types and visual properties.

For the ImageNet stimuli[31], we selected 50 categories of objects with 10 images for each object category. The object categories included arachnid, battery, bark, beverage, board, bread, brier, building,

car, cat, collection, crustacean, dainty, dog, electrical device, electronic device, equipment, fare, fern, fish, flower, frog, fruit, fungus, furniture, game bird, gymnast, herb, hole, insect, light, man clothing, moped, musical instrument, needlework, nest, plate, reptile, ridge, rock, rodent, star, sugar maple, support, tool, utensil, vegetable, vessel, weapon, and young mammal. These categories were selected to sample broadly across multiple conceptual dimensions, including animacy (e.g., animate: dog, frog, rodent; inanimate: car, plate, tool) and naturalness (e.g., natural: tree bark, fruit, rock; artificial: electronic device, musical instrument, clothing). By including both animate and inanimate stimuli as well as natural and manmade objects, the stimulus set allowed us to disentangle these two key but often confounded dimensions of object representation (Supplementary Fig. 3). In addition, the categories varied in terms of taxonomic class (e.g., animals, plants, artifacts), functional role (e.g., tools, furniture, clothing), and perceptual properties (e.g., shape complexity, texture, material composition). Importantly, these dimensions align with known object dimensions reported in prior studies[18,29], including animacy versus inanimacy, natural versus artificial, and shape-based features such as spikiness versus stubby/rounded forms. This alignment provides a principled basis for interpreting neural responses along perceptually and conceptually meaningful axes (see Supplementary Fig. 3), as established in previous work on object representation in the brain[18,29].

For the Microsoft COCO stimuli[40], we selected 10 object categories, with 50 images per category. The selected categories were: airplane, apple, bear, bird, car, chair, dog, elephant, person, and zebra. These categories were chosen to form a well-controlled stimulus set that maintained diversity across both animate and inanimate as well as natural and artificial dimensions. Specifically, the set includes animate natural entities (e.g., bear, elephant, person, zebra), inanimate artificial objects (e.g., airplane, chair, car), and inanimate natural objects (e.g., apple). This selection again allowed us to sample from key conceptual and perceptual dimensions known to shape object representation in the brain (see Supplementary Fig. 4)[18,29]. By incorporating both categorical and perceptual variation, the stimulus set supports analyses of how neural activity reflects structured dimensions of real-world object space.

Together, the diversity in both datasets ensured that our findings were not biased toward a specific subset of object categories and that the computational principles we identified were generalizable across different domains of object recognition. Additionally, our category choices aligned with previous studies investigating object representations in the higher visual cortex, allowing for meaningful comparisons with prior literature[18,29].

### Experimental procedure

Stimuli were presented on a flat monitor screen in front of the patients, who sat comfortably on the bed in a patient room. We used a one-back task for each set of stimuli. In each trial, a single object image was presented at the center of the screen for a fixed duration of 1 second, with a uniformly jittered inter-stimulus interval (ISI) of 0.5 to 0.75 seconds. Each image subtended a visual angle of approximately 10°. Patients pressed a button if the current image was identical to the immediately previous image, with one-back repetitions occurring in 9% of trials. Each image was shown once unless repeated in one-back trials; and responses from one-back trials were excluded to ensure an equal number of responses for each image. This task kept patients engaged with the images while avoiding potential biases from focusing on specific image features[32].

### Extraction of visual features

We employed two well-known deep neural networks (DNNs), ResNet-101[75] and AlexNet[76], to extract visual features for each ImageNet and COCO image. Following the same procedure[27,28], we fine-tuned the top layer of each DNN to confirm that the pre-trained models could

discriminate between objects and to ensure their suitability as feature extractors. We also used the fine-tuning accuracy to determine the most appropriate model for feature extraction. Additionally, to construct a common feature space using images from both the ImageNet and COCO datasets, we extracted features for 1000 images using AlexNet. It is worth noting that neither feature extraction nor the construction of feature spaces utilized any information from MTL neurons; therefore, the clustering of neurally encoded categories in feature spaces was not by construction.

### Visually responsive iEEG channels

We used the mean HGP in the time window from 0.1 to 0.6 seconds after stimulus onset as the neural response to each stimulus. A channel was considered visually responsive if it showed a significantly different response compared to the baseline (0.5 to 0.02 seconds before stimulus onset).

### Category-selective coding

We selected iEEG channels that demonstrated category-selective coding among those responsive to at least one category. Specifically, we first used a one-way ANOVA to identify channels with significantly different responses to object categories. Next, we applied an additional criterion to identify the *selected categories*: the neural response to a category had to be 1.5 standard deviations (SD) above the mean of neural response from all categories during baseline. These object categories, whose responses stood out from the global mean, were considered the encoded categories. We refer to channels encoding a single object category as single-category (SC) channels and those encoding multiple categories as multiple-category (MC) channels. Our previous study demonstrated that this procedure effectively identifies category-selective coding[27,28].

### Axis-based feature coding

To identify iEEG channels that demonstrated axis-based feature coding (i.e., encoding a linear combination of visual features), we employed partial least squares (PLS) regression using DNN feature maps. Axis-coding channels were selected from among the visually responsive channels, as this method assumes the encoding of visual features. The PLS method has been shown to effectively study neural responses to DNN features[17,30]. We used 4 components for each layer, which explained at least 80% of the variance and were determined through 10-fold cross-validation to minimize prediction error. The most preferred tuning axis for each axis-coding channel (i.e., the first PLS component) was obtained using all images.

To determine statistical significance, we used a permutation test with 1000 runs to assess whether a channel encoded a significant axis-coding model. In each run, we randomly shuffled the object labels and used 50% of the objects as the training dataset. We constructed a model by deriving regression coefficients using the training dataset, predicted responses for each object in the remaining 50% of objects (i.e., test dataset), and computed the Pearson correlation between the predicted and actual responses in the test dataset. The distribution of correlation coefficients computed from the shuffled data (i.e., the null distribution) was then compared to the correlation from the unshuffled data (i.e., the observed response). If the correlation coefficient of the observed response was greater than 95% of those from the null distribution, the axis-coding model was considered significant. This procedure has been shown to be highly effective in selecting neurons with significant axis-coding models[15]. The correlation coefficient also indicated the model's predictability, which is the strength of axis coding, and could thus be compared across different channels.

To cross-validate axis-coding models across datasets, we extracted visual features using the same DNN (i.e., AlexNet) for both ImageNet and COCO images. For each channel, we trained a PLS regression model using all images from the ImageNet dataset and used

the obtained regression coefficients to predict neural responses to the COCO images. We then computed the Pearson correlation between the predicted and actual responses to the COCO images and used the permutation procedure (1000 runs) described above to assess statistical significance and strength of axis coding. The most preferred tuning axis for each axis-coding channel (i.e., the first PLS component) was obtained using images from the combined ImageNet and COCO datasets ($n = 1000$ images).

To test the temporal characteristics of axis-based feature coding, we fitted the PLS model to the time series of axis-coding channels. For computational efficiency, the raw time series were first binned using a sliding window of 100 ms with a 20 ms overlap. We then computed the strength of axis coding by fitting the PLS model at each time point using a 10-fold cross-validation procedure. This process was repeated twice, and the strength of axis coding was averaged across repetitions for each time point. The resulting time course of axis coding was then z-scored relative to the pre-trial baseline (−0.5 to −0.02 seconds) for each channel. The significance of axis coding at each time point was compared to the mean baseline performance using paired t-tests across channels (Fig. 3f). We retained only consecutive significant data points lasting longer than 5 bins (100 ms).

### Region-based feature coding in the VTC neural feature space

We first constructed a neural feature space using axis-coding channels from the VTC. For each axis-coding channel, we extracted its most preferred tuning axis (derived from the PLS regression; see above) and pooled these axes across channels from a brain area. Next, we applied PCA on these axes to obtain a two-dimensional neural feature space, which represented the primary visual information encoded in that brain area. In Fig. 4, we included all axis-coding channels from the FG, as axis coding was most prominent there (Fig. 3c–e). However, we obtained similar results when including axis-coding channels from both the FG and ITG.

To select MTL neurons demonstrating region-based feature coding in the VTC neural feature space, we first estimated a continuous spike density map in the neural feature space by smoothing the discrete neural response map with a 2D Gaussian kernel. The kernel size was proportional to the number of clusters (i.e., images from the same category were grouped together) within each feature space, the feature space dimension, and an empirical scaling factor (sq) estimated for each feature space (ImageNet: sq = 0.021; COCO: sq = 0.05). We then assessed statistical significance for each pixel using permutation testing: in each of 1000 runs, we randomly shuffled the object labels. The p-value for each pixel was calculated by comparing the observed spike density to the null distribution derived from the permutations. A mask was applied to exclude edge and corner pixels of the spike density map where no objects were present, as these regions were prone to false positives. We selected the regions containing significant pixels (permutation $P < 0.01$, cluster size >2.5% of pixels within the mask). If an MTL neuron contained a region with significant pixels, it was considered to demonstrate "region-based feature coding" in the VTC neural feature space. Our previous studies showed that this procedure is effective in identifying neurons with region-based feature coding[27,28].

It is worth noting that if the encoded objects are not distributed homogeneously (or uniformly) in the neural feature space, a systematic pattern (e.g., a peak) may appear in the permuted distribution. However, we have shown that our procedure reliably identifies coding regions even under heterogeneous distributions[27].

### Depth of selectivity (DOS) index

To summarize the response of category-selective channels, we quantified the depth of selectivity (DOS) for each channel:

$DOS = \frac{n - (\sum_{j=1}^{n} r_j)/r_{\max}}{n-1}$, where $n$ is the number of categories ($n = 50$), $r_j$ is the mean response to category $j$, and $r_{max}$ is the maximal mean response across all categories. DOS varies from 0 to 1, with 0 indicating an equal response to all categories and 1 exclusive response to one category, but not to any of the other categories. Thus, a DOS value of 1 is equal to maximal sparseness of category coding. The DOS index has been used in many prior studies investigating visual selectivity[77–79].

### Response latency

We estimated the onset latency of task-induced HGP using a trial-by-trial method established in previous studies[80,81]. To avoid potential confounds from temporal smoothing, we conducted the analysis on unsmoothed and non-downsampled data. The signal was first binned using a sliding window of 30 ms (60 sample points) with a 4-ms step size. For each trial, responsive data points were identified by thresholding the binned data at the average pre-trial baseline (−0.3 to −0.02 seconds) across all trials plus one standard deviation. We retained only consecutive responsive data points lasting longer than 25 bins (100 ms). The response onset latency was defined as the first responsive data point in each trial, and we estimated the median latency across trials for each channel. Trials that failed to detect the latency (i.e., did not meet the 25 consecutive responsive bins criterion) were excluded. Only channels with more than 50 trials that could obtain onset latency were included in the final analysis.

We used a similar procedure to compute the latency for category-selective coding. For each category-coding channel, we performed a one-way ANOVA across 50 categories on the binned data and identified significant time points based on the p-value ($P < 0.05$). We retained only consecutive significant data points that lasted longer than 25 bins (100 ms), and the first significant data point was defined as the category-selective onset latency for the given channel.

Similarly, the latency for axis coding was estimated using the p-value from the t-test, with a threshold of 7 consecutive bins (140 ms) set for each axis-coding channel. To test whether the latency of axis coding is related to anatomical location, we normalized the latency of each channel to the best-performing channel within each session. This procedure effectively controlled for individual differences in response latency across participants[81]. The latency was then correlated with the y-coordinate values in the MNI space across channels. Note that this analysis was restricted to the FG, as other significant axis-coding ROIs had very few channels (ITG: $n = 18$; AH: $n = 4$) with detectable onset latency.

We acknowledge that the estimation of latency is subject to imprecisions introduced by temporal smoothing related to the binned HGP signal. Therefore, the absolute latency value should be interpreted with caution, as different methods may lead to heterogeneous values[43].

### Representational similarity analysis (RSA)

We employed RSA[32,82,83] to compare the neural representations in the VTC and MTL areas. For a given ROI, we constructed an image-by-image dissimilarity matrix (DM) using the pairwise Euclidean distances between object images based on the responses of the selected channels. Specifically, for each pair of images, we calculated the Euclidean distance of the neural responses to these two images across all axis-coding or category-coding channels in the ROI. The representational similarity between two ROIs was then computed by correlating the two DMs (e.g., the FG axis-coding DM and the AH category-coding DM) using Spearman correlation, which does not assume a linear relationship. We employed a Mantel test[84] to determine the statistical significance of the RSA. Specifically, we estimated a null distribution by permuting the rows and columns of one DM prior to correlation, repeating the procedure 1000 times.

## Time-frequency analysis

We performed a time-frequency analysis to estimate the mean event-related power spectrum (1–250 Hz) across trials within the time window from −1 to 2 seconds relative to stimulus onset. Specifically, we applied a Morlet wavelet transform, using 2 cycles for the lowest frequency, with the subsequent frequency increasing linearly by 0.5 cycles/Hz. The estimated event-related power was then normalized to the pre-trial baseline (−0.5 to 0 seconds relative to stimulus onset; Fig. 6a).

## Inter-areal phase synchrony

We quantified the strength of inter-areal neural synchrony with the phase-locking value (PLV)[85], which measures the degree of consistency in phase between two channels, independent of their absolute phases and amplitudes, with values ranging from 0 to 1. The PLV for each channel pair was calculated as:

$$\text{PLV} = \frac{1}{N} \left| \sum_{n=1}^{N} \exp(i[\theta_{n,a} - \theta_{n,b}]) \right| \quad (1)$$

where we first computed the phase ($\theta$) differences between channels $a$ and $b$ and then averaged them across trials for a given frequency. We calculated the PLV across frequencies (2–250 Hz) for VTC-MTL channel pairs, with each pair containing one visually responsive channel in the VTC and one category-selective channel in the MTL. For each trial, we extracted the mean phase within 0 to 1.5 seconds after stimulus onset. For each session, we selected channel pairs that included the channel with the best axis-coding performance in the FG and ITG, and computed the time-frequency-resolved PLV using a sliding window of 100 ms, stepped by 25 ms.

We conducted a permutation test to determine the statistical significance of the PLV by estimating a null distribution through random shuffling of the trial labels for each channel pair and computing the corresponding PLV. This procedure was repeated 1000 times for each pair, and the observed PLV that exceeded the 95th percentile of the surrogate data was considered significant. We compared the observed mean PLV with the mean null distribution of all channel pairs for group statistics. For the time-resolved PLV, we applied an additional cluster-based criterion, retaining only clusters consisting of continuous significant pixels with a cluster size larger than 2.5% of the total number of pixels. The time-resolved PLV results were smoothed using a cubic spline interpolation method (spline.m function in MATLAB) for visualization purposes.

## Granger causality analysis

To test the directionality of the interaction between the VTC and MTL, we calculated spectral Granger causality, which quantifies the prediction error of the signal in the frequency domain by introducing another time series. The time series of each channel was first low-pass filtered at 85 Hz and downsampled to 250 Hz. We then normalized the signal within each trial to eliminate amplitude differences and improve signal stationarity. Before fitting the normalized signal to the multivariate autoregressive model, we determined the model order for each session using the Multivariate Granger Causality (MVGC) Toolbox[86], based on the Akaike information criterion (AIC). The Granger causality index was computed within 0 to 1.5 seconds after stimulus onset for both directions (VTC to MTL, MTL to VTC). We applied the same permutation procedure to determine statistical significance as in the PLV analysis.

## Spike–field pairwise phase consistency (PPC) between the MTL and VTC

We quantified the spike–field pairwise phase consistency (PPC) using established methods[87–89]. The PPC measures the average cosine similarity (i.e., in-phaseness) of any pair of spikes from the same neuron in the iEEG phase domain, reflecting the degree of synchronization between the firing of individual MTL neurons and the phase of oscillatory components in VTC channels. For every frequency $f$, we determined the iEEG phase at the time of each spike by computing the Fourier spectrum of the iEEG around the spike using the complete iEEG trace from 0.25 to 1.25 seconds after stimulus onset with a multi-taper method. Specifically, we used discrete prolate spheroidal sequences [DPSS] tapers, which are a set of Slepian tapers, with a taper number of 5 and a taper length corresponding to the number of sample points for 3 cycles per frequency (e.g., the sequence length and time-half-bandwidth product of tapers for frequency 20, 22, 24, 26 and 100 are: 151/3.02, 137/3.014, 125/3, 115/2.99, and 31/2.48, respectively). We focused on the gamma frequency band, as gamma phase-locking is suggested to reflect the activation of local networks for encoding specific identities[90]. The phase similarity was quantified using the PPC, calculating the average phase difference across all possible spike pairs within a time window from 0.25 to 1.25 seconds after stimulus onset. We calculated the PPC for each region-coding neuron in the MTL with all possible simultaneously recorded VTC channels. While the PPC corrects for sample-size bias, its estimates can be highly variable for units with a low spike count[87]. Therefore, we included only neurons with at least 20 spikes after stimulus onset to reduce variance in the group average[87].

For each neuron-channel pair, the PPC was estimated separately for in-region and out-region stimuli. To balance the spike count across conditions (as the number of in-region trials was less than out-region trials), we randomly subsampled the same number of spikes from out-region trials for each neuron–channel pair. The results remained consistent when we used bootstrapping and subsampled the out-region trials 100 times. To test statistical significance, we compared the mean PPC between the two conditions within our frequency range of interest (30–80 Hz) across all neuron–channel pairs using a paired $t$-test.

## Reporting summary

Further information on research design is available in the Nature Portfolio Reporting Summary linked to this article.

## Data availability

All data that support the findings of this study are publicly available on OSF (https://osf.io/x9u84/). Source data are provided with this paper.

## Code availability

The source code for this study is publicly available on OSF (https://osf.io/x9u84/).

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

## Acknowledgements

We thank all patients for their participation. This research was supported by the NIH (K99EY036650 [R.C.], R01MH129426 [S.W.], R01MH120194 [J.T.W.], R01EB026439 [P.B.], U24NS109103 [P.B.], U01NS108916 [P.B.], U01NS128612 [P.B.], R21NS128307 [P.B.], P41EB018783 [P.B.]), AFOSR (FA9550-21-1-0088 [S.W.]), NSF (BCS-1945230 [S.W.]), Brain & Behavior Research Foundation (33261 [R.C.]), and McDonnell Center for Systems Neuroscience ([R.C.]). The funders had no role in study design, data collection and analysis, decision to publish, or preparation of the manuscript.

## Author contributions

R.C. and S.W. designed the research. R.C., J.Zhang, Y.W., P.B., and S.W. performed experiments. J.T.W. performed surgery. R.C., J.Zhang, J.Zheng, and S.W. analyzed data. R.C., J.T.W., and S.W. wrote the paper. All authors discussed the results and contributed toward the manuscript.

## Competing interests

The authors declare no competing interests.
