## [Transparent Peer Review file · Nature Communications]

Computational single-neuron mechanisms of visual object coding in the human temporal lobe

Corresponding Author: Dr Shuo Wang

Version 0:

Reviewer comments:

Reviewer #1

(Remarks to the Author)

The iEEG data and single cell results from human temporal cortex reported by Cao et al are invaluable. In general such data are very much in demand as it becomes increasingly clear that fMRI data suffer from a very serious temporal resolution problem. More specifically, the authors report axis tuning in human FG, IT gyri and MTL extending results from non-human primates, as well as a very neat tuning of MTL single neurons in the feature space defined by the feature axes observed in the iEEG temporal leads. Both are very nice findings.

Yet the manuscript in its present form includes several severe flaws

1 the basic model outlined in fig 1 is not proven: the authors insist that area in feature space coding can only be detected with single cell recording, not iEEG. As there are no single cell recordings in FG nor IT gyri, we do not know whether such tuning does not already exist in FG or IT cortex. Thus it might be that the area in feature space is constructed in IT cortex and then simply transmitted to MTL (and perhaps other regions such as the temporal pole; TP); this is the more a problem that at iEEG level little difference between TE and MTL was reported, if anything leads were more responsive and selective in TE;

2 The authors make a constant confusion between semantic and episodic memory. The question addressed here is visual object identity (important for recognition), even if stimuli were organized in arbitrary categories, not the properties of concepts typical of semantic memory. There is considerable evidence that semantic processing occurs in TP (Lambon Ralph et al Nat NS Rev 2017, Tiesinga et al Scientific reports 2023), even if MTL neurons have been labelled concept cells to state that they are sensory invariant.

3 Little difference in results between AH and amygdala is reported, responses and selectivity at iEEG and single cell level is at least as strong in Amygdala as in PH or AH (fig 4 and 5). Yet these structures are supposed to have very different behavioral functions, a finding not discussed. In fact such results support the view that the single cell selectivity is broad cast from a common input, eg FG or IT cortex.

4 Statistical analysis. When comparing areas with respect to number of selective/responsive leads or latencies, it is likely that leads from different patients were included with numbers varying between areas. In this case mixed effects models with patients as random variable may be more appropriate than the chi square or unreported statistical tests: see Yu et al Neuron 2022 and Tiesinga et al 2023.

Other shortcomings are:

5 The authors gloss over intriguing difference between axis selectivity reported in the manuscript and that described originally in monkey TE by the Tsao group: 1) in monkey 4 axes have been reported, not 2 and 2) neurons tuned for these axes are localized in different parts of TE (here no clear segregation was reported); 3) axis tuning was measured with single cells (here iEEG, and results between these methods can differ); 4) in monkey the features for which axis tuning was observed, were shape parameters, extrapolated from earlier single cell studies, here features were derived deep learning and thus undefined although a coarse correspondence with the properties of animacy and man-made/natural objects, was stated.

6 methods used to derive links between areas are relatively unspecific and global, although effort were made to show they are linked to axis/ category selectivity of leads or preferred categories of neurons. Phase locking between leads not very specific (see ter Wal et al Neuroimage 2018); the much superior Granger connectivity was calculated on filtered data (<85Hz) and was restricted to lower frequency bands. Using the broad gamma band (50-150Hz) which reflect neuronal activity, Tiesinga et al 2023 obtained very specific links between particular leads in TE and TP. A more specific pattern of connection may be observed between FG and MTL with this method.

7 the choice of object categories is completely arbitrary in the main and control studies.

Other weaknesses:

8 in the result section on MTL single neurons there is too much reasoning on single cell coding (as in early H & W papers). Now we know that what matters is a population of tuned neurons which can retrieve the identity of object category;

9 Figures unclear in places: eg explain color codes eg fig 5e; panels with large numbers of photographs of categories are impossible to read

9 overstatements in text: eg in results section it is claimed that the iEEG data delineate responsive temporal cortex, but this requires that also non responsive areas are reported; in the discussion a topographic organization is mentioned, but it is unclear to which result this relates.

10 were any differences in latencies between subjects observed, as was reported by Tiesinga et al 2023?

Guy Orban

Reviewer #2

(Remarks to the Author)

Here, Cao take advantage of a unique data set capturing intracranial EEG (iEEG) from the ventral temporal cortex and single unit recordings from the medial temporal lobe as participants perform a simple one-back task in which they are viewing hundreds of images drawn from different categories. The authors build on prior work in NHPs demonstrating that activity in the VTC appears to be organized along a set of coding axes, where each axis represents the projection along a weighted combination of object features. They then compare this coding to the coding of individual neurons in the MTL, and find that the neuronal spiking responses in the MTL are organized within regions of this VTC coding space. This is a highly interesting and relevant study, as it begins to address the important question of how the human brain transforms information from densely distributed coding in the temporal lobe ventral cortex to sparse spiking activity in the medial temporal lobe. The analyses are well constructed, and the data are clearly presented. There are several suggestions, however, that might help strengthen the overall claims of the study.

One of the main claims of the manuscript is that feature coding occurs in the VTC whereas region coding within that feature space occurs in the MTL. In particular, they examine how individual neurons respond to the features defined by a PCA of the VTC feature axes. However, in one of the earlier results, the authors show that indeed MTL neuronal activity (specifically the hippocampus, Fig 3e) also encode the object feature space. As such, it would be helpful to understand how the region coding within the feature space differs from the direct feature coding observed in the anterior hippocampus. What is the benefit of performing this extra manipulation.

When examining region coding, the authors use the feature space defined only by the fusiform gyrus. Yet there are other regions of the VTC that also capture this feature space, as shown by their data. Why restrict the region coding to only the FG feature space? Moreover, in subsequent analyses examining phase locking and Granger causality, rather than just focus on the FG, they then expand their analyses to examine the entire VTC. Why would the difference between the two analyses.

The authors present an analysis in which they compare feature based coding in the VTC to category based coding in the MTL using representational similarity analysis. It would be helpful if they could clarify exactly how this analysis was conducted and what was compared. Reading through the methods did not clarify this point, but this seems like it would be a nice segue into subsequent analyses on region coding.

The authors demonstrate significant phase locking in lower frequency bands between the VTC and MTL. But the analysis if Granger causality demonstrates connectivity at much higher frequencies. Could the authors provide some insight as to why the discrepancy? Moreover, when examining the locking of individual neurons to gamma phase, the authors find a subset that indeed exhibit significant phase locking. Here the reported numbers, it appears, are $n=9$, and $n=14$. Are these meaningful?

On a related note, it would probably be helpful if the authors were a little cautious in the interpretation of the connectivity results. Specifically, they claim that their analyses demonstrate that VTC coding channels provide more relevant information to the hippocampus during coding. Without an explicit measure of information and a clear analyses showing how coding is tied to this connectivity, the authors may want to consider tempering this claim.

Reviewer #3

(Remarks to the Author)

Summary of the Study

This study investigates the neural mechanisms underlying object representation in the human ventral temporal cortex (VTC) and medial temporal lobe (MTL) using intracranial EEG (iEEG) and single-unit recordings. The authors propose a computational framework in which the VTC encodes visual features via axis-based feature coding, which is then transformed into region-based feature coding in the MTL, supporting semantic representation. Through functional connectivity analyses, they demonstrate bidirectional interactions between these regions and provide evidence that axis-coding in the VTC is linked to semantic coding in the MTL. This work builds on prior findings in non-human primates and extends them to human recordings, offering novel insights into the transition from perceptual to semantic representations in the brain. However, several aspects of the analysis require further clarification, particularly regarding alternative coding schemes, the interpretation of functional connectivity, and the temporal dynamics of processing. Additionally, refining the presentation of figures and ensuring consistency in methodological justifications would further strengthen the paper.

Major Strengths

- The study addresses an important question in visual neuroscience: how object representations transition from perceptual encoding in the VTC to conceptual encoding in the MTL.
- The combination of iEEG and single-unit recordings provides a unique and powerful dataset, allowing for a detailed analysis of neural coding at multiple scales.
- The use of partial least squares (PLS) regression with deep neural network (DNN) feature representations is methodologically rigorous and aligns with recent advances in computational neuroscience.
- Functional connectivity analyses using phase-locking value (PLV) and Granger causality (GC) strengthen the argument for dynamic interactions between the VTC and MTL.
- The validation with an additional dataset (COCO) enhances the generalizability of the findings.

Major Points for Revision

1. Evidence for axis coding in the VTC

- The significance of PLS regression alone is insufficient to establish axis coding as the dominant encoding strategy in the VTC. While the regression's significance partially controls for exemplar-based coding, the relatively low R^2 values suggest that axis coding may not be the most explanatory model.
- The authors should directly compare axis coding with other encoding strategies, particularly category-based encoding, which is mentioned at the beginning of the manuscript. It is essential to demonstrate that axis coding explains the neural responses in the VTC better than alternative models.
- One approach would be to fit a regression model that includes both axis coding and category-based coding variables and compare their contributions. Alternatively, a correlation-based analysis or Representational Similarity Analysis (RSA) could be used to construct dissimilarity matrices based on both axis coding and category-based coding. The authors should examine whether the dissimilarity matrix derived from axis coding better aligns with the neural responses in the VTC.
- Additionally, the demonstration of cross-dataset generalizability and consistency of the axes is a major strength of the study. However, the investigation of object space remains insufficient. Identifying the nature of the coding axes is crucial to further validate the axis-based feature coding in the VTC. Specifically, more detailed characterization of the extracted axes (e.g., how they align with known dimensions such as animate vs. inanimate) would strengthen the argument for axis-based encoding.

2. Region-based feature coding in the MTL relative to alternative coding schemes

- The finding that MTL receptive fields exhibit region-based feature coding within the VTC feature space is highly interesting. However, it is necessary to evaluate whether this coding scheme provides a better explanation of the data compared to alternative models.
- Specifically, the authors should compare region-based coding with category-based coding and localized axis coding to determine whether region-based feature coding is the most informative representation.
- A direct comparison could involve fitting models that incorporate different coding schemes and assessing their relative explanatory power. Alternatively, the authors could perform RSA using dissimilarity matrices based on these different coding schemes to test which model best aligns with the neural responses in the MTL.
- Clarifying this distinction would strengthen the argument that region-based coding is the preferred computational strategy for transforming VTC feature space into MTL representations.

3. Relationship between face-selective channels and axis coding in the VTC

- Previous ECoG studies (e.g., McCarthy et al. *Cereb Cortex* (1999), Matsuo et al. *Cereb Cortex* (2015)) have demonstrated that face-selective channels are organized in patches within the VTC. Given this prior work, it is crucial to examine the relationship between face-selective channels and axis-coding channels in the current study.
- The authors should explicitly test whether face-selective channels exhibit axis coding or whether they represent a distinct form of category-selective coding. This is particularly relevant for the fusiform face area (FFA), where category selectivity has been well-documented.
- A key question is whether face selectivity in the human FFA, as defined in this study, corresponds to axis coding as seen in non-human primate (NHP) face patches.
- If face-selective channels do align with axis-coding channels, this should be explicitly demonstrated with supporting analyses. Conversely, if they do not align, this should be discussed in terms of differences between category-selective and axis-based feature coding.
- A possible approach would be to compare the locations of face-selective channels with axis-coding channels and test whether the neural responses in face-selective regions are better explained by axis coding or category-based models.

4. Relationship between regional coding and semantic coding

- The relationship between regional coding and semantic coding is not entirely clear. While the study suggests that regional coding in the MTL represents a transition from perceptual to semantic representations, regional coding could, in principle, occur at lower levels of the visual hierarchy as well. For example, in early visual processing, simple cells combine to form complex cells, which also exhibit localized feature selectivity.
- The definition of regional coding in this study is based on a high-dimensional feature space derived from DNNs, which emphasizes complex feature representations. However, this does not necessarily imply a direct link to semantic representations.
- The authors should clarify how regional coding in the MTL differs from lower-level feature encoding and why it should be considered a semantic transformation rather than an extension of perceptual processing.
- One possible approach would be to compare regional coding in the MTL with a measure of semantic similarity (e.g., word embeddings or conceptual clustering) to determine whether it aligns better with abstract category representations rather than purely visual feature spaces.

• Additionally, the theoretical framework for how regional coding bridges perceptual and semantic representations should be more explicitly described to strengthen its validity.

Minor Points

1. Fig. 1: If single-neuron recordings were not performed in the VTC, the plotted circles should not be depicted in a way that suggests they represent firing rates. The figure should be adjusted to accurately reflect the type of data collected (e.g., high-gamma power from iEEG rather than spike rates).
2. Fig. 2b: The color scheme for the fusiform gyrus (FG) and inferior temporal gyrus (IT) appears to be reversed.
3. Fig. 2e, f: The latency calculation may be influenced by response amplitude, as larger responses often lead to shorter latencies. This pattern appears to be present in the current data. If hierarchical processing is a key discussion point, it would be beneficial to verify whether the observed latency differences are genuinely reflective of processing order rather than differences in response magnitude. One way to address this is to normalize response amplitudes across regions (e.g., through subsampling or amplitude-matched comparisons) and then re-evaluate the latency differences. Alternatively, latency can be measured using different thresholds, such as relative change rate or peak time, to verify whether the results remain consistent.
4. Fig. 3d: A thin boundary line or shading, along with a corresponding label in the figure legend, would make the figure more informative.
5. Fig. 3f: The delayed onset of axis coding in the anterior hippocampus (AH) is an interesting finding. However, it raises the question of whether this signal reflects true axis coding or whether it is confounded by region coding. Since region coding inherently involves encoding localized receptive fields in the VTC feature space, it may manifest with similar temporal characteristics as late-onset axis coding. The authors should discuss whether the observed AH activity aligns more closely with the characteristics of axis coding or region coding.
6. Fig. 3g: The results in Fig. 3(g) could be influenced by response amplitude, potentially making it harder to distinguish whether the observed effects are due to differences in coding strength or simply variations in signal magnitude. To improve clarity, the authors could normalize response amplitudes across regions or examine whether the patterns remain consistent when analyzing only channels with similar response magnitudes.
7. Fig. 3h: If possible, the original representational dissimilarity matrices used to compute the correlation values in Fig. 3h should be shown, either in the main figure or as a supplementary figure. This would help readers better understand the structure of the neural representations being compared and how axis coding in the VTC relates to category coding in the MTL. Additionally, even in statistically significant comparisons, the absolute correlation values appear to be quite low. The authors should discuss what this implies about the strength of the relationship between these representational spaces. Possible interpretations include: Neural representations in the VTC and MTL are related but remain largely distinct. The transformation from axis coding to category coding introduces substantial variability, resulting in lower similarity. Other factors, such as noise in the data or methodological limitations, may contribute to the low correlation. If the authors can provide additional insights into why these correlation values are low despite statistical significance, it would improve the clarity of the findings.
8. Fig. 4a: The current visualization is difficult to interpret due to overlapping stimuli. To improve readability, the authors should adjust the presentation to reduce visual clutter. One effective approach would be to highlight examples of stimuli located near category boundaries. This would help illustrate that while the visual features are continuous, a clear semantic boundary exists in the representational space. It is important to ensure a clear and representative example of such a case.
9. Fig. 4b right lower: In permuted data, systematic patterns should typically disappear. However, a response peak is still observed here. This suggests that some structural bias (e.g., imbalance in sample size, spatial non-uniformity) may be influencing the results. Do you have any insights or considerations regarding this observation?
10. Fig. 4e and f: The interpretation of these results is unclear. A direct comparison with VTC data would likely provide valuable context, but given that single-unit data from the VTC is not available, this may not be possible. The authors should clarify how these results support their conclusions about region-based coding in the MTL. If the authors acknowledge that the lack of direct VTC comparison limits the interpretability of these results, they should discuss this limitation in the discussion section. Additionally, explaining why these results are important for understanding the broader hypothesis of the study (e.g., how they support the concept of region-based coding in the MTL) would help readers grasp the significance of these findings.
11. Fig. 4 and related text: The statement, "fewer region-coding channels were detected than expected by chance, indicating that region coding did not occur at the mesoscopic level", would be more informative if the exact percentage of detected region-coding channels was provided. This result raises an important methodological concern: does the absence of region coding in iEEG recordings simply reflect differences in recording resolution rather than a true absence of region coding at the mesoscopic level? Given that single-unit recordings in the MTL detected region coding, the authors should discuss whether this discrepancy arises from the nature of iEEG signals, which pool activity over larger populations of neurons, potentially masking fine-scale region-based tuning. If region coding does not manifest at the iEEG level in the MTL, this raises the question of whether iEEG recordings in the VTC might also fail to detect certain types of coding. Could this methodological limitation affect the comparisons between VTC and MTL? A more detailed discussion may be necessary to

clarify these points. Providing additional context or explanations could help improve the interpretability of the results and address potential concerns.

12. Fig. 5g: The results should be presented using the same scale as Fig. 4g to ensure consistency in data visualization. Aligning the scales (ranges) would make it easier for readers to directly compare the validation dataset results (Fig. 5g) with the original dataset results (Fig. 4g).

13. Fig. 6a: The PLV results in Fig. 6a provide important insights into the functional connectivity between the VTC and different MTL subregions. Given these findings, it is essential to carefully examine how they correspond to the representational similarity results shown in Fig. 3h. The fact that FG–AH shows coupling in the high-frequency range is particularly intriguing. This pattern suggests a unique role for AH in processing visual information received from the FG. The authors should explicitly analyze whether the strength of PLV in specific frequency bands correlates with the representational similarity results from Fig. 3h. If stronger functional connectivity (PLV) aligns with greater representational similarity, this would provide additional support for the proposed computational framework. If stronger functional connectivity (PLV) aligns with greater representational similarity, this would provide additional support for the proposed computational framework.

14. Fig. 6b–c: To better understand the role of different MTL subregions, it would be beneficial to analyze Fig. 6(b–c) by separating the results for each subregion (e.g., anterior hippocampus (AH), posterior hippocampus (PH), and amygdala). If the sample sizes are sufficient for certain subregions, those should at least be analyzed separately to provide a more fine-grained view of the connectivity differences. This is particularly important in relation to Fig. 3h, where different subregions may have shown different representational similarity with the VTC. If the PLV results (Fig. 6) and representational similarity results (Fig. 3h) align in a region-specific manner, this would provide additional support for the proposed coding transition between the VTC and MTL.

15. The finding that FG and IT exhibit stronger feedforward GC to AH in the <10Hz range is particularly interesting, as it suggests a low-frequency mechanism for information flow from VTC to the MTL. However, this effect does not appear as prominently in Fig. 6f. This discrepancy could be due to differences in the way the analysis is conducted (e.g., spectral GC in Fig. 6d vs. time-resolved GC in Fig. 6f), or subregion-specific variations within the MTL—perhaps only AH exhibits this effect, while PH or the amygdala do not, or the effect being too weak when examined in the time domain rather than the frequency domain. To clarify this, the authors should analyze GC separately for each MTL subregion if sample sizes permit. If AH uniquely exhibits this low-frequency feedforward GC, this should be explicitly discussed, as it would suggest a distinct role for AH in integrating visual information from the VTC. Additionally, the authors should consider how these connectivity results relate to Fig. 3h, which examined representational similarity. If stronger connectivity at <10Hz corresponds to higher representational similarity, this would provide additional support for the proposed functional link between the regions.

16. Fig. 7a, b: The authors focus on gamma oscillations (30–80 Hz) for spike-field coupling analysis, which is well-justified in the context of visual processing. However, lower frequency oscillations, particularly in the theta range (3–8 Hz), are also critically involved in memory and semantic processing. Given that Fig. 6 identified strong low-frequency coupling (especially in the <10 Hz range), it is important to investigate whether spike-field phase locking occurs at these frequencies as well. In the lower right plot of Fig. 7b, there appears to be some phase locking in low-frequency bands. This suggests that meaningful coupling may also be present outside the gamma range. The authors should expand their analysis to include theta-band phase coupling and compare its strength to gamma coupling. If strong theta-band locking is also observed, this could suggest a multi-frequency communication mechanism between the VTC and MTL, with gamma frequencies supporting feature encoding and theta frequencies supporting integration into memory/semantic networks. If theta coupling is weak or absent, this would reinforce the idea that high-frequency oscillations dominate in this particular task context. In either case, a brief discussion on why gamma was chosen and how the findings align (or contrast) with known roles of theta oscillations in MTL processing would be beneficial.

17. Fig. 7(d): The observed spike-field coupling appears primarily after 400ms post-stimulus, which is quite late for a purely visual response. This raises an important question: how does this timing align with the broader interpretation of axis coding and VTC-MTL interactions? The main analysis window (0.1–0.6s) used elsewhere in the study does not fully capture this late effect, suggesting a potential misalignment between different analyses. One possible explanation is that the coupling observed in Fig. 7d reflects a later stage of processing related to higher-level integration (e.g., memory retrieval or semantic association) rather than initial visual encoding. Interestingly, the timing of this effect aligns well with the late-onset axis coding observed in the AH in Fig. 3f. This suggests that the coupling in Fig. 7d could be closely related to the delayed axis coding observed in the AH. The authors should explicitly discuss whether these findings indicate a multi-stage processing hierarchy, where early responses (0.1–0.4s) are dominated by direct visual feature encoding and later responses (~400ms onward) reflect semantic or associative processing. If possible, an analysis directly comparing the time course of axis coding strength in the AH (Fig. 3f) with the timing of phase locking in Fig. 7d would help clarify this relationship. If this delayed timing is a key feature of MTL processing, it would be useful to examine whether similar late-onset effects appear in other aspects of the data, such as representational similarity or functional connectivity. Ultimately, a clearer discussion of why the observed coupling occurs so late and how it fits into the broader computational framework would improve the interpretation of these results.

18. Fig. 7 additional: if possible, the authors should separate the VTC into IT and FG when analyzing phase coupling, as this would allow for better alignment with the findings from Fig. 3(h) on representational similarity. Given the functional differences between FG and IT, examining them separately might reveal distinct connectivity patterns with the MTL. If IT and FG exhibit different coupling characteristics with the MTL, this would provide additional support for a hierarchical processing structure within the VTC-MTL pathway. Visualizing the stimuli that drive these responses could further strengthen the

paper's conclusions. Since this study includes both iEEG and single-neuron recordings, this is highly valuable data, showcasing example stimuli that selectively engage region-coding neurons in the MTL and their corresponding VTC channels would enhance clarity. If the authors can illustrate how specific visual features or object categories are represented in both regions, this would make the interpretation of functional connectivity results more intuitive. In any case, since this is highly valuable data, illustrating the visual responses of the circuits identified from the responses could help better convey the significance of the statement, "Lastly, we investigated whether region coding in the MTL is related to visual processing in the VTC. In 9 patients (13 sessions), we recorded iEEG in the VTC and single neurons in the MTL simultaneously, which allowed us to directly test the functional connectivity between MTL region-coding neurons and VTC channels."

Reviewer #4

(Remarks to the Author)

In the present study, Cao et al. conducted intracranial EEG (iEEG) recordings together with single-neuronal recordings in the ventral temporal cortex (VTC) and the medial temporal lobe (MTL) in epileptic patients observing large number of visual objects. The authors analyzed neuronal responses to the presentation of visual objects using DNN-based visual feature extraction, as well as principal component analysis, and found that the VTC exhibits axis-based feature encoding, and that the MTL neurons encode receptive fields within the VTC neural feature space. They also conducted functional connectivity analysis with Granger causality and spike-LFP coherence analysis revealing bidirectional interaction between the VTC and the MTL.

The authors have accomplished a lot of work, the experiments and analysis have been performed properly, and the results are reliable and important for understanding the neuronal computational mechanisms underlying object recognition in humans. However, more comprehensive description and clarification of conceptual questions are needed before the paper can be accepted for publication.

Introduction does not clearly explain the computational framework of the study, which prevents it from conveying the importance of the study. The second paragraph states the key questions and hypotheses of the study. However, they are very difficult to understand, especially for the general reader outside the field of cognitive neuroscience. In particular, it will be difficult to understand the meaning and relationships among terms such as axis-based coding, neural axis, region-based coding, sparse coding, dense coding, receptive field, and neural feature space. The third paragraph and Fig. 1 are also difficult to understand and confusing. It is difficult to understand how the neural feature space is constructed from axis coding in the VTC. Also, it is difficult to understand the relationships between the feature dimension (Fig. 1, left panel) and the neural axes (Fig. 1, right panel). This is because Introduction and Fig. 1 omitted that principal component analysis was used to construct the neuronal feature space from the VTC neural axes. This process is not explained until the Results and Methods section, which can lead to misunderstanding and confusion for the readers. A sentence in Fig. 1 legend, "MTL neurons receive processed visual input from the VTC and encode a receptive field (i.e., encoding region) within the neural feature space...". Readers may have difficulty understanding what is a receptive field within the neural feature space because the term receptive field usually means a receptive field in the visual field. Overall, the authors should expand and rewrite Introduction, as well as improve Fig. 1 and its legends.

In Abstract, the authors stated: "The VTC exhibited axis-based feature coding, and by constructing a neural feature space using VTC neural axes, we observed that MTL neurons encode a receptive field within the VTC neural feature space. This computational framework explains how dense feature-based representations in the VTC are translated into sparse semantic-based representations in the MTL." These are the central finding and implication of the study. However, since the neural feature space is composed of features based on the 'visual' axis, the finding that MTL neurons encode receptive fields in the neural feature space alone is unlikely to explain how representations are converted to representations based on 'semantics'. The authors should clarify the logic supporting the claim that this computational framework explains how feature-based representations in the VTC are translated into semantic-based representations in the MTL.

Results

In Figure 3c, the PH did not show an above-chance rate of axial feature coding, while the AH showed an above-chance rate of axial feature coding bilaterally. This may suggest the presence of a posterior-anterior gradient of axis-based feature coding in the hippocampus. Do the authors have any thoughts on this finding?

Version 1:

Reviewer comments:

Reviewer #1

(Remarks to the Author)

The authors made extensive revisions to the manuscript to accommodate the comments of all 4 reviewers.

With respect to my comments (reviewer 1), the changes made to address my points 3 to 6, 8 to 10 are fine, those related to points 1 and 3 go in the right direction but could be further improved. Sadly, one of my main comments, my point 2, related to the claim that a semantic-based coding emerges in MTL, was made worse by the changes made to address the comments of reviewer 4.

Point 1; the authors provide some evidence, but from another species, which is only a suggestion in the present context, as the authors admit in the rebuttal. Hence, this limitation should be clearly indicated in the discussion.

Point 7: there still is a confusion between animate/inanimate and natural/artificial as the following sentence taken from the rebuttal clearly shows: 'The selection ensures the presence of both living and non-living entities, covering a range of animate (e.g., bear, elephant, person) and inanimate (e.g., airplane, chair, car) objects.' In their reply to reviewer 3 the authors made a better job of separating these two dimensions, and should make sure this done throughout the manuscript. A personal note for the authors concerning this choice of stimuli. In my own vision works (especially in the videos made of natural actions), I made sure only natural objects and surroundings were shown. My argument was that artificial objects were around only for some 100 years and the brain could not have adapted to process these novel objects. The present data show that this not true and that the fast learning hippocampal system is able to represent these objects. Maybe this underscores a remarkable strength of the human brain.

This leaves point 2, where the authors made no change or even emphasized even more (in their reply to reviewer 4) that hippocampus builds a semantic based representation from the input provided by VTC. This flies in direct contradiction to the extremely well established neuropsychological data indicating a clear dissociation between the temporal pole, devoted to semantic memory, and hippocampus, subserving episodic memory of facts and personal experiences (see eg Quiroga et al Nat NS review 2015).

There is nothing in the data provided in the manuscript to indicate that the novel representation which emerges in HP is semantic-based; the main argument of the authors is that the representation is not any longer perceptual. Yet, semantic is not the opposite of perceptual; what the authors have in mind is that the representation in the HP has a meaning, i. e. refers to something in the outside world. A less controversial choice to replace semantic-based could be referential, integrative or even better conceptual (referring to the concept cells described in HP by the Fried group); the latter choice would be optimal as the neurons reported here in HP bear striking similarities to the concept cells of Quiroga et al, and are in all likelihood the same population.

In fact, the data provided in the manuscript show that the representation in HP is NOT semantic in nature: indeed in reply the reviewer 3 they show that the semantic matrix of their stimuli accounts for only 8% of the variance in neural activity of HP. While this more than the perceptual matrix (the intention of the authors), it is also clearly contradicting the claim that the representation is semantic-based.

Furthermore, the authors do not report their main data (fig 3 in old manuscript) accurately: they claim it shows the HP neurons associate different features; this is incorrect: what the figure shows is that the HP neurons associate particular strengths of features provided by the axis selectivity of VTC neurons. Such association of strengths is typical of a single object/person, not a semantic category, as they authors themselves acknowledge in one of their replies to reviewer 3. In this reply they insist that the HP representation is not that of categories. Hence it is difficult to understand how the authors can claim that the HP representation is semantic-based.

In their rebuttal the authors claim: 'We observed category-selective MTL cells, which have long been regarded as fundamental building blocks of semantic memory 2-5—one component of declarative memory, alongside episodic memory.' I do not understand this claim, as nothing in the present manuscript support this claim, to the contrary (see above)

They also point to an 'extensive literature has also demonstrated semantic processing in MTL'

The oldest of these references (Kreiman) refers to the old literature describing face and body patches in IT/VTC as category selective. We now know these are simply regional specializations of the feature space in IT/VTC. In a paper recently published in Current Biology, the same group of authors showed that the analysis presented here for objects also is valid for faces. Hence this paper does not support the authors claim.

Reber et al 2015 claims to have evidence for semantic category processing in HP. These authors analyzed the population vector summing all HP neurons and show that it carries some information about the superordinate category of their stimuli. This however was in fact an analysis of the noise in the firing rate (the analysis also held when restricting it to the unresponsive neurons), which has no use in coding object identity (thus no use for episodic memory). The present results suggest this might simply reflect implicit category information present in the feature space of the input to HP. Indeed the fig 4a of the present manuscript (revised) has some striking similarity with fig 3F in Reber et al. Thus one might speculate that the feature space represented by VTC input might be analyzed in parallel in two different ways in MTL and TP. This reference fails to show that neuronal responses to visual stimuli carry category information, and thus does not support the claim of the authors.

Finally the papers from the Fried group (mainly with Quiroga as first author) described the concept cells which match very closely the properties of the HP neurons described in the present manuscript. While clearly describing how the properties of these neurons make them suitable for recording personal experiences and episodic memory, this group sometimes refers to them as semantic processing (see eg the Nat NS Rev 2015 Quiroga et al). In the use of this word they point to the fact that the HP neurons represent something in the physical world- very like the real world entities I coined, Orban et al Front. Psychol 2014- and thus have a meaning. Used this way 'semantic processing' does not imply any relation to semantic memory or semantic categories.

Thus if the authors persist in using the word 'semantic-based' they should at least clarify what they mean by this word. However, given that it is now placed at the center of the manuscript (unlike in the Quiroga papers), it seems to be much wiser to avoid using this word and replace it by conceptual processing. As mentioned above, this refers to the concept cells described by the Fried group, which are well established, and matches the meaning of concept: 'a generic mental image abstracted from percepts', according to the Webster.

Reviewer #2

(Remarks to the Author)

The authors have provided a revision to their manuscript in response to the first round of reviews. In general, it appears that

the authors have sufficiently addressed many of the criticisms raised in the initial reviews, and have tempered some of their claims regarding the interpretation of the results. This is a valuable manuscript presenting a rare dataset, and will have of high interest for researchers interested in understanding visual object recognition and coding in the human brain.

Reviewer #3

(Remarks to the Author)

General Comment

I appreciate the authors' thoughtful and detailed responses to my previous comments, as well as the substantial revisions made to the manuscript. I particularly value the additional analysis regarding shared computational mechanisms for face and object coding in the VTC, as well as the enhanced mechanistic analysis provided in the latter half of the paper. However, there may still be room for further examination regarding the validation of the computational framework presented in the first part.

Overall, the study addresses an important and compelling question in systems neuroscience. The proposed computational framework, which links axis-based coding in the VTC with region-based semantic representation in the MTL during object recognition, is highly valuable and thought-provoking.

In the following, I will provide comments on each of the authors' previous responses. In doing so, I will also attempt to suggest possible ways to more rigorously evaluate the proposed computational framework. Italicized segments indicate the authors' responses, whereas standard (non-italicized) text contains my own remarks.

Major Points for Revision

1. Evidence for axis coding in the VTC

We thank the reviewer for the question and suggestions. First, we apologize for an error in estimating the R^2 for the partial least squares (PLS) model, which has now been corrected in Supplementary Fig. 2a. Please note that although the trend remains consistent, the mean R^2 is now 0.6, indicating a good fit of the model.

The correction of the R^2 values from 0.1 to 0.6 is indeed substantial, and I appreciate the authors' transparency in addressing this error. However, such a large change raises concerns about the original implementation of the model and how the correction was made. It would be helpful if the authors could briefly explain the nature of the error (e.g., whether it was a coding mistake, a misinterpretation of the R^2 definition, or some other issue) and clarify how the corrected values were validated. Providing this information would enhance confidence in the revised analysis and its implications for the validity of the axis-coding model.

Additionally, the R^2 values reported for the pool5 layer appear to be highly similar across regions (Supplementary Fig. 2a), which seems to contradict the patterns shown in panel b of the same figure and the associated main text description suggesting regional differences. It would be beneficial for the authors to clarify whether this apparent discrepancy is due to a difference in what is being measured (e.g., number of significant channels vs. model fit quality) or whether the similarity in R^2 curves across regions has a specific interpretational significance.

Second, to determine whether axis coding or category-based coding better explains neural responses in the VTC, we compared the R^2 from the PLS model with that derived from an ANOVA applied to neural responses using categorical labels for the category-based coding model. Indeed, we found that across all visually responsive channels, the axis model explained significantly higher variance compared to the category model in both the FG (axis: 0.63 ± 0.038 [mean \pm SD]; category: 0.17 ± 0.087 ; two-tailed paired t-test: $t(200) = 91.94$, $P < 10^{-20}$) and ITG (axis: 0.62 ± 0.030 ; category: 0.12 ± 0.046 ; $t(73) = 63.34$, $P < 10^{-20}$; Supplementary Fig. 2d). We have included the following results in the revised manuscript:

The direct comparison between the axis and category models using variance explained clearly supports the superiority of axis-based coding in both FG and ITG regions. This is a strong and well-motivated addition. While the statistical differences are robust and the results are clearly presented, I have a few concerns regarding the interpretation of this comparison that merit clarification.

The PLS model for axis-based coding is based on continuous DNN features that inherently contain high-dimensional and fine-grained visual information, whereas the ANOVA model relies on discrete categorical labels. Given this asymmetry in the richness of input features, the superior performance of the PLS model may be expected. Could the authors comment on how they controlled for the differences in input dimensionality and information content between the two models?

Relatedly, further clarification is warranted regarding the construction of the category-based model. The ANOVA model appears to rely on ImageNet-derived category labels comprising approximately 50 relatively fine-grained object classes (e.g., "young mammal," "rodent," "frog," "reptile"), many of which are closely related. Such fine-grained labeling, combined with potentially limited trial numbers per class, raises the possibility that the model's R^2 values may be underestimated. It would be helpful if the authors could clarify the rationale behind this categorical structure and, additionally, evaluate whether alternative grouping schemes—such as clustering based on superordinate semantic categories—would affect the comparison with the axis-based model. Such analyses would strengthen the validity and generalizability of the conclusions. Previous studies have shown that higher layers of CNNs represent semantic category-level information (e.g., Bau et al., 2017; Khaligh-Razavi & Kriegeskorte, 2014). Given this, and considering that the activity of individual VTC channels is well explained by deep CNN features, the relatively low R^2 values obtained from the ANOVA-based category model are somewhat unexpected. It would therefore be valuable to discuss possible reasons for this discrepancy.

Overall, the comparison between the PLS and ANOVA models is a valuable addition. However, I encourage the authors to justify more carefully the conclusion that axis-based coding outperforms category-based coding, from a principled and biologically grounded perspective—particularly given that the way category coding is modeled is a critical factor in this comparison. In this regard, the authors might also consider incorporating alternative models that take into account lexical or semantic distances—such as those derived from word embeddings—as they have explored in later sections of the manuscript. This could provide a more graded and neurobiologically plausible account of semantic representation in the VTC and MTL.

It is worth noting that we could not fit a single regression model that includes both axis coding and category-based coding variables to directly compare their contributions. Additionally, we would like to clarify that while representational similarity

analysis (RSA) can be used to construct image-by-image dissimilarity matrices (DMs) for visual features (i.e., by correlating visual feature vectors between each pair of images) and for neural responses (i.e., by correlating neural vectors between each pair of images), it cannot be used to construct DMs based on coding schemes. Therefore, we could not use this approach to test whether the DM derived from axis coding better aligns with neural responses in the VTC. In other words, RSA can be used to examine the similarity between coding schemes or between brain areas (e.g., Fig. 3h), but not to compare how well different coding schemes explain neural responses.

The authors correctly point out the methodological challenges in directly comparing axis-based and category-based coding models within a unified regression framework, as well as the limitations of RSA in assessing model fit. However, as discussed in the preceding point, given the differing nature of the models (PLS vs. ANOVA), it remains unclear whether the higher R^2 values observed for the PLS model truly reflect superior explanatory power, or whether they may instead result from differences in model complexity or representational mismatch. While it is true that RSA is not designed to compare explanatory power directly, it could still serve as a complementary approach. Specifically, comparing dissimilarity matrices (DMs) derived from axis-based, category-based, and neural representations may offer useful insights into the nature of VTC representations (Kriegeskorte et al., *Front. Syst. Neurosci.* (2008). Khaligh-Razavi & Kriegeskorte, *PLoS Comput. Biol.* (2014), Jozwik et al., *Journal of Neurosci.* (2016)). In fact, the authors have already performed a similar analysis in Revision Fig. 6, which appears to yield valuable findings.

Third, we thank the reviewer for acknowledging the cross-dataset generalizability and consistency of the axes as a major strength of our work. We also appreciate the reviewer's suggestion. In response, we have characterized the extracted feature axes in greater detail and compared them with known object feature dimensions. The following results have been included in the revised manuscript:

Thank you for the additional analyses and the detailed revise. The new results characterizing the neural axes along the natural–artificial, animacy, and spiky–stubby dimensions provide useful insight and clearly demonstrate that the extracted axes align with previously reported visual feature dimensions. The cross-dataset replication using the Microsoft COCO stimuli further supports the robustness of these dimensions.

2. Region-based feature coding in the MTL relative to alternative coding schemes

We thank the reviewer for raising this important question. Indeed, in our recent study 13, we have demonstrated the absence of axis coding in MTL neurons using the same set of object stimuli. Notably, in that work, we also showed that region-based coding in the MTL provides a more comprehensive mechanism for explaining category selectivity. Specifically, the region code does not rely on categorical membership of individual images, as long as the images share similar visual features. While images from the same category often cluster together in feature space, which can be captured by category-selective coding, images from different categories may also cluster due to shared visual features (e.g., structure, texture), which category coding alone cannot account for. The broader, feature-based representation observed in MTL neurons suggests that region-based coding serves as a more general and robust model for transforming the VTC feature space into MTL representations 13.

I understand that in their recent work, the authors report the absence of axis coding in MTL neurons and instead propose a region-based coding scheme in the MTL. This form of coding can be viewed as intermediate between axis-based and label-based category coding. In terms of semantic abstraction, one might conceptualize a hierarchy from axis coding (low-level), to region-based coding (intermediate), to label-based category coding (high-level), with corresponding models being the PLS model, semantic embedding models, and the ANOVA model, respectively. While this may be a somewhat simplified interpretation or may not fully capture the authors' framework, it nonetheless offers a useful way to contextualize the different coding schemes and their associated analyses.

Given this, it would be informative to evaluate neural responses in each region using all three models. Although direct comparisons between regions may be challenging due to differences in measurement modalities (e.g., iEEG vs. single-unit recordings), I believe that within-region comparisons of explanatory power—along with representational similarity analyses (RSA) comparing dissimilarity matrices predicted by each model—are both meaningful and feasible.

3. Relationship between face-selective channels and axis coding in the VTC

We thank the reviewer for the suggestions. First, we would like to clarify that the current study focused on general object coding rather than face coding. We agree that it is an important question to investigate whether face selectivity in the human FFA corresponds to axis coding, as observed in non-human primate (NHP) face patches. However, this question falls slightly outside the scope of the current study, as we did not specifically include face images. We would also like to clarify that our results did not rely on face/object selectivity, and we did not argue for face/object-selective processing or domain specificity in face/object processing. Rather, the VTC, particularly the fusiform gyrus (FG), may exhibit similar computational principles for both faces (cf. 25) and non-face objects (our present study), consistent with previous macaque studies demonstrating common computational principles of axis coding for both faces¹⁵ and objects¹⁴.

Thank you for the clarification. I appreciate that the focus of the present study is on general object coding rather than face-selective processing. Nevertheless, I sincerely appreciate the additional analysis you performed, which I found highly interesting and informative.

Second, as the reviewer suggested, we conducted additional control analyses using faces. We recorded neural responses to both faces and objects from a subset of 11 patients (14 sessions), allowing us to further examine whether axis coding generalizes across object and face stimuli. We found that face-selective channels (10/57, 17.54%) were not significantly more likely to be axis-coding channels compared to the overall population (40/240, 16.67%) in the VTC (χ^2 -test: $P = 0.87$), suggesting that axis coding is not specific to face-selective channels. Additionally, we found that 29 out of 72 VTC channels (40.28%) that exhibited axis coding for object stimuli also exhibited axis coding for face stimuli. These results suggest that axis coding functions as a general coding principle for visual stimuli in the VTC, encompassing both faces and objects. Similarly, when examining the FG specifically, we found that face-selective channels (8/26, 30.77%) were not significantly more likely to be axis-coding channels compared to the overall population (36/135, 26.67%; χ^2 -test: $P = 0.67$). Moreover, 26

out of 57 FG channels (45.61%) that exhibited axis coding for object stimuli also exhibited axis coding for face stimuli.

Thank you for conducting the additional analyses using face stimuli. The results directly demonstrate that axis coding is not limited to face-selective channels, and instead may represent a more general coding principle for visual information in the VTC. The finding that a substantial proportion of axis-coding channels respond similarly to both object and face stimuli further supports this interpretation. I appreciate the effort to clarify this point, as it helps distinguish axis coding from domain-specific selectivity and highlights its broader relevance across stimulus categories.

Lastly, we further analyzed the relationship between category coding and axis coding for general objects and found that 95 out of 157 (60.5%) category-selective channels in the VTC also exhibited axis coding, suggesting that axis coding is a common representational format among category-selective channels in the VTC. Specifically in the FG, 50 out of 66 (75.76%) category-selective channels also exhibited axis coding.

This additional analysis meaningfully strengthens the conclusion that axis coding serves as a general representational format within the VTC, particularly among category-selective channels. The high proportion of overlap—especially in the FG—provides compelling support for the idea that axis-based representations are a core feature of category-selective processing in the human ventral visual stream.

4. Relationship between regional coding and semantic coding

We thank the reviewer for this important question. First, we apologize for not clearly stating that semantic clustering emerges within the neural feature space constructed by visual axes (Fig. 1). Specifically, although the neural feature space is based on visual features, it ultimately organizes information semantically. Therefore, the neural feature space—and the region coding of MTL neurons within it—translates feature-based representations in the VTC into semantic-based representations in the MTL. We have clarified this key point in the Abstract, Introduction, and in the legend of Fig. 1. Please also refer to our detailed responses to Reviewer 4's first two questions.

Third, we further examined whether abstract category representations or purely visual features better align with region coding in the MTL using representational similarity analysis (RSA). To this end, we constructed neural, visual, and semantic representational dissimilarity matrices (RDMs). Specifically, we built a category-by-category visual RDM using DNN features (Revision Fig. 6a, left), which we previously used to test axis coding. Similarly, we constructed a neural RDM using responses from all region-coding neurons in the MTL (Revision Fig. 6a, middle). To build the semantic RDM (Revision Fig. 6a, right), we applied the SGPT model 26, a large language model, to extract semantic embeddings from text descriptions of each image. These text descriptions were generated using the ALBEF model based on the object images 27. We found that MTL region-coding neurons exhibited greater representational similarity with semantic representations (Spearman's $\rho = 0.26$, $P = 0.0005$) than with visual representations (Spearman's $\rho = 0.16$, $P = 0.0009$), establishing a link between region coding and semantic representation in the MTL and supporting the idea that region coding serves as an intermediate mechanism for transforming visual features into semantic object representations.

In summary, given that region coding organizes representations semantically, it should be considered a semantic transformation rather than a mere extension of perceptual processing. The theoretical framework describing how region coding bridges perceptual and semantic representations has been made more explicit in the revised manuscript. Please refer to our substantially revised Abstract, Introduction, and Fig. 1 legend.

I appreciate the authors' clarification. In particular, the point that the neural feature space is constructed based on visual axes yet naturally gives rise to semantic clustering is both interesting and compelling. This perspective effectively highlights the role of region-based coding as a potential bridge between perceptual and semantic representations. I also commend the authors for reflecting this conceptual framework in the revised Abstract and Introduction. Nonetheless, I still find it somewhat unclear how regional coding in the MTL gives rise to semantic representations. While the example I previously gave regarding V1 might not have been ideal, the broader point remains: in visual processing, it is quite common for higher-level receptive fields to emerge from the integration or convergence of lower-level feature representations.

In this sense, while the transformation observed between the VTC and MTL may be interpreted as semantic in nature, it could also reflect a more general hierarchical integration process, similar to what is observed between intermediate visual areas such as V4 and IT. That is, the emergence of apparent semantic structure may not necessarily imply a dedicated semantic transformation, but could arise from the cumulative convergence of feature representations.

Therefore, interpreting this as a semantic transformation through "regional coding" requires caution, especially because the term "regional coding" itself can carry implicit semantic connotations. To avoid circular reasoning, it would be preferable to explicitly define "regional coding" in non-semantic terms and describe its relation to semantic structure as reflective or suggestive, rather than causative.

That said, it is also possible that the nature of convergence differs between these stages, and future work may help elucidate whether and how the VTC-to-MTL transformation departs from purely perceptual hierarchical integration, and to what extent it reflects uniquely semantic processing.

The new analyses using representational similarity matrices (RDMs) based on visual features, neural data, and semantic embeddings are particularly helpful in grounding the interpretation. This corresponds precisely to the RSA (Representational Similarity Analysis)-based comparative analysis I proposed in Major Point 1. The authors' implementation of RDM comparisons between neural, visual, and semantic representations aligns well with the suggestion to use RSA as a complementary method for evaluating model fit and exploring the nature of representational structures. Given its utility, I would recommend applying this approach to other models as well, such as the axis-based and category-based models, to enable a more comprehensive comparison across representational frameworks.

Minor Points

1. *We have corrected it in the revised manuscript. In the figure legend, we now clarify that "neural response" can refer to iEEG high-gamma power (HGP), as in the present study, or to single-neuron activity, as in classical studies of axis coding in the*

VTC.

Thank you for clarifying in the revised legend.

2. *We thank the reviewer for pointing this out. We have corrected it in the revised manuscript.*

Thank you for correcting the color assignment.

3. *We thank the reviewer for the suggestion and we agree with the reviewer. We have replicated the results using response peak latency, and consistent with the findings based on response onset latency, we found that response peak latency in the FG (419.5 ± 61.4 ms [median \pm SD]) was significantly shorter than in all other regions (IT: 460.5 ± 40.8 ms; PH: 472.0 ± 32.5 ms; AH: 474.5 ± 38.9 ms; amygdala: 462.5 ± 43.0 ms; all P s < 0.0001 ; Revision Fig. 7).*

Thank you for addressing this point. The inclusion of response peak latency analyses (Revision Fig. 7) strengthens the claim that the observed latency differences reflect genuine processing hierarchies rather than being confounded by response amplitude. The consistency between onset and peak latency results supports the robustness of the hierarchical interpretation.

4. *We thank the reviewer for the suggestion and we have now provided boundary lines for the FG and ITG in Fig. 3d, with labels in the figure legend*

Thank you for implementing the suggested changes. The inclusion of boundary lines and labels for FG and ITG in Fig. 3d significantly improves the interpretability of the figure.

5. *We thank the reviewer for raising this important question. We agree that the late-onset latency of axis coding in the AH may align with region coding. In our previous study, we also demonstrated that an elevated response at the border of the feature space can drive both axis coding and region coding 1. However, it is important to note that we did not observe a significant number of iEEG channels exhibiting region coding in the AH, suggesting that axis coding may better account for the iEEG responses in this region. The lack of region-coding channels also prevented a direct comparison of the latencies between axis and region coding. We have acknowledged this in the revised manuscript:*

Thank you for the thoughtful clarification. The authors have appropriately acknowledged the temporal similarity between axis and region coding, particularly in the AH, and have discussed the limitations in distinguishing between the two at the iEEG level. The explanation that few region-coding channels were detected in the AH helps justify the interpretation that the observed activity likely reflects axis coding. The added discussion in the revised manuscript adequately addresses the concern. I agree that future studies are warranted to dissociate these mechanisms more directly.

6. *We thank the reviewer for pointing this out, and we agree with reviewer that response latency could be influenced by response amplitude. To address this, we normalized the response of each channel to its maximum amplitude before calculating response latency and obtained consistent results: we observed a significant correlation between the onset latency of axis coding and the y-coordinate in MNI space within the FG (Revision Fig. 8a), even after controlling for individual differences in response latency (Revision Fig. 8b, as in Fig. 3g). We have included the following results in the revised manuscript:*

Thank you for performing the amplitude normalization to address potential confounds in latency estimation. To further strengthen this point, it would be helpful to show whether the normalization indeed removed any pre-existing correlation between response amplitude and latency. Specifically, could the authors report the correlation between response amplitude and latency both before and after normalization? This would clarify whether the observed latency gradient is independent of response magnitude.

7. *We thank the reviewer for pointing this out and the suggestions. First, we have included the representational dissimilarity matrices (DMS) in Supplementary Fig. 5.*

I appreciate the addition of Supplementary Fig. 5, which enhances the transparency of the RSA by showing the original dissimilarity matrices used for each ROI. The authors' discussion of the relatively low but statistically significant correlation values is well-reasoned. I agree that the distinct nature of the coding schemes—feature-based representations in the VTC versus categorical representations in the MTL—as well as the nonlinear nature of the transformation between them likely contribute to this pattern.

Interestingly, the axis-coding RDM in the FG (Supplementary Fig. 5a) exhibits a clearer block structure suggestive of category-level clustering than some of the category-coding RDMs in the MTL (Fig. 5c–e). While this might appear counterintuitive, it is consistent with the notion that visual feature spaces derived from naturalistic stimuli can preserve semantic regularities. On the other hand, category-coding channels may operate in a more sparse or selective fashion, reducing the resolution of similarity across all categories. Including a brief discussion of this apparent discrepancy would help clarify how these distinct coding schemes contribute to the transition from perceptual to semantic representations. A brief discussion of this apparent discrepancy would help readers reconcile the functional roles of these distinct coding schemes.

8. *We thank the reviewer for the suggestion. We have revised Fig. 4a to improve the visualization of the object*

representational space. Specifically, we grouped the 50 object categories into 6 broader groups and used color coding to illustrate the semantic boundaries. We also added semantic labels for individual categories at their median coordinates to provide fine-grained category information. We believe this revised format not only shows the detailed distribution of object categories within the space but also better highlights the semantic structure and boundaries.

I appreciate the improved visualization in the revised Fig. 4a. Grouping object categories into broader semantic clusters and color-coding them has greatly enhanced readability. The addition of median-category labels provides helpful granularity and clarifies how semantic boundaries emerge in a feature-based representational space. This updated format effectively illustrates the key concept that although visual features vary continuously, categorical structure becomes apparent in the neural coding scheme.

However, one element that may have been lost in the revised figure is the direct visualization of object-level visual features. To better convey the continuity of visual appearance across category boundaries, it may be helpful to provide a few example stimuli located near those boundaries. This addition could enhance the figure by more explicitly demonstrating how semantic structure emerges atop continuous visual similarity, thereby reinforcing the central claim of perceptual-to-semantic transformation.

9. We thank the reviewer for pointing this out. The response peak in the permuted density map was due to the heterogeneous (non-uniform) distribution of objects in the VTC neural feature space. We have acknowledged this methodological caveat in the revised manuscript and confirmed that our procedure reliably identifies coding regions even under heterogeneous distributions:

Thank you for addressing the concern regarding the residual response peak in the permuted density map. Your explanation—attributing the effect to a heterogeneous distribution of object representations in the VTC neural feature space—and your acknowledgment of this caveat in the revised manuscript are appreciated.

10. We thank the reviewer for the suggestions. Our goal here is to provide a group-level summary to present a comprehensive encoding profile of region-coding neurons across MTL subregions. In other words, the purpose of this analysis is to characterize the coding properties of MTL neurons within the VTC-defined feature space, rather than to compare the coding properties of MTL and VTC directly. Furthermore, the feature space in this analysis was constructed using VTC responses, so it is not appropriate to use this space to study VTC responses themselves. We have further elaborated on how these results support the concept of region-based coding:

Thank you for your clarification, while I now better understand that the goal of Fig. 4e and 4f is to characterize the encoding properties of region-coding neurons within the VTC-defined feature space—rather than to directly compare MTL and VTC—the current metrics (i.e., number of categories and number of images per tuning region) may not fully capture the key claim that these neurons selectively respond to visual features that cut across semantic category boundaries.

For instance, although the average tuning region contains relatively few categories and images, this alone does not clarify whether the encoded regions correspond to narrow semantic domains or to non-semantic clusters of visual similarity. The interpretation that region-coding neurons are sensitive to continuous visual features rather than categorical labels would be more directly supported if the preferred images spanned multiple categories despite visual coherence, or if the image set showed low semantic but high visual similarity.

To more robustly support this interpretation, I suggest including additional analyses or visualizations—such as examples of selected image clusters per neuron overlaid with category labels, or quantifications of within-cluster semantic entropy or visual similarity. These additions would help clarify whether region-based tuning indeed transcends semantic category boundaries, thus strengthening the central claim of region-based feature representation in MTL neurons.

11. We thank the reviewer for the comments and suggestions. We have now provided the exact percentage of detected region-coding channels in the text:

Thank you for this thorough and thoughtful revision. The authors now provide a clear explanation for the absence of region coding in MTL iEEG recordings, including both the spatial pooling of signals and possible differences in coding detectability between single-unit and mesoscopic recordings. Including the detection rate (3.31%) was also helpful for interpreting the result.

12. We thank the reviewer for the suggestion. We have updated Fig. 5g to enable a more intuitive comparison between the ImageNet and COCO datasets. We have also noted that the same scale is used as in Fig. 4g-i.

Thank you for addressing this point.

13. We thank the reviewer for the suggestions. First, we would like to clarify that the RSA in Fig. 3h was conducted across channels—that is, we computed image-by-image correlations using all channels from a brain area, and then correlated the resulting image-by-image dissimilarity matrices between brain areas (see Methods for details). In contrast, the PLV analysis in Fig. 6 was conducted between each pair of channels across brain areas. Therefore, we could not directly correlate the representational similarity results from Fig. 3h with the PLV results. However, we qualitatively compared the results from Fig. 3h with those from Fig. 6

Thank you for the clarification. The finding that FG-AH pairs exhibit strong PLV in the low-frequency bands (theta and alpha), in line with their high representational similarity (as shown in Fig. 3h), provides converging evidence for a functional connection between these regions. While I understand the technical limitations that prevent direct correlation between PLV

and RSA, your qualitative comparison meaningfully supports the overall framework.

14. *We thank the reviewer for the suggestions. First, we have included the PLV results for each MTL subregion (PH, AH, and amygdala) in the revised Fig. 6a, b. We apologize for omitting the PH in the previous manuscript; however, the PH did show consistent results. We have included the following results in the revised manuscript (we have also included the PH in Fig. 6 and Supplementary Fig. 7):*

Thank you for incorporating the subregion-specific PLV analyses in the revised Fig. 6a and 6b. The dissociation of frequency-specific PLV patterns across subregions—particularly theta-band differences for AH, alpha for PH, and delta for the amygdala—provides important insights into the functional heterogeneity within the MTL. Considering prior studies linking low-frequency oscillatory coupling between these regions to language and memory functions, this is a particularly thought-provoking finding that offers new avenues for understanding the division of labor within the MTL.

Additionally, I would like to point out that all three MTL subregions—AH, PH, and the amygdala—show consistently higher PLV with axis-coding VTC channels across multiple frequency bands, particularly in the theta and alpha ranges (Fig. 6b and Revision Fig. 9b). This suggests that the increased phase synchronization is not limited to a specific MTL subregion but may reflect a more global network-level coordination pattern driven by axis-based coding in the VTC. It would be helpful if the authors could elaborate on whether this distributed PLV enhancement reflects a general broadcast mechanism for axis-coded information, or if it may still support region-specific coding transitions when combined with representational similarity patterns (e.g., Fig. 3h).

15. *We thank the reviewer for the insightful observation and suggestions. We agree with the reviewer's interpretations regarding Fig. 6. We have included the following discussion in the revised manuscript: "The FG and ITG exhibited stronger feedforward Granger causality (GC) to the AH in the low-frequency range, suggesting that low-frequency oscillations may serve as a mechanism for transmitting visual information from the VTC to the MTL during object perception. Interestingly, this effect was more pronounced in the spectral GC analysis (Fig. 6d) than in the time-resolved GC analysis (Fig. 6e). This discrepancy may reflect differences in the sensitivity of these methods—spectral GC is better suited for detecting sustained frequency-specific interactions, whereas time-resolved GC emphasizes transient, temporally localized effects. Additionally, the feedforward influence may be more specific to the AH, with weaker effects in other MTL subregions such as the PH or amygdala. These findings highlight the importance of combining multiple analytical approaches and considering subregional specificity within the MTL when examining directed functional connectivity in the human brain."*

Thank you for providing the subregion-specific breakdown of the GC analysis. I find the discussion particularly thoughtful and appropriately cautious in interpreting the frequency- and region-specific patterns. The distinction between spectral and time-resolved GC is especially helpful, and the identification of stronger feedforward GC to the AH in the low-frequency band aligns well with the observed representational similarity. I agree that these converging lines of evidence strengthen the interpretation that the AH plays a distinct role in mediating visual–semantic integration between the VTC and MTL.

16. *We thank the reviewer for the suggestions and agree with the reviewer's insights. In the revised manuscript, we have analyzed spike–field phase consistency in the theta frequency range (Supplementary Fig. 9a-d). While we found that MTL region-coding neurons also fired spikes that were phase-locked to theta oscillations (4–8 Hz) in the VTC (Supplementary Fig. 9d)—supporting the critical role of theta oscillations in MTL processing, as noted by the reviewer—the pairwise phase coherence (PPC) of the MTL spike–VTC iEEG pairs was not significantly greater for in-region stimuli compared to out-region stimuli in the theta range (Supplementary Fig. 9d). This effect was only significant in the gamma-frequency range (Fig. 7e), suggesting that gamma oscillations may play a dominant role in MTL neurons during feature coding of visual objects. We have clarified this point in the revised manuscript and further discussed how this finding aligns with the existing literature.*

Thank you for conducting the additional analyses of spike–field coupling in the theta frequency range and for presenting the detailed results in Supplementary Fig. 9. I appreciate the effort to address the potential role of lower frequency oscillations in VTC–MTL interactions. The finding that MTL region-coding neurons exhibit theta-phase locking to VTC activity, although not significantly modulated by stimulus region, is nonetheless informative. It supports the general role of theta oscillations in MTL coordination, while also highlighting the functional specificity of gamma-band coupling in visual object processing. Overall, I find your additions and discussion to be carefully reasoned and informative.

17. *We thank the reviewer for the insightful questions and comments. First, we would like to clarify that the time window (0.1–0.6 s relative to stimulus onset) used for the iEEG analysis was determined based on the temporal profile of the response (Fig. 2e). To ensure the robustness of our findings, we replicated the main analyses using a longer time window (0–1 s relative to stimulus onset) and observed consistent results. As the reviewer noted, the MTL spike–VTC field coupling (Fig. 7d) occurs at a late latency (~400 ms), which aligns with the spiking profile of MTL neurons observed in our previous studies 1,13,31-33. Accordingly, we applied a different time window (0.25–1.25 s relative to stimulus onset) when analyzing single-neuron activity in the current study. Therefore, the time windows applied to both the iEEG and single-neuron analyses were optimized based on their respective temporal response profiles. We have included the following discussion in the revised manuscript:*

Thank you for the detailed and thoughtful response. I appreciate the authors' clarification regarding the rationale behind the use of distinct time windows for the iEEG and single-neuron analyses, and their effort to ensure robustness across temporal windows. The alignment of the late spike–LFP coupling with previous findings on MTL neuronal latencies is convincing and appropriately contextualized.

I also find the authors' interpretation—that this late coupling reflects higher-level integrative processes such as memory retrieval or semantic association—well justified. The proposed link between the late-onset axis coding in the AH and the delayed spike–field synchronization is compelling, even if a direct comparison is not feasible due to differences in signal source.

Overall, the revised discussion substantially improves the interpretability and coherence of the findings

18. *We thank the reviewer for the suggestions. In Fig. 7a-c, we presented the spike–field coupling between MTL region-coding neurons and VTC channels, separately for the FG and ITG. In the revised manuscript, we further included time-resolved PPC analyses for the FG and ITG (see the newly added Supplementary Fig. 9e). Notably, FG axis-coding channels exhibited stronger synchronization with MTL region-coding neurons compared to ITG channels, indicating that the observed*

effects were primarily driven by FG channels. This finding aligns with the RSA results in Fig. 3h, which showed greater representational similarity between the FG and MTL. We have incorporated this result into the revised manuscript: The new visualization of in-region stimuli along the most-preferred axis of phase-locked VTC channels (Fig. 7f, g) is informative and nicely illustrates the proposed functional link between VTC feature encoding and MTL region coding. The clustering of responses along a specific tuning axis is particularly compelling. That said, the example in Fig. 7g (ITG channel) appears to show stimulus distributions that may be more category-dependent, as stimuli from the same object category seem to cluster together along the axis. This raises the possibility that the axis-coding dimension in this case may reflect categorical distinctions more than continuous visual features. It may be worth discussing this point in more detail, and with greater caution, especially given the potential differences in coding strategies across VTC subregions. A more explicit analysis of the visual properties encoded by these axes (e.g., shape, category, semantics) could help clarify their representational content. Additionally, the term "upstream VTC regions" might be potentially misleading, depending on the intended direction of processing; a more neutral term such as "VTC feature-coding regions" might improve clarity.

Reviewer #4

(Remarks to the Author)

I would like to express my gratitude to the authors for their comprehensive rewrite of both the Abstract and Introduction, as well as for updating Figure 1 and its legend in response to my request. These changes have significantly enhanced the clarity of the paper. Consequently, I believe this paper should be accepted for publication.

Version 2:

Reviewer comments:

Reviewer #1

(Remarks to the Author)

the authors have answered all my comments and questions.
the manuscript is now in excellent shape.

I noted one typo: a verb (something like 'viewed') is missing in the second line of the section: 'neural responses to visual objects in the VTC and MTL'

Reviewer #3

(Remarks to the Author)

The authors have carefully addressed the previous comments, and the additional analyses have meaningfully advanced the evaluation of the proposed computational framework. Nevertheless, the responses have also revealed certain important operational issues concerning model-fitting metrics, and a few critical points remain unclear. Additional comments are therefore provided below.

Major Points:

We thank the reviewer for raising this question, and we apologize for not explaining it thoroughly in our previous response. We implemented a cross-validation procedure to evaluate model performance and calculate statistical significance. The previously reported R^2 was based on the held-out test dataset (50%). However, to determine which DNN layer best models the neural responses—in other words, which set of features best fits the data—it is more reasonable to estimate R^2 using the full dataset, as the final axis-coding models were trained on all available data. This approach not only increases the amount of data but also includes the training set, which naturally results in a higher R^2 . While we acknowledge that a large difference in performance between training and testing sets may suggest overfitting, this did not affect our DNN layer selection. Both the testing dataset (previous Supplementary Fig. 2a) and the full dataset (current Supplementary Fig. 2a) consistently indicated that layer pool5 yielded the best model fit. It is also worth noting that pool5 did not have the highest dimensionality among the layers, making it less likely to be the most overfitted.

We thank the reviewer for pointing this out. This apparent discrepancy likely arises from differences in what is being measured. Specifically, Supplementary Fig. 2a shows R^2 values for all visually responsive channels, calculated using the full dataset. In contrast, Supplementary Fig. 2b and Fig. 3d show the strength of axis coding, quantified as the correlation coefficient between the observed and predicted responses in the test dataset. The predicted responses were generated using models trained only on the training dataset (see Methods for details). It is likely that the models fit the data equally well across channels, but the inherent predictability varied—reflecting differences in data consistency. In summary, R^2 values reflect how well the model explains neural responses and are used to compare relative model fits across feature layers, whereas the strength of axis coding reflects generalization performance.

Thank you for the clarification regarding the R^2 analysis. However, the current approach still raises concerns regarding potential overfitting and the interpretability of model comparisons.

As described, the R^2 values were estimated using models trained and tested on the full dataset, which likely inflates the R^2 values, particularly for the more flexible PLS models. Notably, much lower R^2 values (~ 0.1) were previously reported when evaluated on held-out data, in contrast to ~ 0.6 on the full dataset. This discrepancy suggests substantial overfitting and

raises questions about the appropriateness of using full-dataset R^2 as an indicator of model generalization. It is understood that the authors intentionally distinguish between model fit (as assessed by full-dataset R^2) and generalization (as assessed by test-set correlation coefficients), and this clarification is appreciated. However, the fundamental problem is that they still use the fitting-based R^2 for comparisons even in situations where generalization performance should be evaluated. In particular, Supplementary Fig. 2d directly compares PLS and ANOVA models based on their respective R^2 values. Given the greater flexibility of the PLS model, such comparisons may inadvertently favor it over the more constrained ANOVA model, irrespective of actual generalization performance. To ensure a fair and interpretable comparison of model performance, it would be preferable to evaluate both models on held-out test data or, at the very least, to report cross-validated R^2 values. Presenting both test-set and full-dataset R^2 values in parallel would enhance transparency, provide insight into the degree of overfitting, and better align with standard practices in the literature (e.g., Yamins et al., PNAS, 2014; Elmoznino et al., PLOS Comput Biol, 2024). Importantly, the authors' additional RSA analyses provide multifaceted evidence that axis-based models derived from DNN features outperform the semantic model (Revision Fig. 1). This is a valuable contribution that is not undermined by the above concerns. Although, the absence of category (ANOVA) model in the RSA analyses appears to be a limitation. Moreover, as I will discuss later, the claim that category models cannot be applied may not be entirely accurate.

We thank the reviewer for pointing this out, and we agree that differences in model fitting can be influenced by differences in input dimensionality. To control for this, we additionally calculated R^2 using a one-dimensional feature (the first PLS component), ensuring that both the PLS model and the ANOVA model had one-dimensional input. Although, as expected, R^2 was reduced when we restricted the PLS input dimensionality, the PLS model still showed a better fit even under this constraint for both the FG (axis: 0.22 ± 0.06 [mean \pm SD]; category: 0.17 ± 0.087 ; two-tailed paired t-test: $t(200) = 54.84$, $P < 10^{-20}$) and ITG (axis: 0.20 ± 0.04 ; category: 0.12 ± 0.046 ; $t(140) = 56.25$, $P < 10^{-20}$). We have included the following results in the revised manuscript:

Thank you for performing the additional analysis using the first PLS component. This successfully controls for input dimensionality and shows that the PLS model still outperforms the ANOVA model under this constraint, which is informative. However, the comparison remains based on full-dataset R^2 values, which are sensitive to overfitting. Therefore, even when dimensionality is matched, such comparisons would be more informative and interpretable if based on held-out or cross-validated R^2 values. It is particularly important to minimize the influence of potential overfitting when evaluating model performance. This is especially true in the present case, where, after controlling for other factors, the R^2 values of different models become similar—highlighting the need for careful interpretation. Without this, it remains difficult to assess whether the observed advantage of the PLS model reflects genuine generalization performance or simply better fit to the training data. This issue still warrants further consideration.

In addition to model comparisons, a more systematic use of RSA analyses applied to both the VTC and MTL could help clarify the respective modes of visual object coding in these regions of the human temporal lobe. (This point is further elaborated later in the review.)

It is important to note that while the highest DNN layers may correspond more closely to semantic representations, the VTC may more closely align with intermediate DNN layers (see Supplementary Fig. 2a for details), which retain rich visual information without abstracting fully to semantic categories. This intermediate-level feature representation in both VTC and DNNs may explain why the axis model—derived from DNN features—is more successful than a coarse category model in capturing VTC activity. Thus, the relatively low R^2 values of the category model likely reflect both the limitations of discrete category labels and the nature of VTC coding itself.

According to Supplementary Fig. 2a, the layer pool5 appears to provide the best fit in terms of R^2 . If this is the case, pool5—which is located immediately before the fully connected (fc) layer—would be more appropriately described as the “penultimate layer” or a “high-level semantic layer” rather than an “intermediate layer”. Thus, the current description may be somewhat inaccurate.

Additionally, while it may not be entirely clear whether this point belongs here, I would like to note that in Supplementary Fig. 2a (R^2), there is a clear difference between res4b22, res5, and fc, whereas in Supplementary Fig. 2b (Pearson's r), these layers show minimal differences. This likely reflects fundamental differences between the two metrics—for example, their sensitivity to bias and scaling. Although the authors have already explained the conceptual differences between these two measures earlier in the manuscript, a brief clarification of how these differences manifest in Supplementary Fig. 2a and 2b would be appreciated and may help readers better interpret the results. Furthermore, it would be helpful if the authors could explicitly state which ResNet architecture (e.g., ResNet-50, ResNet-101, etc.) was used in the analysis, as this information is currently not clearly specified.

We thank the reviewer for the thoughtful comments and suggestions. We agree that incorporating alternative models that account for lexical or semantic distances would be valuable for understanding which features best explain VTC responses. As suggested by the reviewer, we extracted word-label embeddings for the 50 included categories using the Global Vectors for Word Representation (GloVe) 17 model. We then compared the correspondence between VTC neural responses and word embeddings versus visual features using RSA. We found that visually responsive VTC channels (Revision Fig. 1a) exhibited a higher, though not statistically significant, correspondence with visual features (Spearman's $\rho = 0.23$) compared to word embeddings (Spearman's $\rho = 0.13$; permutation test against the null distribution of differences: $P = 0.13$). This pattern held when considering the entire population of VTC channels (Revision Fig. 1b), suggesting that category-based coding does not better explain VTC responses at the population level. As expected, channels classified as axis-coding (Revision Fig. 1c) demonstrated a significantly higher correspondence with visual features (Spearman's $\rho = 0.39$) than with word embeddings (Spearman's $\rho = 0.22$; $P = 0.005$). Notably, the same pattern was observed even in category-selective channels (Revision Fig. 1d). Together, these results support the conclusion that visual models provide a better explanation of VTC neural responses than word-based category models.

We thank the reviewer for the clarification and we agree with the reviewer. In the revised manuscript, we have consolidated the comparison between the PLS and ANOVA models (see response above). Additionally, as in the previous Revision Fig. 6, we further compared axis-based and lexical/semantic-based coding using RSA (as categorical labels used in the ANOVA model could not be used to construct a dissimilarity matrix). Please refer to our response to the question above.

Thank you for incorporating the analysis based on word embeddings and comparing it with visual feature models using RSA. This is a valuable addition that further clarifies the nature of neural coding in the VTC. Applying such analyses to rare and valuable human neurophysiological data is especially meaningful, particularly given that similar analyses in non-human primates might face limitations in interpretability or generalizability.

It is notable that even category-selective channels aligned better with visual features than with word embeddings, supporting the idea that category selectivity in the VTC may emerge from continuous tuning to visual features. It may be helpful to relate this explicitly to the results from the ANOVA-based category model, which assumes a discrete structure. Clarifying this relationship would help unify the findings across the modeling approaches.

I also found it somewhat difficult to fully understand the statement that an ANOVA model “could not be used to construct a dissimilarity matrix.” Initially, I thought this might reflect the limitation that each stimulus was presented only once, which would prevent reliable estimation of stimulus-level responses. However, looking at Supplementary Fig. 5 (where a neural dissimilarity matrix appears to have been constructed for the MTL), this may not be the case. If trial-level responses for each stimulus can be obtained, it should be possible to construct a category-based idealized dissimilarity matrix under the ANOVA model—for example, one in which within-category pairs are coded as 0 and between-category pairs as 1. If stimulus-level analyses are indeed infeasible in the VTC, one potential workaround would be to construct such matrices based on coarser categories, as in the R^2 analyses with broader category groupings. In fact, systematically comparing different category granularities could itself be informative.

More broadly, if both R^2 -based model fitting and RSA were systematically applied to the VTC and MTL using visual features (e.g., from DNNs), word embeddings (e.g., GloVe), and category-based models (at various levels of granularity), it may help clarify the computational basis of axis versus region coding, and shed further light on how visual representations are transformed into more abstract, conceptual formats along the human temporal lobe.

We thank the reviewer for the suggestions. The reviewer has accurately captured our proposed computational framework, and we agree with the insightful conceptualization of “a hierarchy from axis coding (low-level), to region-based coding (intermediate), to label-based category coding (high-level), with the corresponding models being the PLS model, semantic embedding models, and the ANOVA model, respectively”. Furthermore, the reviewer is correct that between-region comparisons are challenging, whereas within-region comparisons using data from a single measurement modality are more feasible. Accordingly, we used iEEG responses for model comparisons in the VTC, and single-unit responses for model comparisons in the MTL. Specifically, as shown in Revision Fig. 1, VTC neural responses were better explained by a visual model than by a lexical/semantic model. In contrast, as shown in our previous Revision Fig. 6, MTL region-coding neurons exhibited greater representational similarity with semantic representations than with visual representations (similar results were obtained using the GloVe model). These findings support the proposed transition in coding models from the VTC to the MTL. It is worth noting that the ANOVA model could not be evaluated using RSA, as the categorical labels it relies on cannot be used to construct a dissimilarity matrix. Furthermore, while explanatory power (R^2) could not be calculated for semantic features—since no specific predictive model was assumed—RSA is more appropriate in this case. Nonetheless, the R^2 based analyses (see above) further support visual, rather than category-based, coding in the VTC.

Thank you for the clear and thoughtful response. I appreciate the structured articulation of the proposed hierarchy, from axis to region-based to label-based coding, and the associated use of PLS, semantic embedding, and ANOVA models. This framework offers a helpful conceptual lens through which to interpret the observed transition from VTC to MTL.

Regarding the point that the ANOVA model could not be used for RSA, it would be helpful to clarify the nature of this limitation. As mentioned in the previous section, if stimulus-level neural responses are available, it should be possible to construct an idealized dissimilarity matrix. I wonder whether this approach was considered, and if so, whether any specific challenges prevented its application in this case.

We thank the reviewer for this insightful and constructive feedback. We appreciate the reviewer’s recognition of our conceptual framework; and we agree that caution is needed when interpreting the transformation from VTC to MTL as “semantic”. We acknowledge that such convergence could be the result of a general hierarchical integration process, similar to those observed along the ventral visual stream (e.g., V4 to IT), rather than a transformation uniquely tied to semantics. We now highlight this alternative possibility in the revised Discussion, emphasizing that the apparent semantic structure in MTL may reflect the cumulative integration of lower-level visual features, and not necessarily a dedicated semantic operation: “An alternative interpretation of the VTC-to-MTL transformation is that the observed representational changes reflect a general hierarchical integration process, rather than a dedicated transformation. Similar to how receptive fields in early visual areas converge to form higher-order visual representations in regions such as V4 and IT, the emergence of region-based coding in the MTL may arise from the cumulative integration of visual features represented in the VTC. In this view, the clustering of neural responses in MTL—sometimes aligning with object categories—could result from overlapping visual feature selectivity, rather than an explicit coding of semantic content. This interpretation is consistent with known principles of hierarchical organization in sensory systems and suggests that apparent semantic structure may emerge naturally from the integration of complex visual information, even in the absence of explicit category representations. Our findings that region-based tuning in the MTL transcends strict category boundaries further support this view. Future work will be critical to determine whether the transformation from perceptual to conceptual representations in the MTL reflects purely integrative processes or additional mechanisms that encode abstract conceptual meaning.” It is worth noting that our current data suggest that MTL region coding captures structured information that aligns more closely with semantic embeddings than with purely visual similarity, as supported by the RSA (please see above). While this correspondence is suggestive, we agree with the reviewer that further studies are needed to disentangle whether the VTC-to-MTL transformation reflects

uniquely semantic processing or a broader convergence process.

Thank you for explicitly addressing the alternative interpretation that the transformation from VTC to MTL may reflect a general hierarchical integration process rather than dedicated semantic coding. This addition makes the Discussion (with Supplementary) more balanced and conceptually rich. In particular, I appreciate the clarification that apparent semantic structure may emerge naturally from the accumulation of visual feature selectivity, as seen in other parts of the ventral stream. The use of both RSA and R^2 -based analyses provides complementary evidence and helps clarify the extent to which MTL coding reflects semantic versus perceptual structure. This issue will indeed be an important direction for future work.

Minor points :

We thank the reviewer again for the suggestion. First, as expected, we found a pre-existing correlation between response amplitude (i.e., axis-coding strength) and latency before normalization ($r = -0.43$, $P = 9.77 \times 10^{-5}$). After normalization (i.e., response amplitude normalized to 1), this correlation was eliminated ($r = 0$, $P = 1$). Therefore, the observed latency gradient was independent of response magnitude.

Thank you for including this follow-up analysis. This strengthens the interpretation that latency differences reflect meaningful functional distinctions beyond mere signal strength.

We thank the reviewer for the suggestion. We agree and have included example object images located near category boundaries to better illustrate how semantic structure emerges atop continuous visual similarity. A new supplementary figure has been added to the revised manuscript:

The addition of example images near category boundaries in the VTC feature space is very helpful. I found this figure particularly insightful and evocative. Since both Fig. 4a and Supplementary Fig. 6 depict essentially the same underlying feature space, this parallel is a strength, but it may also be perceived as slightly redundant. To make the figures more complementary, one option could be to color-code Supplementary Fig. 6 by the six broader super categories (e.g., animal, equipment, etc.), rather than by all 50 fine-grained categories. In this way, Fig. 4a would continue to emphasize semantic structure and category labels, while Supplementary Fig. 6 would highlight visual continuity and example stimuli, helping to more clearly differentiate the respective roles of the two figures.

We thank the reviewer for this excellent suggestion. To further illustrate region-based feature representations in MTL neurons, we visualized object images that fell within the tuning regions of example region-based neurons (upper panels). Additionally, we generated word clouds to qualitatively depict the relative frequency of object categories represented by each neuron (bottom panels). These illustrations show that most object images encoded by Cell #586 (Fig. 4b) exhibit nested structures, while those represented by Cell #630 (Fig. 4c) feature radical shapes and small sizes. Notably, the clustered images spanned a variety of semantic categories. Furthermore, the number of images from each category within the tuning region varied, and many images from the same category were located outside the tuning region—suggesting that the clustering was driven more by visual features than by semantic meaning. Together, these results indicate that region-based tuning indeed transcends semantic category boundaries.

The addition of Fig. 4b and 4c, along with the word clouds, is highly effective in illustrating the specific nature of region-based tuning in MTL neurons. The descriptions such as “nested structure” and “radical shape” offer compelling insight into how tuning appears to rely on visual features rather than semantic categories.

Interestingly, these visual characteristics might themselves reflect higher-order abstract concepts, which makes the findings particularly intriguing. It could be helpful to relate these qualitative examples to the quantitative analyses suggested above involving semantic categorization at multiple levels, as this would help ground the interpretation of tuning properties more firmly in the data.

We thank the reviewer for the insightful observation and suggestions. We agree that the increased phase synchronization is not limited to a specific MTL subregion, but may instead reflect a broader, network-level coordination pattern driven by axis-based coding in the VTC. Accordingly, this distributed PLV enhancement may represent a general broadcast mechanism for axis-coding information. We have expanded our prior discussion as follows:

Thank you for the thoughtful and well-integrated revision. The expanded discussion clarifies how widespread PLV enhancement across the MTL may reflect a common broadcast mechanism originating from axis-based coding in the VTC. We thank the reviewer again for the helpful suggestions and comments. We agree that the example in Fig. 7g (from an ITG channel) shows stimulus clustering that appears more category-dependent, suggesting that in some cases, axis-coding dimensions may partially reflect categorical distinctions. This is not unexpected, for two reasons: (1) region coding inherently encompasses and explains category selectivity and there are coding regions that contain stimuli primarily from a single category; and (2) object images from the same category often share similar visual features, which are preserved along the coding axes, thereby leading to their clustering along those axes. However, this observation highlights the diversity of representational content across VTC subregions and the possibility that axis coding can span both lower-level visual features and higher-level categorical structure. Therefore, we have included the following discussion in the revised manuscript:

The authors have provided a thorough and compelling discussion of the dual aspects of axis coding, capturing both continuous visual features and, in some instances, category-related structure. This interpretation is further supported by the analyses presented in Supplementary Fig. 3 and Fig. 7g, which suggest that axis-coding may capture not only continuous visual features but also aspects of categorical meaning.

To further strengthen this claim, it may be valuable to explore whether the encoded axes align with semantic dimensions using more explicit model-based approaches. For example, PLS regression could be used to predict axis directions from semantic or visual features, and semantic embeddings could serve as a basis for constructing representational dissimilarity matrices (RDMs) for RSA comparisons. Additionally, applying ANOVA models along the axis direction could help quantify the extent to which categorical information is captured. These approaches may help clarify the representational nature of axis coding across the VTC.

Version 3:

Reviewer comments:

Reviewer #3

(Remarks to the Author)

Comments

I sincerely thank the authors for their persistent and careful efforts in addressing the concerns I raised. Their additional analyses have enhanced the transparency of the work and provided a reasonable interpretation of the valuable dataset. As a result, the overall quality and completeness of the manuscript have significantly improved.

Although there may be a difference in perspective regarding the assumptions underlying the RSA analyses, it is recognized that RSA was originally introduced here as one of multiple approaches to evaluate the plausibility of axis- and region-based coding schemes. In this regard, the conclusions of the present study are considered to be sufficiently supported by the current set of analyses.

Further comments are provided below.

We thank the reviewer for the additional comments. First, as suggested, we have now included R² values from both the test dataset and the full dataset in the revised Supplementary Fig. 2a to enhance reporting transparency. Notably, both the held-out test dataset and the full dataset consistently indicated that layer pool5 yielded the best model fit. Supplementary Fig. 2. (a) Goodness-of-fit (R²) of the partial least squares (PLS) regression with deep neural network (DNN) features for each DNN layer. The layer with the highest R² was selected for further analysis. Error bars denote \pm SEM across channels. R² was calculated using both the full dataset (solid line) and a held-out test dataset (dotted line; see Methods). Both datasets indicated that layer pool5 provided the best fit to the neural responses.

Thank you for performing the additional analysis. It is reassuring to see that the superiority of layer pool5 is consistently supported. However, when accounting for potential overfitting, its advantage appears to be less pronounced. The systematic patterns of layer dependence and regional differences observed in the test dataset—particularly given the small error bars—suggest that these effects are unlikely to be due to noise or random variability, but may instead reflect meaningful structure. While the conclusion that pool5 offers the best overall fit appears robust, the relatively high R² values observed for intermediate layers (e.g., res3b3) in the test dataset may indicate that, in certain regions, intermediate layers provide a better explanation of neural responses. Furthermore, the larger discrepancy between the full and test datasets at higher layers suggests that features from upper layers, while fitting the training data well, may have limited generalizability to novel data. This observation is particularly intriguing and may warrant further discussion. If space permits, it could be helpful to briefly note these points in the Supplementary Discussion. Of course, the final decision is up to the authors.

Second, we have now included a comparison with the ANOVA model using a held-out test dataset in the revised Supplementary Fig. 2d. This ensures that the comparison is transparent while acknowledging the methodological differences between the two models. We have clarified this point explicitly in the revised Supplementary Fig. 2d legend. It is important to note that ANOVA does not involve training/testing procedures or cross-validation; its R² is computed directly from the full dataset and thus does not provide an estimate of generalization performance.

Thank you for the additional analysis. While I appreciate the clarification that ANOVA, in its standard implementation, does not involve training/testing or cross-validation procedures, I would like to note that, in principle, cross-validation can be applied to ANOVA models as well, since they are linear models with discrete predictors. The risk of overfitting is generally lower compared to more flexible models such as PLS, but the distinction remains methodological rather than fundamental. Regardless of the issue of training/testing, I find it noteworthy that the R² values from the ANOVA model appear higher in the fusiform gyrus (FG) than in the inferior temporal gyrus (ITG). The lack of difference between ANOVA and PLS models in FG may reflect the possibility that the relationship between axis coding and categorical boundaries is not necessarily consistent across regions. This could suggest that the nature of feature representation—and its alignment with categorical structure—differs between FG and ITG.

Lastly, we thank the reviewer for recognizing the contribution of our RSA analyses. Regarding the category (ANOVA) model, it is important to note that ANOVA is categorical by design, and categorical labels are not well suited for computing dissimilarity values in RSA (please refer to our response below for more details). Because RSA requires continuous variables to calculate correlations, we did not test the ANOVA model in our previous analysis (but we explored this analysis in this revision). We appreciate the opportunity to clarify this conceptual distinction, and we have further fleshed out this analysis below.

Thank you for the conceptual clarification and the additional analyses. However, I believe the argument that “ANOVA is categorical by design, and categorical labels are not well suited for computing dissimilarity values in RSA (please refer to our response below for more details). Because RSA requires continuous variables to calculate correlations” may not be logically robust. The core objective of representational RSA is to assess the correspondence between the representational structure proposed by a model—whether continuous or discrete—and the neural similarity structure.

What the authors may have intended to emphasize is the following: when categorical labels are used to construct an idealized RDM, all within-category pairs receive a value of 0 and all between-category pairs a value of 1. As a result, such RDMs exhibit minimal variability across stimulus pairs, which limits the sensitivity and resolution of correlation-based comparisons with neural RDMs. In this sense, discrete RDMs may be at a disadvantage for RSA, not because they are inherently incompatible, but because they are less capable of capturing fine-grained similarity patterns across stimuli. Therefore, it would be more appropriate to describe this as a limitation rather than an incompatibility. This is precisely why

combining RSA with other approaches such as R^2 -based model fitting can be advantageous in capturing complementary aspects of the neural representational structure.

We thank the reviewer for the additional comments. We have now calculated R^2 using the first PLS component with a held-out test dataset. The PLS model still showed a better fit for the held-out test dataset in both the FG (axis: 0.27 ± 0.042 [mean \pm SD]; category: 0.17 ± 0.087 ; two-tailed paired t -test: $t(127) = 9.56$, $P = 1.24 \times 10^{-16}$) and ITG (axis: 0.25 ± 0.036 ; category: 0.12 ± 0.046 ; $t(45) = 10.97$, $P = 2.66 \times 10^{-14}$). This control analysis further strengthens the conclusion that axis-based models provide a more accurate account of neural responses than category models in the human VTC.

Does this correspond to the right panel of Supplementary Fig. 2d? The legend and the statistical results reported here seem inconsistent, so I would appreciate it if you could double-check this. It may also be helpful to revise the main text accordingly to ensure clarity and consistency. The final decision is entirely up to the authors.

We thank the reviewer for the additional comments. First, we would like to clarify that the category-selective channels were identified using the ANOVA-based category model (which is why we examined these channels), and we thus further linked this ANOVA-based model to visual feature and semantic coding. Second, while it is true that, in principle, one could construct an idealized dissimilarity matrix using categorical labels, such a matrix would consist of binary values (0 for within-category and 1 for between-category). This structure does not provide the continuous variability needed for computing meaningful correlations in RSA, as it would yield a matrix with 0s along the diagonal blocks and 1s elsewhere. In other words, the categorical nature of ANOVA prevents it from capturing graded similarities or differences across stimuli, which are central to RSA analyses. For this reason, we did not apply the ANOVA model within the RSA framework in our previous revision. We appreciate the opportunity to clarify this conceptual limitation.

However, here, we explored this analysis by correlating the idealized RDM from the ANOVA model (Revision Fig. 1a) with VTC (Revision Fig. 1b) and MTL (Revision Fig. 1c) channels, and compared these results with correlations obtained using visual and lexical representations. We found that in the VTC, visual features explained neural responses better than lexical/word features or categorical labels (permutation test: visual versus ANOVA category: $P < 0.0001$; word versus ANOVA category: $P = 0.21$). In the MTL, lexical/word features provided a better explanation of neural responses (visual versus ANOVA category: $P = 0.55$; word versus ANOVA category: $P = 0.042$). These results are consistent with our proposed computational pathway.

While I understand the authors' rationale for avoiding categorical (binary) RDMs in RSA, I believe the argument could benefit from further clarification. The authors note that "the categorical nature of ANOVA prevents it from capturing graded similarities or differences," which is true in terms of continuous variability. However, the core aim of RSA is to evaluate the correspondence between the structure of representational spaces—regardless of whether they are continuous or discrete—and neural similarity matrices. From this perspective, the discrete structure of category-based models does not inherently disqualify them from RSA analyses.

As I mentioned earlier, a more precise framing might be that binary category-based RDMs offer limited resolution, with all within-category pairs assigned 0 and all between-category pairs assigned 1. This coarse granularity could reduce sensitivity in correlational analyses, particularly when comparing models. Nonetheless, this should be regarded as a limitation in discriminability or resolution, rather than a categorical inapplicability.

In this regard, I appreciate the authors' effort to conduct RSA, which I believe adds important value to the manuscript. However, regarding Revision Fig. 1a: if the "Idealized RDM" was constructed based on ANOVA categorical labels, one would expect to see square-shaped diagonal blocks, each with a size corresponding to the number of stimuli within each category. In contrast, Fig. 1a currently shows nearly uniform values (presumably 1) across both within-category and between-category pairs, which does not appear to visually reflect the key structural features of an idealized category RDM. The revised results themselves are highly interesting, but since this point is not explicitly addressed in the current version of the manuscript, I leave it to the authors to decide whether and how to clarify or revise the presentation.

We thank the reviewer again for this valuable suggestion. Indeed, these are important directions for further analysis. In fact, we have examined these questions in a separate manuscript (Wang Y, Brunner P, Willie JT, Cao R, Wang S. Neural computations of visual, semantic, and memorability features in the human brain. PsyArXiv), using the same dataset (with a few additional patients). In that work, we applied PLS regression to construct axis-coding models for visual and semantic features and used semantic embeddings to build RDMs for RSA comparisons. We conducted a detailed characterization and comparison of visual and semantic models along the ventral visual pathway, including both the VTC and MTL. It is also worth noting that Supplementary Fig. 3 of the current manuscript illustrates the alignment of axis-coding channels with model-based visual feature dimensions.

The authors' responses and additional analyses are much appreciated. Future work may benefit from further examining the psychological plausibility of modeling category representations using ANOVA or alternative frameworks, and from conducting a detailed comparison of visual, semantic, and category models along the ventral visual pathway. Overall, the study provides a clear and important contribution by applying careful analyses to a valuable dataset, offering new insights into the neuronal transformations within the ventral visual stream.

Reply to comments from Reviewer 1

The iEEG data and single cell results from human temporal cortex reported by Cao et al are invaluable. In general such data are very much in demand as it becomes increasingly clear that fMRI data suffer from a very serious temporal resolution problem. More specifically, the authors report axis tuning in human FG, IT gyri and MTL extending results from non-human primates, as well as a very neat tuning of MTL single neurons in the feature space defined by the feature axes observed in the iEEG temporal leads. Both are very nice findings.

Yet the manuscript in its present form includes several severe flaws

I the basic model outlined in fig 1 is not proven: the authors insist that area in feature space coding can only be detected with single cell recording, not iEEG. As there are no single cell recordings in FG nor IT gyri, we do not know whether such tuning does not already exist in FG or IT cortex. Thus it might be that the area in feature space is constructed in IT cortex and then simply transmitted to MTL (and perhaps other regions such as the temporal pole; (TP); this is the more a problem that at iEEG level little difference between TE and MTL was reported, if anything leads were more responsive and selective in TE;

We thank Dr. Orban for the expert and constructive comments.

This is an important question. Dr. Orban is correct that the present study does not include single-neuron recordings in the fusiform gyrus (FG) or inferior temporal gyrus (ITG). However, in a separate study investigating the same computational framework for faces (using the identical task but with faces only), we obtained cellular recordings from a monkey's inferotemporal cortex (ITC; analogous to the human VTC). Specifically, we recorded from 53 multi-unit activity (MUA) channels. By applying the same procedure to assess axis coding on these MUA channels, we observed robust axis coding of visual features in the monkey's ITC ($n = 53$, 100%; strength of axis coding = 0.48 ± 0.13 [mean \pm SD]). In contrast, only 16 out of 53 monkey ITC neurons (30.19%; binomial $P = 4.34 \times 10^{-10}$) exhibited region coding, a significantly lower percentage than that of axis coding (100%; χ^2 -test: $P = 4.73 \times 10^{-14}$). Notably, the significant proportion of ITC neurons exhibiting region coding was likely driven by their strong axis coding, as an elevated response at the border of the feature space can drive both axis and

region coding ¹. Therefore, unlike the MTL, the ITC/VTC primarily exhibited axis coding rather than region coding. This suggests that region coding in the MTL was not simply transmitted from the VTC.

2 The authors make a constant confusion between semantic and episodic memory. The question addressed here is visual object identity (important for recognition), even if stimuli we organized in arbitrary categories, not the properties of concepts typical of semantic memory. There is considerable evidence that semantic processing occurs in TP (Lambon Ralph et al Nat NS Rev 2017, Tiesinga et al Scientific reports 2023), even if MTL neurons have been labelled concept cells to state that they are sensory invariant.

We thank Dr. Orban for pointing this out. First, we would like to clarify that we did not mention or make any arguments regarding episodic memory. Our present results pertain solely to visual object processing. Second, Dr. Orban is correct that our focus is on visual object identity, which is crucial for recognition. We observed category-selective MTL cells, which have long been regarded as fundamental building blocks of semantic memory ²⁻⁵—one component of declarative memory, alongside episodic memory. Third, we agree with Dr. Orban that a significant amount of semantic processing occurs in the temporal pole ^{6,7}. However, extensive literature has also demonstrated semantic processing in MTL neurons ^{2,8-13}. Lastly, in the revised manuscript, we have ensured that all statements regarding memory (which appear in only three instances) are accurate.

3 Little difference in results between AH and amygdala is reported, responses and selectivity at iEEG and single cell level is at least as strong in Amygdala as in PH or AH (fig 4 and 5). Yet these structures are supposed to have very different behavioral functions, a finding not discussed. In fact such results support the view that the single cell selectivity is broad cast from a common input, eg FG or IT cortex.

We thank Dr. Orban for pointing this out. In this study, we analyzed three subregions of the MTL (amygdala, AH, and PH) separately. Dr. Orban is correct that responses and selectivity at both the iEEG and single-neuron levels were similar across these subregions (especially for region coding). While these MTL subregions are associated with distinct functions, numerous studies have shown similar responses across them for visual processing of faces and objects ^{2,8-13}. In our previous studies, we specifically

compared the visual coding of faces ¹ and objects ¹³ between the amygdala and hippocampus and observed qualitatively similar results in both regions. Therefore, our present findings are highly consistent with the literature.

Furthermore, we agree with Dr. Orban that the similarity between MTL subregions may arise from a common input, such as the FG and/or ITG. Notably, this possibility is supported by the similar functional connectivity of the amygdala and hippocampus with the VTC, as demonstrated in the present study (**Fig. 6; Fig. 7**).

We have included the following discussion in the revised manuscript:

“We observed comparable region coding across MTL subregions (PH, AH, and amygdala). While these regions are associated with distinct cognitive and affective functions, extensive literature has shown that they share commonalities in the visual processing of faces and objects ^{2,8-13}. Our previous studies directly comparing face ¹ and object ¹³ coding in the amygdala and hippocampus also found qualitatively similar results between these regions, making our present findings highly consistent with the literature. This similarity may stem from a common input, such as the FG and/or ITG, as supported by the similar functional connectivity of the amygdala and hippocampus with the VTC (**Fig. 6; Fig. 7**).”

4 Statistical analysis. When comparing areas with respect to number of selective/responsive leads or latencies, it is likely that leads from different patients were included with numbers varying between areas. In this case mixed effects models with patients as random variable may be more appropriate than the chi square or unreported statistical tests: see Yu et al Neuron 2022 and Tiesinga et al 2023.

We agree with Dr. Orban that the number of channels contributed by different patients should be considered in between-area comparisons. As Dr. Orban suggested, we applied a linear mixed effects model to compare the percentage of visually responsive channels and latencies across brain areas, with patient ID as a random variable. The results remained consistent (see **Revision Fig. 1** for details). We have included these control analyses in the revised manuscript:

“The results remained consistent using a linear mixed effects model where patient index was included as a random variable (Percentage ~ Region + 1|Patient; Ps < 0.05 for all comparisons between the FG and other regions).”

Revision Fig. 1. Percentage of visually responsive channels for each brain region. Asterisks indicate a significant difference in the percentage using a linear mixed effects model. *: P < 0.05, **: P < 0.01, ***: P < 0.001, and ****: P < 0.0001. FG: fusiform gyrus. ITG: inferior temporal gyrus. AH: anterior hippocampus. PH: posterior hippocampus. Each circle represents a patient, and the error bars denote ±SEM across patients.

And:

“The response onset latency in the FG (138.5 ± 32.0 ms [median ± SD]) was significantly shorter than in all other regions (ITG: 150.5 ± 21.7 ms; PH: 146.5 ± 14.9 ms; AH: 148.5 ± 15.6 ms; all Ps < 0.05; **Fig. 2e, f**), while the response onset latency in the amygdala (158.5 ± 20.5 ms) was significantly longer than in all other regions (all Ps < 0.05; **Fig. 2e, f**). The results remained consistent when using a linear mixed effects model with patient index included as a random variable (Latency ~ ROI + 1|Patient; all Ps < 0.05).”

Other shortcomings are:

5 The authors gloss over intriguing difference between axis selectivity reported in the manuscript and that described originally in monkey TE by the Tsao group: 1) in monkey 4 axes have been reported, not 2 and 2) neurons tuned for these axes are localized in different parts of TE (here no clear segregation was reported); 3) axis tuning was measured with single cells (here iEEG, and results between these methods can differ); 4) in monkey the features for which axis tuning was observed, were shape parameters, extrapolated from earlier single cell studies, here features were derived deep learning and thus undefined although a coarse correspondence with the properties of animacy and man-made/natural objects, was stated.

We thank Dr. Orban for pointing this out. First, while the number of axes used could be considered arbitrary (we selected two axes to facilitate the region coding analysis), we would like to clarify that in ¹⁴, “cells were clustered into four networks according to the first two components of their preferred axes, forming a map of object space”. Therefore, the monkey study also identified two primary axes. However, we acknowledge that these axes may represent different stimulus variations. In our study, we observed a similar animate-inanimate axis as in ¹⁴ (representing animacy and artificiality), but we observed weaker encoding of the stubby–spiky axis compared to what was reported in ¹⁴ (see our new **Supplementary Fig. 3** and **Supplementary Fig. 4**). This discrepancy is likely due to differences in the stimuli used across studies. Second, we agree with Dr. Orban that neurons tuned to different axes can be localized in distinct regions of the ITC, and we have discussed this in the revised manuscript. Third, we acknowledged the extension of axis coding to the mesoscopic iEEG level (our study) from the microscopic neuronal level in monkey studies ¹⁴. Fourth, it is worth noting that deep neural networks were employed for feature extraction and PCA was used for dimensionality reduction (¹⁴ used for raw features, while we used it for neural axes) in both cases. The primary differences in tuned axes may be attributed to the differences in stimuli, with the monkey study using simple, isolated objects, whereas our study used natural scene objects with rich backgrounds.

We have included the following discussion in the revised manuscript:

“Importantly, as one of the first studies, we revealed axis-based feature coding of objects in the human VTC (including both the FG and ITG), as previously reported in non-human primates ¹⁴⁻¹⁷. Notably, this robust axis coding was observed at the mesoscopic neural population level (in contrast to the microscopic neuronal level shown in primate studies ¹⁴), indicating that the representation of visual features may involve the coordinated activity of larger groups of neurons. The preferred axes of these neural populations represented visual changes related to variations in animacy and artificiality (Supplementary Fig. 3; Fig. 4a), consistent with primate studies ¹⁴. However, we observed weaker encoding of the stubby–spiky dimension compared to what was reported in ¹⁴ (Supplementary Fig. 3; Supplementary Fig. 4; Fig. 4a), a discrepancy likely due to differences in the stimuli used across studies (see also Supplementary Fig. 3 vs. Supplementary Fig. 4). Specifically, the primate study used simple, isolated objects, whereas our study examined natural scene objects with rich backgrounds. It is worth noting that DNNs were employed for feature extraction and PCA was used for dimensionality reduction (¹⁴ used for DNN features, while we used it for neural axes) in both cases. Furthermore, extending findings from primate studies ¹⁴, our time-resolved feature coding analysis revealed the temporal dynamics of neural coding across different brain areas. It also demonstrated the propagation of visual feature encoding along the ventral pathway, consistent with the hierarchical processing of visual objects ^{18,19}. Additionally, our recordings covered the anterior FG and ITG—areas that have been less discussed due to challenges in obtaining reliable signals ²⁰. Lastly, it is worth noting that axis-coding channels from different brain regions may encode distinct visual features, as primate neurons tuned to different axes can be localized in distinct regions of the ITC ¹⁴.”

6 methods used to derive links between areas are relatively unspecific and global, although effort were made to show they are linked to axis/ category selectivity of leads or preferred categories of neurons. Phase locking between leads not very specific (see ter Wal et al Neuroimage 2018); the much superior Granger connectivity was calculated on filtered data (<85HZ) and was restricted to lower frequency bands. Using the broad gamma band (50-150Hz) which reflect neuronal activity, Tiesinga et al 2023 obtained very specific links between particular leads in TE and TP. A more specific pattern of connection may be observed between FG and MTL with this method.

We thank Dr. Orban for pointing this out and appreciate these insightful comments. First, as Dr. Orban mentioned, we conducted specific analyses for axis-coding channels and region-coding neurons. Second, we acknowledge the concern regarding potential volume conduction effects on phase-locking value (PLV) measurements, which could lead to spurious increases in synchronization and reduced specificity. To minimize these artifacts, we implemented two procedures: (1) applying bipolar referencing to reduce common noise sources, and (2) determining statistical significance using a permutation procedure, in which trial labels were shuffled, and the observed PLV was compared to the 95th percentile of the surrogate distribution. Furthermore, multivariate Granger causality (GC) is less susceptible to this issue. As suggested by Dr. Orban, we performed GC analysis on signals that included broadband gamma (2-170 Hz). We found that bidirectional synchronization between the VTC and MTL was significant specifically at lower frequencies (<20 Hz; **Revision Fig. 2**), consistent with the results from the PLV analysis.

Revision Fig. 2. Bidirectional Granger causality (GC) between the VTC and MTL. Shaded area denotes \pm SEM across channel pairs. The dashed lines indicate the 95th percentile of the null distribution estimated through permutation. Purple: feedforward (from the VTC to the MTL) Granger causality. Cyan: feedback (from the MTL to the VTC) Granger causality. Dots on the top indicate significant Granger causality at that frequency ($P < 0.05$ across 5 consecutive frequencies after FDR correction ²¹).

7 the choice of object categories is completely arbitrary in the main and control studies.

We thank Dr. Orban for pointing this out. Our selection was not completely arbitrary; rather, it was guided by the need for a diverse and representative set of object categories that span a broad range of semantic and visual properties. Specifically, for the ImageNet stimuli, we selected 50 object categories to encompass a wide variety of natural and artificial objects, ensuring a broad representation across taxonomic (e.g., animals, plants, artifacts) and functional (e.g., tools, furniture, clothing) domains. These categories were chosen to include different levels of visual complexity (e.g., simple geometric objects like plates vs. complex natural forms like crustaceans) and structural properties (e.g., rigid objects like buildings vs. deformable objects like ferns). The selection also aimed to capture distinctions along known perceptual and semantic dimensions, such as animacy (e.g., rodents vs. vehicles) and material composition (e.g., metallic tools vs. organic fruits). For the Microsoft COCO stimuli in the control task, we selected 10 categories with 50 images each to provide a separate, well-controlled stimulus set that still maintained diversity across natural and artificial categories. The chosen categories cover a range of animate (e.g., bear, elephant, person) and inanimate (e.g., airplane, chair, car) objects, ensuring the presence of both living and non-living entities. Both datasets are widely used in computer vision research, and these categories are among the most commonly represented, providing a well-balanced set for comparison. Together, the diversity in both datasets ensures that our findings are not biased toward a specific subset of object categories and that the computational principles we identify are generalizable across different domains of object recognition. Additionally, our category choices align with previous studies that have investigated object representations in high-level visual cortex, allowing for meaningful comparisons with prior literature.

We have included the following justification in the revised manuscript:

“The choice of object categories was guided by the need for a diverse and representative set spanning a broad range of semantic and visual properties.”

And:

“These categories encompassed a wide variety of natural and artificial objects, ensuring a broad representation across taxonomic (e.g., animals, plants, artifacts) and functional (e.g., tools, furniture,

clothing) domains. We aimed to capture distinctions along known perceptual and semantic dimensions, such as animacy (e.g., rodents vs. vehicles) and material composition (e.g., metallic tools vs. organic fruits), while also including objects with different levels of visual complexity and structural properties.”

And:

“These categories were chosen to provide a well-controlled stimulus set that maintained diversity across natural and artificial categories. The selection ensures the presence of both living and non-living entities, covering a range of animate (e.g., bear, elephant, person) and inanimate (e.g., airplane, chair, car) objects.”

And:

“Together, the diversity in both datasets ensured that our findings were not biased toward a specific subset of object categories and that the computational principles we identified were generalizable across different domains of object recognition. Additionally, our category choices aligned with previous studies investigating object representations in the higher visual cortex, allowing for meaningful comparisons with prior literature.”

Other weaknesses:

8 in the result section on MTL single neurons there is too much reasoning on single cell coding (as in early H & W papers). Now we know that what matters is a population of tuned neurons which can retrieve the identity of object category;

We thank Dr. Orban for pointing this out. First, because region coding is a novel coding mechanism, we believe it is important to provide a clear introduction to its principles. Reviewer 4 also suggested that we elaborate on this aspect for clarity. Second, while we acknowledge that a population of tuned VTC neurons is critical for constructing a neural feature space and retrieving object identity, single-neuron coding remains fundamental. Individual MTL neurons can exhibit distinct selectivity and tuning properties, which contribute to the overall population dynamics. Understanding these individual properties is essential for fully characterizing the neural representation of object categories.

9 Figures unclear in places: eg explain color codes eg fig 5e; panels with large numbers of photographs of categories are impossible to read

We thank Dr. Orban for pointing this out. We have added color codes (i.e., object labels) to **Fig. 5e** and reformatted the feature spaces containing a large number of images into a color-text format to improve category legibility. Please refer to our revised **Fig. 4** and **Fig. 5**.

9 overstatements in text: eg in results section it is claimed that the iEEG data delineate responsive temporal cortex, but this requires that also non responsive areas are reported; in the discussion a topographic organization is mentioned, but it is unclear to which result this relates.

We thank Dr. Orban for pointing this out. We agree that non-responsive areas are important. To clarify, non-responsive areas were also reported in **Fig. 2b**, which depicts our full coverage. We have revised the manuscript to clarify the discussion on topographic organization:

“In this study, we first demonstrated the topographical organization of visual representations for a large set of object categories at a fine-grained scale (**Fig. 2b-d**), as illustrated in primate physiology²² and human fMRI studies^{23,24}.”

10 were any differences in latencies between subjects observed, as was reported by Tiesinga et al 2023?

We thank Dr. Orban for the question. We performed an analysis of covariance (ANCOVA) to test whether response latency differed across subjects within each brain area while controlling for the coordinates (x, y, z) of the channels ($\text{Latency} \sim \text{Patient} + x + y + z$). Indeed, visual response latencies were significantly different across subjects within the FG ($P = 1.72 \times 10^{-7}$), ITG ($P = 0.017$), amygdala ($P = 0.001$), and AH ($P = 0.002$). These results are thus consistent with 7.

Reply to comments from Reviewer 2

Here, Cao take advantage of a unique data set capturing intracranial EEG (iEEG) from the ventral temporal cortex and single unit recordings from the medial temporal lobe as participants perform a simple one-back task in which they are viewing hundreds of images drawn from different categories. The authors build on prior work in NHPs demonstrating that activity in the VTC appears to be organized along a set of coding axes, where each axis represents the projection along a weighted combination of object features. They then compare this coding to the coding of individual neurons in the MTL, and find that the neuronal spiking responses in the MTL are organized within regions of this VTC coding space. This is a highly interesting and relevant study, as it begins to address the important question of how the human brain transforms information from densely distributed coding in the temporal lobe ventral cortex to sparse spiking activity in the medial temporal lobe. The analyses are well constructed, and the data are clearly presented. There are several suggestions, however, that might help strengthen the overall claims of the study.

One of the main claims of the manuscript is that feature coding occurs in the VTC whereas region coding within that feature space occurs in the MTL. In particular, they examine how individual neurons respond to the features defined by a PCA of the VTC feature axes. However, in one of the earlier results, the authors show that indeed MTL neuronal activity (specifically the hippocampus, Fig 3e) also encode the object feature space. As such, it would be helpful to understand how the region coding within the feature space differs from the direct feature coding observed in the anterior hippocampus. What is the benefit of performing this extra manipulation.

We thank the reviewer for the expert and constructive comments.

The reviewer raised an important question. We focused on the ventral temporal cortex (VTC) to construct a neural feature space because of the vast literature indicating that this brain area encodes visual features using an axis code (i.e., it could have axes for the neural feature space). While the anterior hippocampus (AH) contained a significant number of axis-coding channels (**Fig. 3c, e**), which could lead to an organized neural feature space (**Revision Fig. 3a**) and showed a significant correlation with the VTC (i.e., fusiform gyrus [FG]) neural feature space (**Revision Fig. 3b**), we observed only 47 (5.06%, binomial $P = 0.43$) region-coding neurons within this space. This was likely because the AH neural feature space was not as organized as the VTC neural feature space, making it less effective in

meaningfully representing visual objects. The AH may carry over visual feature information from the VTC, but this information could be degraded.

Revision Fig. 3. Comparison of neural feature spaces. **(a)** Neural feature space constructed using axis-coding channels from the anterior hippocampus (AH). **(b)** Representational similarity between axis-coding channels in the fusiform gyrus (FG), inferior temporal gyrus (ITG), and AH. The similarity was calculated using Spearman’s correlation between the representational distances among neural populations in each area. Statistical significance was determined using the Mantel test, where the observed correlation was compared to a null distribution generated by shuffling the matrices’ values 1000 times. All correlations were significant (Bonferroni correction for multiple comparisons).

When examining region coding, the authors use the feature space defined only by the fusiform gyrus. Yet there are other regions of the VTC that also capture this feature space, as shown by their data. Why restrict the region coding to only the FG feature space? Moreover, in subsequent analyses examining phase locking and Granger causality, rather than just focus on the FG, they then expand their analyses to examine the entire VTC. Why would the difference between the two analyses.

We thank the reviewer for pointing this out. We used the neural feature space derived from the FG to probe region coding because it was the most robust in axis coding, and the organization of the resultant neural feature space appeared the most meaningful. However, we were able to reliably replicate our

findings using combined axis-coding channels from both the FG and ITG (i.e., the entire VTC; **Revision Fig. 4**). We have clarified this in the revised manuscript.

“Principal component analysis (PCA) was applied to these axes to create a two-dimensional neural feature space, representing the primary visual information encoded in the VTC (**Fig. 4a**; note that we used axis-coding channels in the FG for subsequent analysis because the FG exhibited the most robust axis coding, but we obtained similar results using combined axis-coding channels from both the FG and ITG).”

Revision Fig. 4. Region-based feature coding of MTL neurons in the VTC neural feature space. Neural feature space was constructed using combined fusiform gyrus (FG) and inferior temporal gyrus (ITG) axis-coding channels. **(a)** Percentage of region-coding neurons in each MTL ROI. **(b)** The number of categories encoded by region-coding neurons. **(c)** The number of objects encoded by region-coding neurons (i.e., the number of object images that fell within the tuning region of a region-coding neuron). **(d)** The population aggregated tuning region in each MTL ROI. Color bars show the counts of overlap between individual tuning regions. Numbers in the density maps show the percentage of the VTC neural feature space covered by the tuning regions of the total observed region-coding neurons.

The authors present an analysis in which they compare feature based coding in the VTC to category based coding in the MTL using representational similarity analysis. It would be helpful if they could clarify exactly how this analysis was conducted and what was compared. Reading through the methods did not clarify this point, but this seems like it would be a nice segue into subsequent analyses on region coding.

We thank the reviewer for pointing this out. We have clarified this analysis as follows (note that we have also included a new **Supplementary Fig. 5** to better illustrate the image-by-image dissimilarity for the representational similarity analysis [RSA]):

“Importantly, representational similarity analysis (RSA) of the neural pairwise distance of visual objects (see **Methods**) showed that visual objects were similarly represented in axis-coding channels in the VTC and category-coding channels in the MTL (**Fig. 3h**; see **Supplementary Fig. 5** for individual dissimilarity matrices), indicating information flow between these brain areas.”

And in **Methods**:

“For a given ROI, we constructed an image-by-image dissimilarity matrix (DM) using the pairwise Euclidean distances between object images based on the responses of the selected channels. Specifically, for each pair of images, we calculated the Euclidean distance of the neural responses to these two images across all axis-coding or category-coding channels in the ROI. The representational similarity between two ROIs was then computed by correlating the two DMs (e.g., the FG axis-coding DM and the AH category-coding DM) using Spearman correlation, which does not assume a linear relationship.”

The authors demonstrate significant phase locking in lower frequency bands between the VTC and MTL. But the analysis of Granger causality demonstrates connectivity at much higher frequencies. Could the authors provide some insight as to why the discrepancy? Moreover, when examining the locking of individual neurons to gamma phase, the authors find a subset that indeed exhibit significant phase locking. Here the reported numbers, it appears, are $n=9$, and $n=14$. Are these meaningful?

We thank the reviewer for pointing this out. The difference in frequency ranges observed between phase-locking and Granger causality likely arises from the distinct neural mechanisms captured by these two methods. Specifically, phase-locking at lower frequencies (e.g., delta, theta, and alpha bands) between the VTC and MTL suggests that these regions engage in synchronized oscillatory activity, potentially supporting integrative processes such as memory consolidation or long-range communication. Lower-frequency oscillations are known to facilitate large-scale network interactions and may reflect coordinated neural dynamics over broader temporal windows. In contrast, our Granger causality analysis revealed directed connectivity at higher frequencies (e.g., beta and gamma bands), which may be associated with fast-paced information transfer and local computations. Higher-frequency oscillations tend to be more transient and localized, supporting rapid processing of visual information and neural encoding in VTC-MTL interactions. Thus, the discrepancy between phase-locking and Granger causality frequencies likely reflects complementary aspects of VTC-MTL interactions: lower-frequency synchrony may underlie broad-scale coordination, while higher-frequency connectivity supports efficient and rapid information transfer. We have clarified this distinction in the revised manuscript.

“In this study, we observed phase-locking at lower frequencies, while Granger causality was present at higher frequencies. This difference in frequency ranges may arise from the distinct neural mechanisms captured by these two methods. Specifically, phase-locking at lower frequencies (e.g., delta, theta, and alpha bands) between the VTC and MTL suggests that these regions engage in synchronized oscillatory activity, potentially supporting integrative processes such as memory consolidation or long-range communication. In contrast, our Granger causality analysis revealed directed connectivity at higher frequencies (e.g., beta and gamma bands), which may be associated with fast-paced information transfer and local computations. Therefore, the discrepancy between phase-locking and Granger causality frequencies likely reflects complementary aspects of VTC-MTL interactions: lower-frequency synchrony may underlie broad-scale coordination, while higher-frequency connectivity supports efficient and rapid information transfer. Future studies are needed to further differentiate these neural processes.”

We apologize for the confusion regarding the small number of individual neurons showing gamma phase-locking. To clarify, these numbers represent neurons exhibiting both region coding and phase-locking. The small count is due to the fact that there were only 35 region-coding neurons out of a total of 189 neurons to begin with (notably, the percentage of region-coding neurons was consistent with our

previous studies ^{1,13}). It is important to note that among 154 non-region-coding neurons, 124 neurons were phase-locked to gamma oscillations in the VTC (80.52%; 75 phase-locked to the FG and 96 phase-locked to the ITG). This indicates a substantial population of MTL neurons exhibiting significant gamma phase-locking to the VTC. Notably, the phase distribution of region-coding neurons was similar to that of non-region-coding neurons (**Revision Fig. 5**). We have clarified these points in the revised **Results** and figure legends:

“Indeed, we found that MTL region-coding neurons fired spikes that were phase-locked to gamma oscillations in the VTC (see **Fig. 7a, b** for examples; 12/35 phase-locked to the FG and 22/35 phase-locked to the ITG; for the entire population: 87/189 phase-locked to the FG and 118/189 phase-locked to the ITG; see **Supplementary Fig. 9a-d** for analysis in the theta frequency range). Similar to the entire population, the majority of MTL region-coding neurons fired close to the flanks of the gamma frequency oscillation (i.e., 90° or 270°; **Fig. 7c**) in both the FG and ITG.”

Revision Fig. 5. Distribution of the preferred phase of MTL non-region-coding neurons that were phase-locked to VTC gamma oscillations (MTL-FG: $n = 75$; MTL-ITG: $n = 96$). The red line indicates the phase notation used.

Lastly, we would like to clarify that our previous analysis used matched in-region and out-region trials. However, in the current analysis, we included all trials, which allowed us to identify more phase-locking neurons.

On a related note, it would probably be helpful if the authors were a little cautious in the interpretation of the connectivity results. Specifically, they claim that their analyses demonstrate that VTC coding channels provide more relevant information to the hippocampus during coding. Without an explicit measure of information and a clear analyses showing how coding is tied to this connectivity, the authors may want to consider tempering this claim.

We thank the reviewer for pointing this out. We agree with the reviewer and have removed this argument in the revised manuscript. We have also revised the next sentence to be “Notably, AH region-coding neurons exhibited significantly stronger phase-locking to VTC axis-coding channels than to VTC non-axis-coding channels (Fig. 7d), suggesting that the AH was more tuned to input from the axis-coding channels in the VTC.”

Reply to comments from Reviewer 3

Summary of the Study

This study investigates the neural mechanisms underlying object representation in the human ventral temporal cortex (VTC) and medial temporal lobe (MTL) using intracranial EEG (iEEG) and single-unit recordings. The authors propose a computational framework in which the VTC encodes visual features via axis-based feature coding, which is then transformed into region-based feature coding in the MTL, supporting semantic representation. Through functional connectivity analyses, they demonstrate bidirectional interactions between these regions and provide evidence that axis-coding in the VTC is linked to semantic coding in the MTL. This work builds on prior findings in non-human primates and extends them to human recordings, offering novel insights into the transition from perceptual to semantic representations in the brain. However, several aspects of the analysis require further clarification, particularly regarding alternative coding schemes, the interpretation of functional connectivity, and the temporal dynamics of processing. Additionally, refining the presentation of figures and ensuring consistency in methodological justifications would further strengthen the paper.

Major Strengths

- The study addresses an important question in visual neuroscience: how object representations transition from perceptual encoding in the VTC to conceptual encoding in the MTL.*
- The combination of iEEG and single-unit recordings provides a unique and powerful dataset, allowing for a detailed analysis of neural coding at multiple scales.*
- The use of partial least squares (PLS) regression with deep neural network (DNN) feature representations is methodologically rigorous and aligns with recent advances in computational neuroscience.*
- Functional connectivity analyses using phase-locking value (PLV) and Granger causality (GC) strengthen the argument for dynamic interactions between the VTC and MTL.*
- The validation with an additional dataset (COCO) enhances the generalizability of the findings.*

We thank the reviewer for the expert and constructive comments. The reviewer had an accurate understanding of our results.

Major Points for Revision

1. Evidence for axis coding in the VTC

- The significance of PLS regression alone is insufficient to establish axis coding as the dominant encoding strategy in the VTC. While the regression's significance partially controls for exemplar-based coding, the relatively low R^2 values suggest that axis coding may not be the most explanatory model.*
- The authors should directly compare axis coding with other encoding strategies, particularly category-based encoding, which is mentioned at the beginning of the manuscript. It is essential to demonstrate that axis coding explains the neural responses in the VTC better than alternative models.*
- One approach would be to fit a regression model that includes both axis coding and category-based coding variables and compare their contributions. Alternatively, a correlation-based analysis or Representational Similarity Analysis (RSA) could be used to construct dissimilarity matrices based on both axis coding and category-based coding. The authors should examine whether the dissimilarity matrix derived from axis coding better aligns with the neural responses in the VTC.*
- Additionally, the demonstration of cross-dataset generalizability and consistency of the axes is a major strength of the study. However, the investigation of object space remains insufficient. Identifying the nature of the coding axes is crucial to further validate the axis-based feature coding in the VTC. Specifically, more detailed characterization of the extracted axes (e.g., how they align with known dimensions such as animate vs. inanimate) would strengthen the argument for axis-based encoding.*

We thank the reviewer for the question and suggestions. First, we apologize for an error in estimating the R^2 for the partial least squares (PLS) model, which has now been corrected in **Supplementary Fig. 2a**. Please note that although the trend remains consistent, the mean R^2 is now 0.6, indicating a good fit of the model.

Supplementary Fig. 2. (a) Goodness-of-fit (R^2) of the partial least squares (PLS) regression with deep neural network (DNN) features for each DNN layer. The layer with the highest R^2 was selected for further analysis. Error bars denote \pm SEM across channels.

Second, to determine whether axis coding or category-based coding better explains neural responses in the VTC, we compared the R^2 from the PLS model with that derived from an ANOVA applied to neural responses using categorical labels for the category-based coding model. Indeed, we found that across all visually responsive channels, the axis model explained significantly higher variance compared to the category model in both the FG (axis: 0.63 ± 0.038 [mean \pm SD]; category: 0.17 ± 0.087 ; two-tailed paired t -test: $t(200) = 91.94$, $P < 10^{-20}$) and ITG (axis: 0.62 ± 0.030 ; category: 0.12 ± 0.046 ; $t(73) = 63.34$, $P < 10^{-20}$; **Supplementary Fig. 2d**). We have included the following results in the revised manuscript:

“Notably, across all visually responsive channels, the axis model explained significantly more variance compared to the category model in both the FG (axis: 0.63 ± 0.038 [mean \pm SD]; category: 0.17 ± 0.087 ; two-tailed paired t -test: $t(200) = 91.94$, $P < 10^{-20}$) and ITG (axis: 0.62 ± 0.030 ; category: 0.12 ± 0.046 ; $t(73) = 63.34$, $P < 10^{-20}$; **Supplementary Fig. 2d**).”

Supplementary Fig. 2. (d) R^2 for the axis-coding PLS model and category-coding ANOVA model. Each circle represents a visually responsive channel, and error bars denote \pm SEM across channels. Asterisks indicate a significant difference between models using a two-tailed paired t -test. ****: $P < 0.0001$.

It is worth noting that we could not fit a single regression model that includes both axis coding and category-based coding variables to directly compare their contributions. Additionally, we would like to clarify that while representational similarity analysis (RSA) can be used to construct image-by-image dissimilarity matrices (DMs) for visual features (i.e., by correlating visual feature vectors between each pair of images) and for neural responses (i.e., by correlating neural vectors between each pair of images), it cannot be used to construct DMs based on coding schemes. Therefore, we could not use this approach to test whether the DM derived from axis coding better aligns with neural responses in the VTC. In other words, RSA can be used to examine the similarity between coding schemes or between brain areas (e.g., **Fig. 3h**), but not to compare how well different coding schemes explain neural responses.

Third, we thank the reviewer for acknowledging the cross-dataset generalizability and consistency of the axes as a major strength of our work. We also appreciate the reviewer’s suggestion. In response, we have characterized the extracted feature axes in greater detail and compared them with known object feature dimensions. The following results have been included in the revised manuscript:

“To further characterize the encoded feature axes of the axis-coding channels and compare them with known object dimensions from previous studies ¹⁴, we performed PCA on the object visual features to derive orthogonal feature dimensions (principal components [PCs]) from our stimuli. Visualization of the PCs revealed the well-known natural–artificial, spiky–stubby, and inanimate–animate dimensions (i.e., the first three PCs; **Supplementary Fig. 3a**), consistent with prior studies ¹⁴. We then examined

how the tuning axis of each axis-coding channel aligned with these feature dimensions using Pearson correlation. We found that across axis-coding brain areas (i.e., FG, ITG, and AH), the vast majority of the tuning axes were correlated with the first three feature dimensions (**Supplementary Fig. 3b**). In particular, the first PC, representing the natural–artificial dimension, was most frequently (FG: 60.00%; ITG: 56.52%; AH: 75.00%; **Supplementary Fig. 3b**) and most strongly (**Supplementary Fig. 3c**) aligned with the extracted neural axes (see **Supplementary Fig. 3d** for individual channels), followed by the third PC, representing animacy (FG: 28.24%; ITG: 21.74%; AH: 25.00%). Unlike findings from the monkey IT cortex ¹⁴, axis-coding channels in the human VTC were less likely—and less strongly—to encode the spiky–stubby dimension (the second PC; FG: 11.76%; ITG: 21.74%; AH: 0%; **Supplementary Fig. 3b, c**). These results were further replicated using a separate dataset (**Supplementary Fig. 4**; see also below).”

Supplementary Fig. 3. Summary of the encoded feature axes for the ImageNet dataset. **(a)** Visualization of the first three visual feature dimensions (principal components [PCs] of the object visual features). These PCs represent the well-known natural–artificial, spiky–stubby, and inanimate–animate dimensions. **(b)** Percentage of axis-coding channels aligned with each visual feature dimension. For each axis-coding channel, we first identified its neural tuning axis from the PLS regression. The best-aligned visual feature dimension was then determined by computing the correlation between the neural axis of a given channel and each of the visual feature dimensions. **(c)** Correlation coefficients between the neural tuning axes and visual feature dimensions. In each box plot, the central mark represents the median, box edges indicate the 25th and 75th percentiles, whiskers extend to non-outlier extremes, and circles denote outliers. **(d)** Distribution of axis-coding channels in the visual feature space, constructed using three principal visual feature dimensions. Colors represent different axis-coding brain areas. FG: fusiform gyrus. ITG: inferior temporal gyrus. AH: anterior hippocampus.

Supplementary Fig. 4. Summary of the encoded feature axes for the Microsoft COCO dataset. Legend conventions as in **Supplementary Fig. 3.**

2. Region-based feature coding in the MTL relative to alternative coding schemes

- The finding that MTL receptive fields exhibit region-based feature coding within the VTC feature space is highly interesting. However, it is necessary to evaluate whether this coding scheme provides a better explanation of the data compared to alternative models.*
- Specifically, the authors should compare region-based coding with category-based coding and localized axis coding to determine whether region-based feature coding is the most informative representation.*
- A direct comparison could involve fitting models that incorporate different coding schemes and assessing their relative explanatory power. Alternatively, the authors could perform RSA using dissimilarity matrices based on these different coding schemes to test which model best aligns with the neural responses in the MTL.*
- Clarifying this distinction would strengthen the argument that region-based coding is the preferred computational strategy for transforming VTC feature space into MTL representations.*

We thank the reviewer for raising this important question. Indeed, in our recent study ¹³, we have demonstrated the absence of axis coding in MTL neurons using the same set of object stimuli. Notably, in that work, we also showed that region-based coding in the MTL provides a more comprehensive mechanism for explaining category selectivity. Specifically, the region code does not rely on categorical membership of individual images, as long as the images share similar visual features. While images from the same category often cluster together in feature space, which can be captured by category-selective coding, images from different categories may also cluster due to shared visual features (e.g., structure, texture), which category coding alone cannot account for. The broader, feature-based representation observed in MTL neurons suggests that region-based coding serves as a more general and robust model for transforming the VTC feature space into MTL representations ¹³.

Furthermore, it is important to note that the three coding schemes cannot be incorporated into a single model, as they are constructed based on distinct principles. Axis coding can be captured using regression models, category coding is captured by ANOVA of object categories, and region coding is defined by permutation tests of spike density within a two-dimensional feature space, requiring a non-parametric approach. Additionally, as mentioned in our response to the previous question, RSA captures *similarity* between two different neural populations or other measurements (e.g., responses of units in deep neural networks). However, RSA is not applicable for *comparing* different coding schemes in our case, because the variance captured by each model is not reflected in the dissimilarity matrices (DMs).

3. Relationship between face-selective channels and axis coding in the VTC

- Previous ECoG studies (e.g., McCarthy et al. Cereb Cortex (1999) , Matsuo et al, Cereb Cortex (2015)) have demonstrated that face-selective channels are organized in patches within the VTC. Given this prior work, it is crucial to examine the relationship between face-selective channels and axis-coding channels in the current study.*
- The authors should explicitly test whether face-selective channels exhibit axis coding or whether they represent a distinct form of category-selective coding. This is particularly relevant for the fusiform face area (FFA), where category selectivity has been well-documented.*
- A key question is whether face selectivity in the human FFA, as defined in this study, corresponds to axis coding as seen in non-human primate (NHP) face patches.*
- If face-selective channels do align with axis-coding channels, this should be explicitly demonstrated with supporting analyses. Conversely, if they do not align, this should be discussed in terms of differences between category-selective and axis-based feature coding.*
- A possible approach would be to compare the locations of face-selective channels with axis-coding channels and test whether the neural responses in face-selective regions are better explained by axis coding or category-based models.*

We thank the reviewer for the suggestions. First, we would like to clarify that the current study focused on general object coding rather than face coding. We agree that it is an important question to investigate

whether face selectivity in the human FFA corresponds to axis coding, as observed in non-human primate (NHP) face patches. However, this question falls slightly outside the scope of the current study, as we did not specifically include face images. We would also like to clarify that our results did not rely on face/object selectivity, and we did not argue for face/object-selective processing or domain specificity in face/object processing. Rather, the VTC, particularly the fusiform gyrus (FG), may exhibit similar computational principles for both faces (cf. ²⁵) and non-face objects (our present study), consistent with previous macaque studies demonstrating common computational principles of axis coding for both faces ¹⁵ and objects ¹⁴.

Second, as the reviewer suggested, we conducted additional control analyses using faces. We recorded neural responses to both faces and objects from a subset of 11 patients (14 sessions), allowing us to further examine whether axis coding generalizes across object and face stimuli. We found that face-selective channels (10/57, 17.54%) were not significantly more likely to be axis-coding channels compared to the overall population (40/240, 16.67%) in the VTC (χ^2 -test: $P = 0.87$), suggesting that axis coding is not specific to face-selective channels. Additionally, we found that 29 out of 72 VTC channels (40.28%) that exhibited axis coding for object stimuli also exhibited axis coding for face stimuli. These results suggest that axis coding functions as a general coding principle for visual stimuli in the VTC, encompassing both faces and objects. Similarly, when examining the FG specifically, we found that face-selective channels (8/26, 30.77%) were not significantly more likely to be axis-coding channels compared to the overall population (36/135, 26.67%; χ^2 -test: $P = 0.67$). Moreover, 26 out of 57 FG channels (45.61%) that exhibited axis coding for object stimuli also exhibited axis coding for face stimuli.

Lastly, we further analyzed the relationship between category coding and axis coding for general objects and found that 95 out of 157 (60.5%) category-selective channels in the VTC also exhibited axis coding, suggesting that axis coding is a common representational format among category-selective channels in the VTC. Specifically in the FG, 50 out of 66 (75.76%) category-selective channels also exhibited axis coding.

We have further included the following discussion in the revised manuscript:

“Shared computational mechanisms for face and object coding in the VTC

While the present study demonstrated computational principles for object stimuli—specifically, how dense, feature-based representations in the VTC are transformed into sparse, category-based representations in the MTL—faces share similar computational principles²⁵. This aligns with previous macaque studies demonstrating shared axis coding mechanisms for both faces¹⁵ and objects¹⁴. Using data from a subset of 11 patients (14 sessions) in which both object and face stimuli were recorded (in separate sessions), our analyses revealed that axis coding in the VTC is not specific to face-selective channels (even within the FG), as face-selective sites were no more likely to exhibit axis coding than the broader channel population. Notably, a substantial proportion of channels showed axis coding for both faces and objects, indicating that axis coding generalizes across stimulus categories. This pattern held both across the VTC and specifically within the FG, where many channels that encoded objects along axes also did so for faces. Therefore, although there are nuanced differences between object and face coding—for example, the ITG was not involved in axis coding for faces²⁵ but was for objects; some channels exhibited face selectivity; and some channels exhibited axis coding exclusively for faces—similar computational mechanisms appear to underlie the encoding of both object and face stimuli. Furthermore, we found that a large majority of category-selective channels also exhibited axis coding, supporting the idea that axis coding serves as a general representational format in the higher visual cortex.”

4. Relationship between regional coding and semantic coding

- The relationship between regional coding and semantic coding is not entirely clear. While the study suggests that regional coding in the MTL represents a transition from perceptual to semantic representations, regional coding could, in principle, occur at lower levels of the visual hierarchy as well. For example, in early visual processing, simple cells combine to form complex cells, which also exhibit localized feature selectivity.*
- The definition of regional coding in this study is based on a high-dimensional feature space derived from DNNs, which emphasizes complex feature representations. However, this does not necessarily imply a direct link to semantic representations.*

- *The authors should clarify how regional coding in the MTL differs from lower-level feature encoding and why it should be considered a semantic transformation rather than an extension of perceptual processing.*
- *One possible approach would be to compare regional coding in the MTL with a measure of semantic similarity (e.g., word embeddings or conceptual clustering) to determine whether it aligns better with abstract category representations rather than purely visual feature spaces.*
- *Additionally, the theoretical framework for how regional coding bridges perceptual and semantic representations should be more explicitly described to strengthen its validity.*

We thank the reviewer for this important question. First, we apologize for not clearly stating that semantic clustering emerges within the neural feature space constructed by visual axes (**Fig. 1**). Specifically, although the neural feature space is based on visual features, it ultimately organizes information semantically. Therefore, the neural feature space—and the region coding of MTL neurons within it—translates feature-based representations in the VTC into semantic-based representations in the MTL. We have clarified this key point in the Abstract, Introduction, and in the legend of **Fig. 1**. Please also refer to our detailed responses to Reviewer 4's first two questions.

Second, while MTL region-coding neurons share a similar computational principle with early visual cortical neurons—namely, encoding a receptive field in the visual field or in a high-dimensional feature space—we have shown that they are not tuned to low-level visual features, nor do they exhibit region coding in feature spaces constructed using early DNN features ¹, which reflect early visual processing (e.g., simple and complex cells). Therefore, MTL region coding does not occur at lower levels of the visual hierarchy; rather, it emerges at higher levels where semantic clustering takes place.

Third, we further examined whether abstract category representations or purely visual features better align with region coding in the MTL using representational similarity analysis (RSA). To this end, we constructed neural, visual, and semantic representational dissimilarity matrices (RDMs). Specifically, we built a category-by-category visual RDM using DNN features (**Revision Fig. 6a**, left), which we previously used to test axis coding. Similarly, we constructed a neural RDM using responses from all region-coding neurons in the MTL (**Revision Fig. 6a**, middle). To build the semantic RDM (**Revision Fig. 6a**, right), we applied the SGPT model ²⁶, a large language model, to extract semantic embeddings

from text descriptions of each image. These text descriptions were generated using the ALBEF model based on the object images ²⁷. We found that MTL region-coding neurons exhibited greater representational similarity with semantic representations (Spearman's $\rho = 0.26$, $P = 0.0005$) than with visual representations (Spearman's $\rho = 0.16$, $P = 0.0009$), establishing a link between region coding and semantic representation in the MTL and supporting the idea that region coding serves as an intermediate mechanism for transforming visual features into semantic object representations.

In summary, given that region coding organizes representations semantically, it should be considered a semantic transformation rather than a mere extension of perceptual processing. The theoretical framework describing how region coding bridges perceptual and semantic representations has been made more explicit in the revised manuscript. Please refer to our substantially revised Abstract, Introduction, and **Fig. 1** legend.

Revision Fig. 6. Representational similarity between region coding and visual and semantic feature spaces. **(a)** Representational dissimilarity matrices (RDMs) and their correlations. **Left:** Visual RDM, built using visual features extracted from ResNet-101. **Middle:** Neural RDM, built from neural responses of region-coding neurons. **Right:** Semantic RDM, constructed using semantic features extracted from SGPT. **(b)** Permutation test.

Minor Points

1. Fig. 1: If single-neuron recordings were not performed in the VTC, the plotted circles should not be depicted in a way that suggests they represent firing rates. The figure should be adjusted to accurately reflect the type of data collected (e.g., high-gamma power from iEEG rather than spike rates).

We thank the reviewer for pointing this out. We have corrected it in the revised manuscript. In the figure legend, we now clarify that “neural response” can refer to iEEG high-gamma power (HGP), as in the present study, or to single-neuron activity, as in classical studies of axis coding in the VTC ^{14,15}.

2. Fig. 2b: The color scheme for the fusiform gyrus (FG) and inferior temporal gyrus (IT) appears to be reversed.

We thank the reviewer for pointing this out. We have corrected it in the revised manuscript.

3. Fig. 2e, f: The latency calculation may be influenced by response amplitude, as larger responses often lead to shorter latencies. This pattern appears to be present in the current data. If hierarchical processing is a key discussion point, it would be beneficial to verify whether the observed latency differences are genuinely reflective of processing order rather than differences in response magnitude. One way to address this is to normalize response amplitudes across regions (e.g., through subsampling or amplitude-matched comparisons) and then re-evaluate the latency differences. Alternatively, latency can be measured using different thresholds, such as relative change rate or peak time, to verify whether the results remain consistent.

We thank the reviewer for the suggestion and we agree with the reviewer. We have replicated the results using response peak latency, and consistent with the findings based on response onset latency, we found that response peak latency in the FG (419.5 ± 61.4 ms [median \pm SD]) was significantly shorter than in all other regions (IT: 460.5 ± 40.8 ms; PH: 472.0 ± 32.5 ms; AH: 474.5 ± 38.9 ms; amygdala: 462.5 ± 43.0 ms; all P s < 0.0001 ; **Revision Fig. 7**). We have noted this result in the revised manuscript:

“Similar results were also derived using response peak latency.”

Revision Fig. 7. Response peak latency for each ROI. Legend conventions as in **Fig. 2f**.

4. *Fig. 3d: A thin boundary line or shading, along with a corresponding label in the figure legend, would make the figure more informative.*

We thank the reviewer for the suggestion and we have now provided boundary lines for the FG and ITG in **Fig. 3d**, with labels in the figure legend.

5. *Fig. 3f: The delayed onset of axis coding in the anterior hippocampus (AH) is an interesting finding. However, it raises the question of whether this signal reflects true axis coding or whether it is confounded by region coding. Since region coding inherently involves encoding localized receptive fields in the VTC feature space, it may manifest with similar temporal characteristics as late-onset axis coding. The authors should discuss whether the observed AH activity aligns more closely with the characteristics of axis coding or region coding.*

We thank the reviewer for raising this important question. We agree that the late-onset latency of axis coding in the AH may align with region coding. In our previous study, we also demonstrated that an elevated response at the border of the feature space can drive both axis coding and region coding ¹.

However, it is important to note that we did not observe a significant number of iEEG channels exhibiting region coding in the AH, suggesting that axis coding may better account for the iEEG responses in this region. The lack of region-coding channels also prevented a direct comparison of the latencies between axis and region coding. We have acknowledged this in the revised manuscript:

“While the AH showed significant axis coding at the mesoscopic iEEG level, its late-onset latency (**Fig. 3f**) may indicate a closer alignment with region coding, as region coding inherently involves encoding localized receptive fields in the VTC feature space and may share similar temporal characteristics with late-onset axis coding. Computationally, we also demonstrated that an elevated response at the border of the feature space can drive both axis coding and region coding ¹. However, it is important to note that we did not observe a significant number of iEEG channels exhibiting region coding in the AH, suggesting that axis coding may better account for the iEEG responses in this region. Future studies are therefore needed to directly compare the onset latencies of axis and region coding in the AH and more broadly along the VTC–MTL pathway.”

6. Fig. 3g: The results in Fig. 3(g) could be influenced by response amplitude, potentially making it harder to distinguish whether the observed effects are due to differences in coding strength or simply variations in signal magnitude. To improve clarity, the authors could normalize response amplitudes across regions or examine whether the patterns remain consistent when analyzing only channels with similar response magnitudes.

We thank the reviewer for pointing this out, and we agree with reviewer that response latency could be influenced by response amplitude. To address this, we normalized the response of each channel to its maximum amplitude before calculating response latency and obtained consistent results: we observed a significant correlation between the onset latency of axis coding and the y-coordinate in MNI space within the FG (**Revision Fig. 8a**), even after controlling for individual differences in response latency (**Revision Fig. 8b**, as in **Fig. 3g**). We have included the following results in the revised manuscript:

“To control for the influence of differences in response amplitude across channels, we normalized the response of each channel to its maximum amplitude before calculating response latency and obtained similar results.”

Revision Fig. 8. Correlation between the response onset latency and MNI y -coordinates of axis-coding channels in the FG ($n = 75$). Each dot represents a channel, and the gray line represents the linear fit. Here, the temporal response of each channel was normalized to its maximum amplitude before calculating response latency. **(a)** Correlation using raw onset latency. **(b)** Correlation using adjusted onset latency. To control for individual differences in response latency, the onset latency for each channel was normalized by subtracting the latency of the channel with the highest axis coding strength in each session.

7. Fig. 3h: If possible, the original representational dissimilarity matrices used to compute the correlation values in Fig. 3h should be shown, either in the main figure or as a supplementary figure. This would help readers better understand the structure of the neural representations being compared and how axis coding in the VTC relates to category coding in the MTL. Additionally, even in statistically significant comparisons, the absolute correlation values appear to be quite low. The authors should discuss what this implies about the strength of the relationship between these representational spaces. Possible interpretations include: Neural representations in the VTC and MTL are related but remain largely distinct. The transformation from axis coding to category coding introduces substantial variability, resulting in lower similarity. Other factors, such as noise in the data or methodological limitations, may contribute to the low correlation. If the authors can provide additional insights into why these correlation values are low despite statistical significance, it would improve the clarity of the findings.

We thank the reviewer for pointing this out and the suggestions. First, we have included the representational dissimilarity matrices (DMs) in **Supplementary Fig. 5**.

Supplementary Fig. 5. Representational dissimilarity matrix for each ROI. **(a, b)** Axis-coding channels in the ventral temporal cortex (VTC). **(c-e)** Category-coding channels in the medial temporal lobe (MTL).

In the revised manuscript, we have also included the following discussion about the low correlation coefficients observed in this RSA:

“Our RSA revealed significant correlations between axis-coding channels in the VTC and category-coding channels in the MTL (Fig. 3h). The relatively low correlation values likely reflect the related yet distinct nature of neural representations in the VTC and MTL: axis coding emphasizes visual features, whereas category coding integrates higher-level semantic information. The transformation between these coding schemes is complex and nonlinear, introducing variability that reduces direct similarity. Additionally, methodological factors—such as noise, inter-subject variability, and spatial resolution—may contribute to the lower correlation values. Despite this, the statistically significant correlations indicate a meaningful relationship, supporting the role of axis coding as an intermediate step in the transition from visual to categorical representations.”

8. Fig. 4a: The current visualization is difficult to interpret due to overlapping stimuli. To improve readability, the authors should adjust the presentation to reduce visual clutter. One effective approach would be to highlight examples of stimuli located near category boundaries. This would help illustrate that while the visual features are continuous, a clear semantic boundary exists in the representational space. It is important to ensure a clear and representative example of such a case.

We thank the reviewer for the suggestion. We have revised **Fig. 4a** to improve the visualization of the object representational space. Specifically, we grouped the 50 object categories into 6 broader groups and used color coding to illustrate the semantic boundaries. We also added semantic labels for individual categories at their median coordinates to provide fine-grained category information. We believe this revised format not only shows the detailed distribution of object categories within the space but also better highlights the semantic structure and boundaries.

Fig. 4. (a) VTC neural feature space constructed using all axis-coding channels in the FG. All stimuli are shown in this space. Color coding indicates broad categories, with individual categories labeled in text.

9. Fig. 4b right lower: In permuted data, systematic patterns should typically disappear. However, a response peak is still observed here. This suggests that some structural bias (e.g., imbalance in sample size, spatial non-uniformity) may be influencing the results. Do you have any insights or considerations regarding this observation?

We thank the reviewer for pointing this out. The response peak in the permuted density map was due to the heterogeneous (non-uniform) distribution of objects in the VTC neural feature space. We have acknowledged this methodological caveat in the revised manuscript and confirmed that our procedure reliably identifies coding regions even under heterogeneous distributions:

“It is worth noting that if the encoded objects are not distributed homogeneously (or uniformly) in the neural feature space, a systematic pattern (e.g., a peak) may appear in the permuted distribution. However, we have shown that our procedure reliably identifies coding regions even under heterogeneous distributions ¹.”

10. Fig. 4e and f: The interpretation of these results is unclear. A direct comparison with VTC data would likely provide valuable context, but given that single-unit data from the VTC is not available, this may not be possible. The authors should clarify how these results support their conclusions about region-based coding in the MTL. If the authors acknowledge that the lack of direct VTC comparison limits the interpretability of these results, they should discuss this limitation in the discussion section. Additionally, explaining why these results are important for understanding the broader hypothesis of the study (e.g., how they support the concept of region-based coding in the MTL) would help readers grasp the significance of these findings.

We thank the reviewer for the suggestions. Our goal here is to provide a group-level summary to present a comprehensive encoding profile of *region-coding neurons* across MTL subregions. In other words, the purpose of this analysis is to characterize the coding properties of MTL neurons within the VTC-defined feature space, rather than to compare the coding properties of MTL and VTC directly. Furthermore, the feature space in this analysis was constructed using VTC responses, so it is not appropriate to use this

space to study VTC responses themselves. We have further elaborated on how these results support the concept of region-based coding:

“The number of object categories (Fig. 4d) and object images (Fig. 4e) covered by each tuning region indicates the size of the “receptive field” of a region-coding neuron within the VTC neural feature space. On average, these regions encompassed 10 to 15 categories and 20 to 30 images, revealing that only a subset of images from different categories fell within each tuning region. This image-level, rather than category-level, encoding suggests that region-coding neurons are tuned to continuous visual features shared across images, rather than to semantic labels.”

11. Fig. 4 and related text: The statement, "fewer region-coding channels were detected than expected by chance, indicating that region coding did not occur at the mesoscopic level", would be more informative if the exact percentage of detected region-coding channels was provided. This result raises an important methodological concern: does the absence of region coding in iEEG recordings simply reflect differences in recording resolution rather than a true absence of region coding at the mesoscopic level? Given that single-unit recordings in the MTL detected region coding, the authors should discuss whether this discrepancy arises from the nature of iEEG signals, which pool activity over larger populations of neurons, potentially masking fine-scale region-based tuning. If region coding does not manifest at the iEEG level in the MTL, this raises the question of whether iEEG recordings in the VTC might also fail to detect certain types of coding. Could this methodological limitation affect the comparisons between VTC and MTL? A more detailed discussion may be necessary to clarify these points. Providing additional context or explanations could help improve the interpretability of the results and address potential concerns.

We thank the reviewer for the comments and suggestions. We have now provided the exact percentage of detected region-coding channels in the text:

“However, fewer region-coding channels (16/511, 3.31%) were detected than expected by chance, indicating that region coding did not occur at the mesoscopic level.”

We have also included a detailed discussion of this result:

“We observed region-based coding at the microscopic single-neuron level, but not at the mesoscopic iEEG level. The absence of region coding in MTL iEEG channels may be attributed to two factors. First, region coding might occur at the single-neuron level, with neighboring neurons encoding different regions that are not adjacent in the feature space. Consequently, an iEEG channel—pooling activity from thousands of neurons—could mask fine-scale region-based tuning. Second, the current region-coding algorithm may be more sensitive to sparse single-neuron responses with high stimulus-dependent variance, whereas iEEG activity typically exhibits less variation across stimuli. These methodological limitations may also apply to VTC channels, potentially preventing the detection of certain types of coding that occur only at fine spatial scales. Therefore, direct comparisons between VTC and MTL based solely on iEEG data should be made with caution, as differences in observed coding schemes may partly reflect disparities in detectability rather than true functional divergence. Future studies employing single-neuron recordings in both regions will be essential to resolve these issues and to validate the representational formats inferred from mesoscopic signals.”

12. Fig. 5g: The results should be presented using the same scale as Fig. 4g to ensure consistency in data visualization. Aligning the scales (ranges) would make it easier for readers to directly compare the validation dataset results (Fig. 5g) with the original dataset results (Fig. 4g).

We thank the reviewer for the suggestion. We have updated **Fig. 5g** to enable a more intuitive comparison between the ImageNet and COCO datasets. We have also noted that the same scale is used as in **Fig. 4g-i**.

13. Fig. 6a: The PLV results in Fig. 6a provide important insights into the functional connectivity between the VTC and different MTL subregions. Given these findings, it is essential to carefully examine how they correspond to the representational similarity results shown in Fig. 3h. The fact that FG–AH shows coupling in the high-frequency range is particularly intriguing. This pattern suggests a unique role for AH in processing visual information received from the FG. The authors should explicitly analyze whether the strength of PLV in specific frequency bands correlates with the representational similarity results from Fig. 3h. If stronger functional connectivity (PLV) aligns with greater representational

similarity, this would provide additional support for the proposed computational framework. If stronger functional connectivity (PLV) aligns with greater representational similarity, this would provide additional support for the proposed computational framework.

We thank the reviewer for the suggestions. First, we would like to clarify that the RSA in **Fig. 3h** was conducted across channels—that is, we computed image-by-image correlations using all channels from a brain area, and then correlated the resulting image-by-image dissimilarity matrices between brain areas (see **Methods** for details). In contrast, the PLV analysis in **Fig. 6** was conducted between each pair of channels across brain areas. Therefore, we could not directly correlate the representational similarity results from **Fig. 3h** with the PLV results. However, we *qualitatively* compared the results from **Fig. 3h** with those from **Fig. 6**.

Specifically, we examined the PLV between VTC and MTL subregions across different frequency bands (**Revision Fig. 9**). Consistent with the results from **Fig. 3h**, we found that the FG showed stronger functional connectivity with MTL subregions in the lower-frequency bands (theta and alpha; see the updated results in **Fig. 6a** and additional discussion below), providing further support for the proposed computational framework. The nuanced differences may be due to the fact that PLV and RSA reflect different aspects of connectivity: PLV captures the degree to which two brain regions synchronize at specific frequencies, indicating dynamic communication and phase-based coordination, whereas RSA measures the similarity of representational geometry across stimuli, revealing whether two regions encode similar information rather than direct channel-wise synchrony. Future research is needed to directly compare PLV and RSA results to better understand how phase-based synchronization relates to representational similarity across brain regions.

Revision Fig. 9. The PLV analysis for each VTC-MTL pair across frequency bands. **(a)** All channel pairs. **(b)** VTC axis-coding channels only. Shown is the average PLV across channel pairs within each frequency band. To determine statistical significance for each VTC-MTL pair, we compared the observed PLV with the 95th percentile of the null distribution using a right-tailed paired *t*-test across channel pairs. *: $P < 0.05$, **: $P < 0.01$, ***: $P < 0.001$, and ****: $P < 0.0001$.

14. Fig. 6b–c: To better understand the role of different MTL subregions, it would be beneficial to analyze Fig. 6(b–c) by separating the results for each subregion (e.g., anterior hippocampus (AH), posterior hippocampus (PH), and amygdala). If the sample sizes are sufficient for certain subregions, those should at least be analyzed separately to provide a more fine-grained view of the connectivity differences. This is particularly important in relation to Fig. 3h, where different subregions may have shown different representational similarity with the VTC. If the PLV results (Fig. 6) and representational similarity results (Fig. 3h) align in a region-specific manner, this would provide additional support for the proposed coding transition between the VTC and MTL.

We thank the reviewer for the suggestions. First, we have included the PLV results for each MTL subregion (PH, AH, and amygdala) in the revised **Fig. 6a, b**. We apologize for omitting the PH in the previous manuscript; however, the PH did show consistent results. We have included the following results in the revised manuscript (we have also included the PH in **Fig. 6** and **Supplementary Fig. 7**):

“The difference in PLV between the VTC and PH was primarily in the theta (3-8 Hz; two-tailed two-sample t -test across channel pairs: $t(115) = 2.22$, $P = 0.029$; **Fig. 6b**) and alpha (7-13 Hz; $t(115) = 2.24$, $P = 0.027$; **Fig. 6b**) frequency ranges, the difference in PLV between the VTC and AH was primarily in the theta frequency range ($t(432) = 4.56$, $P = 6.62 \times 10^{-6}$; **Fig. 6b**), and the difference in PLV between the VTC and amygdala was primarily in the delta frequency range (1-3.5 Hz; $t(253) = 6.15$, $P = 3.05 \times 10^{-9}$; **Fig. 6b**).”

Fig. 6. (a) Phase-locking value (PLV) shows increased VTC-MTL coherence in the lower frequency range. The PLV was calculated to quantify the strength of coherence for each VTC-MTL channel pair. Shaded error bars denote \pm SEM across channel pairs. The light gray line indicates the 95th percentile of the null distribution estimated through permutation (see **Methods** for details). The numbers indicate the pairs of channels used. The PLV across channel pairs at each frequency was compared to the 95th percentile threshold of the null distribution using a right-tailed one-sample t -test. Dots on the top indicate significant PLV at that frequency ($P < 0.05$ across 5 consecutive frequencies after FDR correction²¹). **(b)** Time-resolved PLV shows greater synchronization between axis-coding VTC channels and MTL channels compared to non-axis-coding VTC channels. The PLV was normalized to the pre-trial baseline. The contours indicate areas with a significant difference (two-tailed two-sample t -test: $P < 0.05$, uncorrected). The number displayed in each plot indicates the total number of channel pairs. On each box, the central mark is the median, the edges of the box are the 25th and 75th percentiles, the whiskers extend to the most extreme data points the algorithm considers to be not outliers, and the outliers are plotted individually. Asterisks in the box plots indicate a significant difference between axis-

coding and non-axis-coding channel pairs using a two-tailed two-sample t -test. *: $P < 0.05$, and ****: $P < 0.0001$.

Second, we would like to clarify that only the ipsilateral side was included in the original **Fig. 6a** because functional connectivity is often analyzed on the ipsilateral side in many prior studies. To align with other analyses, we now include both hemispheres in the revised manuscript. Results from the ipsilateral side are now shown in **Supplementary Fig. 7** and **Supplementary Fig. 8**. Notably, it is now the PH that shows gamma-band phase-locking, in addition to lower-frequency phase-locking (**Fig. 6a**; see also **Revision Fig. 9**).

Third, as the reviewer suggested, we have provided time-resolved PLV results for each pair of VTC and MTL subregions (**Supplementary Fig. 8**; note that we did not obtain reliable correlation results as shown in **Fig. 6c**, likely due to limited sample sizes), allowing for comparison with the RSA results shown in **Fig. 3h**. We found that the FG exhibited stronger phase-locking with MTL subregions, consistent with the findings in **Fig. 3h**. We have included the following results in the revised manuscript:

“Furthermore, by separately analyzing each pair of VTC and MTL subregions (**Supplementary Fig. 8**), we found that the results were largely driven by the FG, consistent with the findings shown in **Fig. 3h**.”

And:

“In addition, we obtained similar results when using ipsilateral channels only (**Supplementary Fig. 7b, c; Supplementary Fig. 8b**) and when using an equal number of channels.”

Supplementary Fig. 8. Phase-locking across VTC-MTL subregions. **(a)** Channel pairs from both hemispheres. **(b)** Channel pairs from the ipsilateral hemisphere. Legend conventions as in **Fig. 6b**.

15. The finding that FG and IT exhibit stronger feedforward GC to AH in the $<10\text{Hz}$ range is particularly interesting, as it suggests a low-frequency mechanism for information flow from VTC to the MTL. However, this effect does not appear as prominently in Fig. 6f. This discrepancy could be due to differences in the way the analysis is conducted (e.g., spectral GC in Fig. 6d vs. time-resolved GC in Fig. 6f), or subregion-specific variations within the MTL—perhaps only AH exhibits this effect, while PH or the amygdala do not, or the effect being too weak when examined in the time domain rather than the frequency domain. To clarify this, the authors should analyze GC separately for each MTL subregion if sample sizes permit. If AH uniquely exhibits this low-frequency feedforward GC, this should be explicitly discussed, as it would suggest a distinct role for AH in integrating visual information from the VTC. Additionally, the authors should consider how these connectivity results relate to Fig. 3h, which examined representational similarity. If stronger connectivity at $<10\text{Hz}$ corresponds to higher representational similarity, this would provide additional support for the proposed functional link between the regions.

We thank the reviewer for the insightful observation and suggestions. We agree with the reviewer's interpretations regarding **Fig. 6**. We have included the following discussion in the revised manuscript:

“The FG and ITG exhibited stronger feedforward Granger causality (GC) to the AH in the low-frequency range, suggesting that low-frequency oscillations may serve as a mechanism for transmitting visual information from the VTC to the MTL during object perception. Interestingly, this effect was more pronounced in the spectral GC analysis (**Fig. 6d**) than in the time-resolved GC analysis (**Fig. 6e**). This discrepancy may reflect differences in the sensitivity of these methods—spectral GC is better suited for detecting sustained frequency-specific interactions, whereas time-resolved GC emphasizes transient, temporally localized effects. Additionally, the feedforward influence may be more specific to the AH, with weaker effects in other MTL subregions such as the PH or amygdala. These findings highlight the importance of combining multiple analytical approaches and considering subregional specificity within the MTL when examining directed functional connectivity in the human brain.”

We have now presented Granger causality (GC) results separately for each VTC–MTL subregion pair in **Fig. 6d**. As noted above, this updated analysis includes both hemispheres, whereas the original **Fig. 6d** showed only the ipsilateral side (now in **Supplementary Fig. 7d**). We further broke down the analysis by frequency band, including both low- and high-frequency ranges (**Revision Fig. 10**). Across frequency bands, FG-AH and ITG-AH pairs—particularly FG-AH in the low-frequency range—exhibited the most prominent feedforward GC (**Fig. 6d** and upper panels of **Revision Fig. 10**). This pattern aligns with the representational similarity findings in **Fig. 3h**, supporting a feedforward functional link between the VTC and AH during object perception. While the effect was more pronounced in the AH, other MTL subregions also showed significant—albeit weaker—GC with both FG and ITG. These findings suggest that although the AH may play a particularly important role in integrating visual information from the VTC, the PH and amygdala are also involved in this process.

Fig. 6. (d) Bidirectional Granger causality (GC) between the VTC and MTL. Shaded area denotes \pm SEM across channel pairs. The dashed lines indicate the 95th percentile of the null distribution estimated through permutation. Purple: feedforward (from the VTC to the MTL) GC. Cyan: feedback (from the MTL to the VTC) GC. Dots on the top indicate significant GC at that frequency ($P < 0.05$ across 5 consecutive frequencies after FDR correction²¹).

Revision Fig. 10. VTC-MTL Granger causality (GC) within specific frequency bands. Upper: feedforward GC. Bottom: feedback GC. Legend conventions as in **Revision Fig. 9.**

16. Fig. 7a, b: The authors focus on gamma oscillations (30–80 Hz) for spike-field coupling analysis, which is well-justified in the context of visual processing. However, lower frequency oscillations, particularly in the theta range (3–8 Hz), are also critically involved in memory and semantic processing. Given that Fig. 6 identified strong low-frequency coupling (especially in the <10 Hz range), it is important to investigate whether spike-field phase locking occurs at these frequencies as well. In the lower right plot of Fig. 7b, there appears to be some phase locking in low-frequency bands. This suggests that meaningful coupling may also be present outside the gamma range. The authors should expand their analysis to include theta-band phase coupling and compare its strength to gamma coupling. If strong theta-band locking is also observed, this could suggest a multi-frequency communication mechanism between the VTC and MTL, with gamma frequencies supporting feature encoding and theta frequencies supporting integration into memory/semantic networks. If theta coupling is weak or absent, this would reinforce the idea that high-frequency oscillations dominate in this particular task context. In either case, a brief discussion on why gamma was chosen and how the findings align (or contrast) with known roles of theta oscillations in MTL processing would be beneficial.

We thank the reviewer for the suggestions and agree with the reviewer's insights. In the revised manuscript, we have analyzed spike-field phase consistency in the theta frequency range (**Supplementary Fig. 9a-d**). While we found that MTL region-coding neurons also fired spikes that were phase-locked to theta oscillations (4–8 Hz) in the VTC (**Supplementary Fig. 9d**)—supporting the critical role of theta oscillations in MTL processing, as noted by the reviewer—the pairwise phase coherence (PPC) of the MTL spike-VTC iEEG pairs was not significantly greater for in-region stimuli compared to out-region stimuli in the theta range (**Supplementary Fig. 9d**). This effect was only significant in the gamma-frequency range (**Fig. 7e**), suggesting that gamma oscillations may play a dominant role in MTL neurons during feature coding of visual objects. We have clarified this point in the revised manuscript and further discussed how this finding aligns with the existing literature.

Supplementary Fig. 9. Additional results for spike–iEEG phase-locking. **(a–d)** Spike–iEEG phase-locking in the theta frequency range (4–8 Hz). Legend conventions as in **Fig. 7**. Of the MTL region-coding neurons, 15/35 were phase-locked to the FG and 20/35 to the ITG. In the full population, 102/189 were phase-locked to the FG and 114/189 to the ITG. In contrast to gamma phase-locking, theta phase-locking showed more evenly distributed phases. The pairwise phase coherence (PPC) of the MTL spike–VTC iEEG pairs was not significantly greater for in-region stimuli compared to out-region stimuli in the theta range (two-tailed paired *t*-test across spike–iEEG pairs: $t(911) = 1.73$, $P = 0.083$). n.s.: not significant. **(e)** Time-resolved PPC in the gamma frequency range for axis-coding vs. non-axis-coding channels, analyzed separately for the FG (top) and ITG (bottom).

We have included the following results:

“Indeed, we found that MTL region-coding neurons fired spikes that were phase-locked to gamma oscillations in the VTC (see **Fig. 7a, b** for examples; 12/35 phase-locked to the FG and 22/35 phase-locked to the ITG; for the entire population: 87/189 phase-locked to the FG and 118/189 phase-locked to the ITG; see **Supplementary Fig. 9a–d** for analysis in the theta frequency range).”

And:

“Furthermore, the PPC of the MTL spike–ITG iEEG pairs was significantly greater for in-region stimuli compared to out-region stimuli in the gamma frequency range (Fig. 7e: two-tailed paired t -test across spike–iEEG pairs: $t(503) = 2.01$, $P = 0.045$; a similar trend was observed for the MTL spike–FG iEEG pairs), but not in the theta frequency range (Supplementary Fig. 9d: $t(911) = -1.73$, $P = 0.083$), suggesting that objects encoded by MTL region-coding neurons had more synchronized activity, specifically with gamma oscillations in the VTC.”

We have included the following discussion:

“Our results also suggest a multi-frequency communication mechanism between the VTC and MTL, with gamma frequencies supporting feature encoding and theta frequencies facilitating integration into memory and semantic networks^{28,29}. While MTL neurons exhibited phase-locking with VTC iEEG signals in the theta frequency range (Supplementary Fig. 9a-c), the PPC of MTL spike–VTC iEEG pairs was not significantly greater for in-region stimuli compared to out-region stimuli (Supplementary Fig. 9d). This further supports the idea that gamma oscillations may play a dominant role in MTL neurons during the feature coding of visual objects. Future studies are needed to better understand this object processing network, particularly at the neural circuit level and with regard to the feedback influence of the MTL on the VTC, which may be related to processes such as stimulus imagination³⁰.”

17. Fig. 7(d): The observed spike-field coupling appears primarily after 400ms post-stimulus, which is quite late for a purely visual response. This raises an important question: how does this timing align with the broader interpretation of axis coding and VTC-MTL interactions? The main analysis window (0.1–0.6s) used elsewhere in the study does not fully capture this late effect, suggesting a potential misalignment between different analyses. One possible explanation is that the coupling observed in Fig. 7d reflects a later stage of processing related to higher-level integration (e.g., memory retrieval or semantic association) rather than initial visual encoding. Interestingly, the timing of this effect aligns well with the late-onset axis coding observed in the AH in Fig. 3f. This suggests that the coupling in Fig. 7d could be closely related to the delayed axis coding observed in the AH. The authors should explicitly discuss whether these findings indicate a multi-stage processing hierarchy, where early responses (0.1–

0.4s) are dominated by direct visual feature encoding and later responses (~400ms onward) reflect semantic or associative processing. If possible, an analysis directly comparing the time course of axis coding strength in the AH (Fig. 3f) with the timing of phase locking in Fig. 7d would help clarify this relationship. If this delayed timing is a key feature of MTL processing, it would be useful to examine whether similar late-onset effects appear in other aspects of the data, such as representational similarity or functional connectivity. Ultimately, a clearer discussion of why the observed coupling occurs so late and how it fits into the broader computational framework would improve the interpretation of these results

We thank the reviewer for the insightful questions and comments. First, we would like to clarify that the time window (0.1–0.6 s relative to stimulus onset) used for the iEEG analysis was determined based on the temporal profile of the response (**Fig. 2e**). To ensure the robustness of our findings, we replicated the main analyses using a longer time window (0–1 s relative to stimulus onset) and observed consistent results. As the reviewer noted, the MTL spike–VTC field coupling (**Fig. 7d**) occurs at a late latency (~400 ms), which aligns with the spiking profile of MTL neurons observed in our previous studies ^{1,13,31-33}. Accordingly, we applied a different time window (0.25–1.25 s relative to stimulus onset) when analyzing single-neuron activity in the current study. Therefore, the time windows applied to both the iEEG and single-neuron analyses were optimized based on their respective temporal response profiles. We have included the following discussion in the revised manuscript:

“The time window (0.1–0.6 s relative to stimulus onset) used for the iEEG analysis was selected based on the observed temporal profile of the response (**Fig. 2e**), which is consistent with the early PLV synchronization observed in **Fig. 6b**. To assess the robustness of our findings, we repeated the main analyses using an extended time window (0–1 s relative to stimulus onset) and found that the results remained consistent. In contrast, the MTL spike–VTC LFP coupling (**Fig. 7d**) emerged at a later latency (~400 ms), in line with the spiking profiles of MTL neurons reported in our previous studies ^{1,13,31-33}. Therefore, we employed a longer time window (0.25–1.25 s relative to stimulus onset) for the single-neuron analyses in the current study. Collectively, the temporal windows used for both iEEG and single-neuron analyses were selected to align with the distinct temporal dynamics of their respective signals, ensuring that the analyses were appropriately tailored to the underlying neural response profiles.”

Second, we agree with the reviewer's interpretation that "the coupling observed in Fig. 7d reflects a later stage of processing related to higher-level integration (e.g., memory retrieval or semantic association) rather than initial visual encoding," and we acknowledge its correspondence with the late onset shown in **Fig. 3f**. We have included the following discussion in the revised manuscript:

"The late onset of MTL spike–VTC field coupling (**Fig. 7d**) likely reflects a later stage of cognitive processing associated with higher-level integration, such as memory retrieval or semantic association, rather than initial visual encoding. Notably, the timing of this coupling aligns closely with the delayed axis coding observed in the AH (**Fig. 3f**), suggesting a potential link between these two neural phenomena. This temporal correspondence raises the possibility that the spike–LFP coupling observed in **Fig. 7d** is functionally related to the emergence of high-level representational coding in the AH. Taken together, these findings support the existence of a multi-stage processing hierarchy, wherein early neural responses (0.1–0.4 s post-stimulus) are primarily driven by bottom-up visual feature encoding, while later responses (from ~400 ms onward) reflect top-down, associative processes involving semantic or mnemonic integration."

Third, we would like to clarify that axis coding in the AH (**Fig. 3f**) was analyzed using AH iEEG channels, whereas the MTL spike–VTC coupling was analyzed using MTL single neurons and VTC iEEG channels. Therefore, due to the lack of a shared signal source, the MTL spike–VTC field coupling and axis coding in the AH cannot be directly compared. We agree with the reviewer that the late latency is typical for MTL neurons (see our response above; see ³⁴ for a systematic review), and that future studies will be valuable for systematically and comprehensively examining late-onset effects in visual object processing in the human MTL.

Lastly, the differences in response latency may be related to differences in oscillatory frequency (see our responses to the question above). Specifically, our results show that MTL–VTC connectivity is dominated by low-frequency oscillations at the mesoscopic iEEG level (**Fig. 6**), whereas high-frequency (gamma) oscillations play a more important role at the microscopic level (**Fig. 7**), with the latter emerging at a later latency. One possible explanation is that the MTL and VTC communicate at distinct frequencies that vary across spatiotemporal scales during object perception. The later spike–field coordination may support higher-level processes, such as transformations between coding schemes,

which require circuit-level synchronization at higher frequencies. We have included the following discussion in the revised manuscript:

“Lastly, the differences in response latency may be related to differences in oscillatory frequency. In contrast to the early onset (<100 ms) of VTC–MTL interactions observed at lower frequencies in mesoscopic iEEG signals (Fig. 6b), neurons in the MTL fired in coordination with VTC axis-coding iEEG channels at a later latency (~400 ms) and in higher-frequency ranges (Fig. 7d). The distinct temporal and spectral profiles revealed by signals from different spatial scales may reflect different stages of object perception. Further studies are needed to investigate how inter-areal synchronization varies across spatial and temporal scales to gain a comprehensive understanding of functional connectivity during object perception.”

18. Fig. 7 additional: if possible, the authors should separate the VTC into IT and FG when analyzing phase coupling, as this would allow for better alignment with the findings from Fig. 3(h) on representational similarity. Given the functional differences between FG and IT, examining them separately might reveal distinct connectivity patterns with the MTL. If IT and FG exhibit different coupling characteristics with the MTL, this would provide additional support for a hierarchical processing structure within the VTC-MTL pathway. Visualizing the stimuli that drive these responses could further strengthen the paper's conclusions. Since this study includes both iEEG and single-neuron recordings, this is highly valuable data, showcasing example stimuli that selectively engage region-coding neurons in the MTL and their corresponding VTC channels would enhance clarity. if the authors can illustrate how specific visual features or object categories are represented in both regions, this would make the interpretation of functional connectivity results more intuitive. In any case, since this is highly valuable data, illustrating the visual responses of the circuits identified from the responses could help better convey the significance of the statement, "Lastly, we investigated whether region coding in the MTL is related to visual processing in the VTC. In 9 patients (13 sessions), we recorded iEEG in the VTC and single neurons in the MTL simultaneously, which allowed us to directly test the functional connectivity between MTL region-coding neurons and VTC channels."

We thank the reviewer for the suggestions. In **Fig. 7a-c**, we presented the spike–field coupling between MTL region-coding neurons and VTC channels, separately for the FG and ITG. In the revised

manuscript, we further included time-resolved PPC analyses for the FG and ITG (see the newly added **Supplementary Fig. 9e**). Notably, FG axis-coding channels exhibited stronger synchronization with MTL region-coding neurons compared to ITG channels, indicating that the observed effects were primarily driven by FG channels. This finding aligns with the RSA results in **Fig. 3h**, which showed greater representational similarity between the FG and MTL. We have incorporated this result into the revised manuscript:

“We further demonstrated that this effect was primarily driven by the FG (**Supplementary Fig. 9e**), which not only aligns well with the VTC–MTL representational similarity profiles (**Fig. 3h**) but also further supports a hierarchical processing structure within the VTC–MTL pathway.”

Supplementary Fig. 9. (e) Time-resolved PPC in the gamma frequency range for axis-coding vs. non-axis-coding channels, analyzed separately for the FG (top) and ITG (bottom).

Furthermore, we appreciate the reviewer’s insightful suggestions regarding visualization. In response, we have visualized the stimuli encoded by MTL region-coding neurons along the most-preferred feature axis of their corresponding phase-locked VTC axis-coding channels (**Fig. 7f, g**). Interestingly, for a given region-coding neuron, the in-region stimuli tend to cluster along a specific point on the tuning axis of the paired iEEG channel, suggesting that MTL region-coding neurons encode stimuli with shared

visual features that are parametrically represented in upstream VTC regions. We have included the following results in the revised manuscript:

“To illustrate the correspondence between region coding and axis coding, we visualized the in-region stimuli of MTL region-coding neurons along the most-preferred axis of their corresponding phase-locked VTC axis-coding channels (Fig. 7f, g). Interestingly, the in-region stimuli tended to cluster at a specific point along the tuning axis of the phase-locked iEEG channels. This finding suggests that MTL region-coding neurons are tuned to visual features that are parametrically encoded by phase-locked upstream VTC areas, providing a physiological basis for functional coupling and supporting our proposed computational framework.”

Fig. 7. (f, g) Illustration of two representative MTL region-coding neurons and their corresponding phase-locked VTC axis-coding channels. (left) The z-scored high-gamma power (HGP) changed as a linear function of the first partial least squares (PLS) component of the feature map. Each dot represents an object image and color coding denotes the object category. The gray line represents the linear fit.

Larger dots correspond to images encoded by the example region-coding neuron. (right) Visualization of the encoded feature axis. Images encoded by the example region-coding neuron are highlighted with a red frame and shown at a larger size.

Reply to comments from Reviewer 4

In the present study, Cao et al. conducted intracranial EEG (iEEG) recordings together with single-neuronal recordings in the ventral temporal cortex (VTC) and the medial temporal lobe (MTL) in epileptic patients observing large number of visual objects. The authors analyzed neuronal responses to the presentation of visual objects using DNN-based visual feature extraction, as well as principal component analysis, and found that the VTC exhibits axis-based feature encoding, and that the MTL neurons encode receptive fields within the VTC neural feature space. They also conducted functional connectivity analysis with Granger causality and spike-LFP coherence analysis revealing bidirectional interaction between the VTC and the MTL.

The authors have accomplished a lot of work, the experiments and analysis have been performed properly, and the results are reliable and important for understanding the neuronal computational mechanisms underlying object recognition in humans. However, more comprehensive description and clarification of conceptual questions are needed before the paper can be accepted for publication.

We thank the reviewer for the expert and constructive comments.

Introduction does not clearly explain the computational framework of the study, which prevents it from conveying the importance of the study. The second paragraph states the key questions and hypotheses of the study. However, they are very difficult to understand, especially for the general reader outside the field of cognitive neuroscience. In particular, it will be difficult to understand the meaning and relationships among terms such as axis-based coding, neural axis, region-based coding, sparse coding, dense coding, receptive field, and neural feature space. The third paragraph and Fig. 1 are also difficult to understand and confusing. It is difficult to understand how the neural feature space is constructed from axis coding in the VTC. Also, it is difficult to understand the relationships between the feature dimension (Fig. 1, left panel) and the neural axes (Fig. 1, right panel). This is because Introduction and Fig.1 omitted that principal component analysis was used to construct the neuronal feature space from the VTC neural axes. This process is not explained until the Results and Methods section, which can lead to misunderstanding and confusion for the readers. A sentence in Fig. 1 legend, “MTL neurons receive processed visual input from the VTC and encode a receptive field (i.e., encoding region) within

the neural feature space...”. Readers may have difficulty understanding what is a receptive field within the neural feature space because the term receptive field usually means a receptive field in the visual field. Overall, the authors should expand and rewrite Introduction, as well as improve Fig.1 and its legends.

We thank the reviewer for the suggestions. We have followed the suggestions and rewritten the Introduction as follows:

“Object recognition is primarily supported by the ventral visual pathway, which spans the ventral temporal cortex (VTC) and extends into the medial temporal lobe (MTL), including the amygdala and hippocampus ^{35,36}. Within this pathway, neurons encode objects at different levels of abstraction, transitioning from lower-level, feature-based representations in the VTC to higher-level, semantic and conceptual representations in the MTL. Specifically, the VTC represents objects using axis-based coding, where neural responses are organized along key feature dimensions such as shape, texture, and curvature ^{37,38}. Each neural axis represents a fundamental direction in this high-dimensional feature space, capturing variations in object appearance. Non-human primate studies ¹⁴⁻¹⁷ have shown that axis-based coding enables efficient and flexible object representation, allowing different objects to be distinguished based on their position along multiple neural axes. Axis-based coding in the VTC results in dense coding, meaning many neurons participate in encoding the fine-grained details of an object’s visual features. This requires a broad and distributed population of neurons ³⁹⁻⁴², with each neuron responding to multiple objects that share specific visual features. In contrast, the MTL employs sparse coding, where only a small subset of neurons respond selectively to specific object categories ^{2,4,43,44}, abstracting away lower-level visual details ^{2,4}. The transformation from dense, feature-based coding in the VTC to sparse, category-based coding in the MTL is critical for efficient memory storage and retrieval. However, the precise neural computational mechanisms underlying this transformation remain unclear.

To address this question, our recent work suggests that MTL neurons encode “receptive fields” (i.e., coding regions) within a visual feature space, which is constructed using deep neural network (DNN) features (rather than neural responses), for faces ¹ and objects ¹³—a phenomenon we refer to as region-based coding. Rather than tracking individual visual features, these neurons selectively respond to specific regions (i.e., receptive fields) within the visual feature space that contain objects with similar

visual features. Critically, region-based coding can provide a potential link between visual feature processing in the VTC and semantic representations in the MTL (Fig. 1). A neural feature space can be constructed from VTC neural axes using principal component analysis (PCA). Notably, the dimensions of this space reflect principal variations in object categories (e.g., ranging from natural to artificial objects or from animate to inanimate objects); and within this space, objects with similar high-level abstract features cluster together, demonstrating semantic-based clustering that reflects their perceptual and conceptual relationships. If MTL neurons exhibit receptive fields (i.e., coding regions) within this neural feature space and respond to stimuli that fall into these regions and share similar semantics, they will demonstrate semantic coding. Therefore, this neural computational framework can explain how the brain translates feature-based representations in the VTC into semantic-based representations in the MTL.

To test this hypothesis, we simultaneously recorded microscopic single-neuron activity in the human MTL and mesoscopic intracranial electroencephalography (iEEG) in both the VTC and MTL while neurosurgical patients viewed 500 naturalistic object images spanning diverse categories. Specifically, we hypothesized that (1) the human VTC exhibits axis coding of visual features similar to that observed in non-human primates^{14,45} (i.e., iEEG responses parametrically vary as a function of DNN features; Fig. 1 bottom left), (2) VTC neural axes collectively form a neural feature space after dimensionality reduction using PCA (Fig. 1 bottom middle), and (3) the human MTL exhibits region coding within the VTC neural feature space (i.e., MTL neurons selectively respond to objects that occupy similar positions within this space; Fig. 1 bottom right).”

We have revised **Fig. 1** and its legend as follows to incorporate this important information:

Fig. 1. A neural computational framework explains the transition from dense, feature-based coding in the ventral temporal cortex (VTC) to sparse, semantic-based coding in the medial temporal lobe (MTL). Neurons and iEEG channels in the VTC (note that here neural response can refer to iEEG high-gamma power [HGP], as in the present study, or to single-neuron activity, as in classical studies of axis coding in the VTC ^{14,15}) encode visual features (e.g., deep neural network [DNN] features) using an axis-based code, where neural coding axes are organized along visual feature dimensions (i.e., parametrically varying as a function of visual features). Axis-based coding in the VTC results in dense coding, requiring a broad and distributed population of neurons. A neural feature space can be constructed using principal component analysis (PCA) of all neural axes from a brain area (e.g., VTC), where the

dimensions of this space represent principal variations in object categories (e.g., ranging from natural to artificial objects or from animate to inanimate objects). Notably, within this space, objects with similar high-level abstract features cluster together, demonstrating semantic-based clustering that reflects their perceptual and conceptual relationships. MTL neurons receive processed visual input from the VTC and encode receptive fields (i.e., coding regions) within this neural feature space, exhibiting region-based coding. Traditionally, a receptive field refers to the portion of the visual field that a neuron responds to. Here, we extend this concept by defining a receptive field within the neural feature space—representing the range of visual feature values that elicit a response from an MTL neuron. Instead of responding to a specific location in the physical visual field, these neurons respond to objects that occupy similar positions in the neural feature space. By responding to stimuli that fall within these regions and share similar semantics, MTL neurons demonstrate semantic coding, providing a crucial link between feature-based visual processing in the VTC and semantic-based representations in the MTL. Together, this computational framework explains how the brain transforms detailed visual feature representations into abstract, concept-driven encoding. Each color represents a different object category.

In Abstract, the authors stated: “The VTC exhibited axis-based feature coding, and by constructing a neural feature space using VTC neural axes, we observed that MTL neurons encode a receptive field within the VTC neural feature space. This computational framework explains how dense feature-based representations in the VTC are translated into sparse semantic-based representations in the MTL.” These are the central finding and implication of the study. However, since the neural feature space is composed of features based on the ‘visual’ axis, the finding that MTL neurons encode receptive fields in the neural feature space alone is unlikely to explain how representations are converted to representations based on ‘semantics’. The authors should clarify the logic supporting the claim that this computational framework explains how feature-based representations in the VTC are translated into semantic-based representations in the MTL.

We thank the reviewer for raising this important question and apologize for not including the critical information that semantic clustering emerges in the neural feature space constructed by visual axes. In other words, although the neural feature space is constructed using visual axes, it ultimately represents semantics. Therefore, the neural feature space and the region coding of MTL neurons within it translate

feature-based representations in the VTC into semantic-based representations in the MTL. We have clarified this in the revised Abstract:

“The VTC exhibited axis-based feature coding, and a neural feature space could be constructed using VTC neural axes, which revealed clustering of visual objects based on semantics. Importantly, MTL neurons encoded receptive fields within the VTC neural feature space, demonstrating semantic-based coding. This computational framework, therefore, explains how dense feature-based representations in the VTC are translated into sparse, semantic-based representations in the MTL.”

We have also provided a detailed explanation of this important information in the Introduction and the legend of **Fig. 1**. Please refer to our response to the question above.

Results

In Figure 3c, the PH did not show an above-chance rate of axial feature coding, while the AH showed an above-chance rate of axial feature coding bilaterally. This may suggest the presence of a posterior-anterior gradient of axis-based feature coding in the hippocampus. Do the authors have any thoughts on this finding?

We thank the reviewer for this insightful observation. The finding that the PH did not show an above-chance rate of axis-based feature coding, while the AH did bilaterally, is indeed suggestive of a posterior-anterior gradient of axis-based feature coding in the hippocampus. This aligns with previous studies showing functional specialization along the longitudinal axis of the hippocampus, where posterior regions are more involved in spatial processing, and anterior regions are more engaged in abstract and higher-order representations, including semantic and conceptual information. Given that axis-based feature coding may support the transformation of visual feature representations into higher-level representations, it is plausible that this transformation becomes more prominent in the AH. Future work could further investigate this gradient by examining whether axis-based feature coding gradually increases along the hippocampal axis and how it relates to functional differences between anterior and posterior hippocampal regions.

References

- 1 Cao, R. *et al.* Feature-based encoding of face identity by single neurons in the human amygdala and hippocampus. *Nature Human Behaviour* (2025).
- 2 Quian Quiroga, R., Reddy, L., Kreiman, G., Koch, C. & Fried, I. Invariant visual representation by single neurons in the human brain. *Nature* **435**, 1102-1107 (2005). https://doi.org/http://www.nature.com/nature/journal/v435/n7045/supinfo/nature03687_S1.html
- 3 Quian Quiroga, R., Kreiman, G., Koch, C. & Fried, I. Sparse but not ‘Grandmother-cell’ coding in the medial temporal lobe. *Trends in Cognitive Sciences* **12**, 87-91 (2008). <https://doi.org/https://doi.org/10.1016/j.tics.2007.12.003>
- 4 Quian Quiroga, R. Concept cells: the building blocks of declarative memory functions. *Nature Reviews Neuroscience* **13**, 587 (2012). <https://doi.org/10.1038/nrn3251>
- 5 Rutishauser, U., Reddy, L., Mormann, F. & Sarnthein, J. The Architecture of Human Memory: Insights from Human Single-Neuron Recordings. *The Journal of Neuroscience* **41**, 883 (2021). <https://doi.org/10.1523/JNEUROSCI.1648-20.2020>
- 6 Lambon Ralph, M. A., Jefferies, E., Patterson, K. & Rogers, T. T. The neural and computational bases of semantic cognition. *Nature Reviews Neuroscience* **18**, 42-55 (2017). <https://doi.org/10.1038/nrn.2016.150>
- 7 Tiesinga, P. *et al.* Uncovering the fast, directional signal flow through the human temporal pole during semantic processing. *Scientific Reports* **13**, 6831 (2023). <https://doi.org/10.1038/s41598-023-33318-5>
- 8 Kreiman, G., Koch, C. & Fried, I. Category-specific visual responses of single neurons in the human medial temporal lobe. *Nat Neurosci* **3**, 946-953 (2000).
- 9 Rutishauser, U. *et al.* Representation of retrieval confidence by single neurons in the human medial temporal lobe. *Nat Neurosci* **18**, 1041-1050 (2015).
- 10 Rey, H. G. *et al.* Encoding of long-term associations through neural unitization in the human medial temporal lobe. *Nature Communications* **9**, 4372 (2018). <https://doi.org/10.1038/s41467-018-06870-2>
- 11 Wang, S., Mamelak, A. N., Adolphs, R. & Rutishauser, U. Encoding of Target Detection during Visual Search by Single Neurons in the Human Brain. *Current Biology* **28**, 2058-2069.e2054 (2018). <https://doi.org/https://doi.org/10.1016/j.cub.2018.04.092>
- 12 Reber, T. P. *et al.* Representation of abstract semantic knowledge in populations of human single neurons in the medial temporal lobe. *PLOS Biology* **17**, e3000290 (2019). <https://doi.org/10.1371/journal.pbio.3000290>
- 13 Cao, R. *et al.* A neuronal code for object representation and memory in the human amygdala and hippocampus. *Nature Communications* **16**, 1510 (2025). <https://doi.org/10.1038/s41467-025-56793-y>
- 14 Bao, P., She, L., McGill, M. & Tsao, D. Y. A map of object space in primate inferotemporal cortex. *Nature* **583**, 103-108 (2020). <https://doi.org/10.1038/s41586-020-2350-5>
- 15 Chang, L. & Tsao, D. Y. The Code for Facial Identity in the Primate Brain. *Cell* **169**, 1013-1028.e1014 (2017). <https://doi.org/10.1016/j.cell.2017.05.011>
- 16 Bashivan, P., Kar, K. & DiCarlo, J. J. Neural population control via deep image synthesis. *Science* **364**, eaav9436 (2019). <https://doi.org/10.1126/science.aav9436>

- 17 Ponce, C. R. *et al.* Evolving Images for Visual Neurons Using a Deep Generative Network Reveals Coding Principles and Neuronal Preferences. *Cell* **177**, 999-1009.e1010 (2019). <https://doi.org/10.1016/j.cell.2019.04.005>
- 18 Riesenhuber, M. & Poggio, T. Hierarchical models of object recognition in cortex. *Nature Neuroscience* **2**, 1019-1025 (1999). <https://doi.org/10.1038/14819>
- 19 Lerner, Y., Hendler, T., Ben-Bashat, D., Harel, M. & Malach, R. A Hierarchical Axis of Object Processing Stages in the Human Visual Cortex. *Cerebral Cortex* **11**, 287-297 (2001). <https://doi.org/10.1093/cercor/11.4.287>
- 20 Rossion, B., Jacques, C. & Jonas, J. The anterior fusiform gyrus: The ghost in the cortical face machine. *Neuroscience and biobehavioral reviews* **158**, 105535 (2024). <https://doi.org/10.1016/j.neubiorev.2024.105535>
- 21 Benjamini, Y. & Hochberg, Y. Controlling the False Discovery Rate: A Practical and Powerful Approach to Multiple Testing. *Journal of the Royal Statistical Society. Series B (Methodological)* **57**, 289-300 (1995).
- 22 Yao, M. *et al.* High-dimensional topographic organization of visual features in the primate temporal lobe. *Nature Communications* **14**, 5931 (2023). <https://doi.org/10.1038/s41467-023-41584-0>
- 23 Grill-Spector, K., Sayres, R. & Ress, D. High-resolution imaging reveals highly selective nonface clusters in the fusiform face area. *Nature Neuroscience* **9**, 1177-1185 (2006). <https://doi.org/10.1038/nn1745>
- 24 Grill-Spector, K. & Weiner, K. S. The functional architecture of the ventral temporal cortex and its role in categorization. *Nat Rev Neurosci* **15**, 536-548 (2014). <https://doi.org/10.1038/nrn3747>
- 25 Cao, R. *et al.* A neural computational framework for face processing in the human temporal lobe. *Current Biology* **35**, 1765-1778.e1766 (2025). <https://doi.org/https://doi.org/10.1016/j.cub.2025.02.063>
- 26 Muennighoff, N. SGPT: GPT Sentence Embeddings for Semantic Search. *CoRR* **abs/2202.08904** (2022).
- 27 Junnan Li *et al.* in *NeurIPS* (2021).
- 28 Herweg, N. A., Solomon, E. A. & Kahana, M. J. Theta Oscillations in Human Memory. *Trends in Cognitive Sciences* **24**, 208-227 (2020). <https://doi.org/10.1016/j.tics.2019.12.006>
- 29 Seger, S. E., Kriegel, J. L. S., Lega, B. C. & Ekstrom, A. D. Memory-related processing is the primary driver of human hippocampal theta oscillations. *Neuron* **111**, 3119-3130 e3114 (2023). <https://doi.org/10.1016/j.neuron.2023.06.015>
- 30 Wadia, V. S. *et al.* A shared code for perceiving and imagining objects in human ventral temporal cortex. *bioRxiv*, 2024.2010.2005.616828 (2024). <https://doi.org/10.1101/2024.10.05.616828>
- 31 Cao, R., Lin, C., Brandmeir, N. J. & Wang, S. A human single-neuron dataset for face perception. *Scientific Data* **9**, 365 (2022). <https://doi.org/10.1038/s41597-022-01482-4>
- 32 Cao, R. *et al.* Neural mechanisms of face familiarity and learning in the human amygdala and hippocampus. *Cell Reports* **43**, 113520 (2024). <https://doi.org/10.1016/j.celrep.2023.113520>
- 33 Cao, R., Brunner, P., Brandmeir, N. J., Willie, J. T. & Wang, S. A human single-neuron dataset for object recognition. *Scientific Data* **12**, 79 (2025). <https://doi.org/10.1038/s41597-024-04265-1>
- 34 Mormann, F. *et al.* Latency and Selectivity of Single Neurons Indicate Hierarchical Processing in the Human Medial Temporal Lobe. *The Journal of Neuroscience* **28**, 8865-8872 (2008). <https://doi.org/10.1523/jneurosci.1640-08.2008>

- 35 Kravitz, D. J., Saleem, K. S., Baker, C. I., Ungerleider, L. G. & Mishkin, M. The ventral visual pathway: an expanded neural framework for the processing of object quality. *Trends in cognitive sciences* **17**, 26-49 (2013). <https://doi.org/10.1016/j.tics.2012.10.011>
- 36 Duchaine, B. & Yovel, G. A Revised Neural Framework for Face Processing. *Annu Rev Vis Sci* **1**, 393-416 (2015). <https://doi.org/10.1146/annurev-vision-082114-035518>
- 37 Loffler, G., Yourganov, G., Wilkinson, F. & Wilson, H. R. fMRI evidence for the neural representation of faces. *Nat Neurosci* **8**, 1386-1391 (2005).
- 38 Cao, R., Li, X., Todorov, A. & Wang, S. A Flexible Neural Representation of Faces in the Human Brain. *Cerebral Cortex Communications* **1**, tgaa055 (2020). <https://doi.org/10.1093/texcom/tgaa055>
- 39 Freeman, W. J. *Mass action in the nervous system*. Vol. 2004 (Citeseer, 1975).
- 40 Hinton, G. E. Distributed representations. (1984).
- 41 Rolls, E. T., Treves, A. & Tovee, M. J. The representational capacity of the distributed encoding of information provided by populations of neurons in primate temporal visual cortex. *Experimental Brain Research* **114**, 149-162 (1997).
- 42 Churchland, P. S. & Sejnowski, T. J. *The computational brain*. (MIT press, 2016).
- 43 Barlow, H. B. Single Units and Sensation: A Neuron Doctrine for Perceptual Psychology? *Perception* **1**, 371-394 (1972). <https://doi.org/10.1068/p010371>
- 44 Valentine, T. A unified account of the effects of distinctiveness, inversion, and race in face recognition. *The Quarterly Journal of Experimental Psychology Section A* **43**, 161-204 (1991). <https://doi.org/10.1080/14640749108400966>
- 45 Yamins, D. L. K. *et al.* Performance-optimized hierarchical models predict neural responses in higher visual cortex. *Proceedings of the National Academy of Sciences* **111**, 8619 (2014). <https://doi.org/10.1073/pnas.1403112111>

Reply to comments from Reviewer 1

The authors made extensive revisions to the manuscript to accommodate the comments of all 4 reviewers.

With respect to my comments (reviewer 1), the changes made to address my points 3 to 6, 8 to 10 are fine, those related to points 1 and 3 go in the right direction but could be further improved. Sadly, one of my main comments, my point 2, related to the claim that a semantic-based coding emerges in MTL, was made worse by the changes made to address the comments of reviewer 4.

Once again, we thank the reviewer for the expert and constructive comments.

Point 1; the authors provide some evidence, but from another species, which is only a suggestion in the present context, as the authors admit in the rebuttal. Hence, this limitation should be clearly indicated in the discussion.

We thank the reviewer for the suggestion. We have included the following discussion in the revised manuscript:

“Another important question concerns whether region-based coding in feature space, as observed in the MTL, might originate earlier in the ventral visual stream—specifically in regions such as the FG or ITG—and simply be inherited by the MTL. While our model (**Fig. 1**) posits that such region-based coding emerges in the MTL, the current study does not include single-neuron recordings from the FG or ITG and therefore cannot directly rule out this possibility. However, we addressed this issue in a separate experiment using the same paradigm (restricted to face stimuli), in which we recorded multi-unit activity (MUA) from the inferotemporal cortex (ITC) of a macaque monkey^{1,2}. The ITC is a well-established homolog of the human VTC, encompassing both the FG and ITG. Applying the same computational framework to assess the presence of axis and region coding in this dataset, we found robust and prevalent axis coding across all 53 MUA channels (100%; strength of axis coding = 0.48 ± 0.13 [mean \pm SD]), while region coding was observed in a significantly smaller fraction of neurons (30.2%; χ^2 -test: $P = 4.73 \times 10^{-14}$), and this effect was likely secondary to the axis structure^{1,2}. These findings suggest that axis-based representations dominate in the ITC/VTC, while region-based coding emerges more

prominently in the MTL. Therefore, it is unlikely that region coding observed in the MTL is simply inherited from the VTC. Instead, our results support a transformation in representational format between the VTC and MTL.”

Point 7: there still is a confusion between animate/inanimate and natural/artificial as the following sentence taken from the rebuttal clearly shows: ‘The selection ensures the presence of both living and non-living entities, covering a range of animate (e.g., bear, elephant, person) and inanimate (e.g., airplane, chair, car) objects.’ In their reply to reviewer 3 the authors made a better job of separating these two dimensions, and should make sure this done throughout the manuscript.

A personal note for the authors concerning this choice of stimuli. In my own vision works (especially in the videos made of natural actions), I made sure only natural objects and surroundings were shown. My argument was that artificial objects were around only for some 100 years and the brain could not have adapted to process these novel objects. The present data show that this not true and that the fast learning hippocampal system is able to represent these objects. Maybe this underscores a remarkable strength of the human brain.

We thank the reviewer for pointing this out. First, we would like to clarify the distinction between the animate/inanimate and natural/artificial dimensions. Although these dimensions are often correlated in everyday experience, they reflect different properties and may be represented separately in the brain. The animate/inanimate dimension refers to the biological status of an object—whether it is alive and capable of goal-directed movement (e.g., humans, animals) or not (e.g., tools, buildings). In contrast, the natural/artificial dimension reflects the origin of the object—whether it occurs naturally in the environment (e.g., trees, rocks, rivers) or is human-made (e.g., cars, computers, furniture). Notably, these dimensions do not map perfectly onto each other. For example, animals are both animate and natural, whereas tools are inanimate and artificial; however, a tree is natural but inanimate, and a robot may appear animate yet is artificial.

We have revised our description in **Methods** as follows:

“These categories were selected to sample broadly across multiple conceptual dimensions, including animacy (e.g., animate: dog, frog, rodent; inanimate: car, plate, tool) and naturalness (e.g., natural: tree

bark, fruit, rock; artificial: electronic device, musical instrument, clothing). By including both animate and inanimate stimuli as well as natural and manmade objects, the stimulus set allowed us to disentangle these two key but often confounded dimensions of object representation (Supplementary Fig. 3). In addition, the categories varied in terms of taxonomic class (e.g., animals, plants, artifacts), functional role (e.g., tools, furniture, clothing), and perceptual properties (e.g., shape complexity, texture, material composition). Importantly, these dimensions align with known object dimensions reported in prior studies ^{3,4}, including animacy vs. inanimacy, natural vs. artificial, and shape-based features such as spikiness versus stubby/rounded forms. This alignment provides a principled basis for interpreting neural responses along perceptually and conceptually meaningful axes (see Supplementary Fig. 3), as established in previous work on object representation in the brain ^{3,4}.”

And:

“For the Microsoft COCO stimuli ⁵, we selected 10 object categories, with 50 images per category. The selected categories were: airplane, apple, bear, bird, car, chair, dog, elephant, person, and zebra. These categories were chosen to form a well-controlled stimulus set that maintained diversity across both animate and inanimate as well as natural and artificial dimensions. Specifically, the set includes animate natural entities (e.g., bear, elephant, person, zebra), inanimate artificial objects (e.g., airplane, chair, car), and inanimate natural objects (e.g., apple). This selection again allowed us to sample from key conceptual and perceptual dimensions known to shape object representation in the brain (see Supplementary Fig. 4) ^{3,4}. By incorporating both categorical and perceptual variation, the stimulus set supports analyses of how neural activity reflects structured dimensions of real-world object space.”

We have also clarified this in **Introduction, Results, and Discussion**.

In addition, we appreciate the reviewer’s insightful comment regarding the use of artificial objects in visual stimuli and the broader implications for how the brain represents novel elements in the environment. While artificial objects are indeed relatively recent in evolutionary terms, our findings suggest that the human brain—particularly the hippocampal system—is capable of rapidly forming structured representations of both natural and artificial categories. This aligns with the notion that the hippocampus supports flexible, experience-dependent learning, allowing it to accommodate the representational demands of culturally novel but behaviorally relevant stimuli. In this light, the presence

of neural encoding for artificial objects may highlight a remarkable strength of the human brain: its capacity to generalize across object types and learn efficiently in the face of rapid environmental change.

This leaves point 2, where the authors made no change or even emphasized even more (in their reply to reviewer 4) that hippocampus builds a semantic based representation from the input provided by VTC. This flies in direct contradiction to the extremely well established neuropsychological data indicating a clear dissociation between the temporal pole, devoted to semantic memory, and hippocampus, subserving episodic memory of facts and personal experiences (see eg Quiroga et al Nat NS review 2015).

We thank the reviewer for further comments and thoughtful discussion. First, we believe the reviewer is referring to Quiroga's *Nature Reviews Neuroscience* paper (*Concept cells: the building blocks of declarative memory functions*), published in 2012 rather than 2015. In this review, Quiroga explicitly argues that concept cells in the human hippocampus encode semantic representations, proposing that these representations constitute the "building blocks for declarative memory functions". Notably, the review does not argue for a strict dissociation between the temporal pole and the hippocampus, nor does it mention the temporal pole or anterior temporal lobe at all. While the temporal pole has been implicated in semantic memory—defined as knowledge about the world—this does not preclude a role for the hippocampus in encoding semantic representations, particularly when such representations are integrated with contextual or associative information.

Second, while some studies showed that lesions to the hippocampus may spare basic semantic memory (but see ⁶ for a classic study showing that the hippocampus supports semantic memory), it does not rule out hippocampal involvement in richer, flexible, or relational aspects of semantic representations, especially those relevant to tasks that require integration across concepts, contexts, or episodic encoding. As Quiroga noted, "concept cells can be seen as representing semantic memories—which are also encoded in the neocortex but perhaps in a more distributed manner—and such semantic representations may be crucial for memory functions, such as generating new associations and episodic memories". This view aligns with a growing body of literature that suggests semantic and episodic memory may not be fully separable at the neural level, particularly in the hippocampus and broader MTL. Our interpretation, that the hippocampus builds structured, semantic/concept-like representations based on input from high-

level visual areas (such as the VTC), is consistent with this integrative perspective and highlights the hippocampus's role in supporting flexible, abstract representations necessary for both semantic generalization and episodic construction.

There is nothing in the data provided in the manuscript to indicate that the novel representation which emerges in HP is semantic-based; the main argument of the authors is that the representation is not any longer perceptual. Yet, semantic is not the opposite of perceptual; what the authors have in mind is that the representation in the HP has a meaning, i. e. refers to something in the outside world. A less controversial choice to replace semantic-based could be referential, integrative or even better conceptual (referring to the concept cells described in HP by the Fried group); the latter choice would be optimal as the neurons reported here in HP bear striking similarities to the concept cells of Quiroga et al, and are in all likelihood the same population.

We thank the reviewer for the clarification and largely agree with the points raised. In particular, the reviewer is correct that the novel region-based coding (i.e., MTL neurons encoding a receptive field in the VTC neural feature space), as shown in our study, does not strictly correspond to object categories or semantic coding—although objects within the VTC neural feature space do tend to cluster according to semantic categories (please also see our response to the next question). Accordingly, we have de-emphasized the notion of semantic coding in the revised manuscript. Specifically, as suggested by the reviewer, we have revised the framing from “semantic” to terms such as “conceptual”, “category-selective”, “integrative”, or “higher-level processing” to better align with the literature, improve accuracy, and minimize potential confusion. Please see below for the key changes:

In Abstract:

“The VTC exhibited axis-based feature coding, and a neural feature space could be constructed using VTC neural axes, within which visual objects clustered according to high-level categorical relationships. Importantly, MTL neurons encoded receptive fields within this VTC neural feature space, exhibiting selective responses to objects that shared perceptual and conceptual similarities. This computational framework, therefore, explains how dense, feature-based representations in the VTC are transformed into sparser, higher-level representations in the MTL.”

In Introduction:

“Critically, region-based coding can provide a potential link between visual feature processing in the VTC and conceptual representations in the MTL (**Fig. 1**). A neural feature space can be constructed from VTC neural axes using principal component analysis (PCA). Notably, the dimensions of this space reflect principal variations in object categories (e.g., ranging from natural to artificial objects or from animate to inanimate objects; see **Methods**)^{3,4}; and within this space, objects with similar high-level abstract features cluster together, demonstrating clustering that reflects their perceptual and conceptual relationships. If MTL neurons exhibit receptive fields (i.e., coding regions) within this neural feature space and respond to stimuli that fall into these regions and share similar high-level properties, they will exhibit category-selective responses. Therefore, this neural computational framework can explain how the brain transforms feature-based representations in the VTC into more abstract and integrative representations in the MTL.”

In Fig. 1 legend:

“**Fig. 1.** A neural computational framework illustrating the transition from axis-based coding in the ventral temporal cortex (VTC) to region-based coding in the medial temporal lobe (MTL). Notably, within this space, objects with similar high-level abstract features cluster together, demonstrating clustering that reflects their perceptual and conceptual relationships. By responding to stimuli that fall within these regions, MTL neurons exhibit selective responses to objects that share similar perceptual and conceptual features, thereby providing a crucial link between feature-based visual processing in the VTC and higher-level representations in the MTL.”

In Results:

“This provides a computational framework linking VTC and MTL representations, allowing us to understand how perceptual representations in the VTC are translated into abstract conceptual representations in the MTL.”

“Together, our results reveal region-based feature coding of MTL neurons within the VTC neural feature space, providing a computational framework for translating perceptual processing in the VTC into conceptual processing in the MTL.”

“Importantly, we further showed that axis-coding channels had a stronger feedforward GC from the VTC to the AH in the theta frequency range compared to non-axis-coding channels (Fig. 6e), confirming the feedforward information flow (i.e., providing structured feature information from the VTC to support higher-level processing in the MTL) as shown by the PLV analysis.”

In Discussion:

“The underlying neural circuits and pathways involve a critical progression of information processing from the VTC to the MTL, where complex visual features are extracted and transformed into meaningful category-specific representations, allowing us to recognize objects regardless of changes in viewpoint, size, or context.”

“Together, our study reveals a computational framework that explains the transition of visual coding from dense feature-based to sparse concept-based representations, providing a mechanistic understanding of the neural processes underlying object recognition.”

“To address this question, our recent studies ^{1,7} uncovered a novel region-based coding mechanism in the human MTL that could explain the transition from dense feature-based representations in the higher visual cortex to sparse concept-based representations in the MTL.”

“Our findings align with the known anatomical connections between the VTC and MTL and the neural pathway for object processing ⁸, suggesting the neural basis of the computational framework for translating perceptual information into conceptual representations.”

“In conclusion, the transition of visual coding from the VTC to the MTL is a crucial process for transforming detailed visual information into higher-order, abstract representations.”

“Our computational framework thus serves as a bridge between VTC and MTL representations, allowing us to decipher how perceptual representations in the VTC are translated into conceptual representations in the MTL.”

We have also revised various places in **Supplementary Discussion**.

In addition, we would like to clarify several points. First, while the term “semantic” may not accurately describe our present results due to the nature of region-based coding, it is appropriate to use “semantic”

when referring to prior literature that demonstrates category-selective or identity-selective (see below) responses in the MTL ^{9,10}.

Second, in the four remaining instances in **Supplementary Discussion** and in this rebuttal, we used the term *semantic* similarly to how others might refer to *conceptual*—both describe relationships between objects that share meaning or categorical membership, regardless of their visual appearance. For example, a “car” and a “truck” may differ substantially in shape or color, but they are both semantically related as vehicles. Thus, when we refer to “semantic representations” or “semantic similarity”, we refer to representations that group objects according to shared meaning or category membership (e.g., animals vs. tools), not necessarily according to perceptual similarity. This framing aligns with previous work in cognitive neuroscience that distinguishes between perceptual and semantic object representations and allows us to probe category-level coding beyond low-level visual similarity.

Lastly, we would like to further clarify the relationship between semantic and conceptual coding. While the terms *semantic* and *conceptual* are often used interchangeably in the literature, there is a subtle distinction: *semantic* typically refers to meaning-based relationships grounded in language or world knowledge (e.g., the knowledge that apples and bananas are both fruits), whereas *conceptual* can refer to more abstract or flexible mental representations that integrate perceptual, functional, and semantic properties. In this sense, *concepts* may be thought of as higher-order constructs that include semantic relationships as one component, but may also encompass contextual or goal-related associations depending on task demands.

In fact, the data provided in the manuscript show that the representation in HP is NOT semantic in nature: indeed in reply the reviewer 3 they show that the semantic matrix of their stimuli accounts for only 8% of the variance in neural activity of HP. While this more than the perceptual matrix (the intention of the authors), it is also clearly contradicting the claim that the representation is semantic-based.

We thank the reviewer for raising this important point. We would like to clarify two aspects to address the apparent contradiction. First, while the semantic similarity matrix explains only a modest proportion (~8%) of the variance in hippocampal responses, this does not preclude the presence of semantic

representations. Many prior studies have demonstrated semantic tuning in the hippocampus using different paradigms and analysis approaches, including those focused on category selectivity, concept representation, and associative structures^{9,11-14}. The relatively low variance explained in our study likely reflects the complexity and heterogeneity of hippocampal coding, which may not align neatly with any single computational semantic model. The key message here is that hippocampal responses align more closely with semantic than with visual features.

Second, and more importantly, we would like to emphasize that our model of region-based coding in MTL is not strictly “semantic” in nature. Rather, it reflects a topological organization in a feature space shaped by both perceptual and conceptual properties, where clusters of stimuli emerge based on shared representational characteristics. This coding framework may serve as an intermediate structure that supports the transition from perceptual representations in the VTC to more abstract, semantic or episodic representations in the MTL (as discussed in^{1,7}). We have clarified this distinction throughout the revised manuscript to avoid overextending the claim that hippocampal representations are semantic per se.

Furthermore, the authors do not report their main data (fig 3 in old manuscript) accurately: they claim it shows the HP neurons associate different features; this is incorrect: what the figure shows is that the HP neurons associate particular strengths of features provided by the axis selectivity of VTC neurons. Such association of strengths is typical of a single object/person, not a semantic category, as they authors themselves acknowledge in one of their replies to reviewer 3. In this reply they insist that the HP representation is not that of categories. Hence it is difficult to understand how the authors can claim that the HP representation is semantic-based.

We thank the reviewer for the comments. We believe the reviewer is referring to the results shown in **Fig. 4** (**Fig. 3** presents iEEG results on axis coding and does not include hippocampal neurons). Furthermore, we believe the phrase “associate particular strengths of features” refers to our region-based coding framework, in which hippocampal neurons encode a receptive field in the VTC neural feature space. The reviewer raises an important point that such region-based coding may not align strictly with semantic categories—it can involve subsets of objects within a category or combinations of objects across categories. In light of this, we have removed references to “semantic-based representations” throughout the revised manuscript. Please refer to our responses above.

In their rebuttal the authors claim: ‘We observed category-selective MTL cells, which have long been regarded as fundamental building blocks of semantic memory 2-5—one component of declarative memory, alongside episodic memory.’ I do not understand this claim, as nothing in the present manuscript support this claim, to the contrary (see above)

We thank the reviewer for pointing this out, and we agree with the reviewer. Accordingly, we have removed this argument (“In particular, neurons in the human MTL have long been linked to the category-specific encoding of visual objects and faces ^{9,12,15}, which are regarded as the building blocks of declarative memory ^{10,11,14,16}.”) from the revised manuscript.

They also point to an ‘extensive literature has also demonstrated semantic processing in MTL ‘

The oldest of these references (Kreiman) refers to the old literature describing face and body patches in IT/VTC as category selective. We now know these are simply regional specializations of the feature space in IT/VTC. In a paper recently published in Current Biology, the same group of authors showed that the analysis presented here for objects also is valid for faces. Hence this paper does not support the authors claim.

We thank the reviewer for the comments. It is important to note that the Kreiman et al. paper (“Category-specific visual responses of single neurons in the human medial temporal lobe”) focused on the MTL, not the IT/VTC, and reported category-selective responses in the MTL, including the hippocampus and amygdala. Therefore, it directly supports our reference to semantic processing in the MTL.

Furthermore, our recent Current Biology paper demonstrated similar computational principles for face representations. In the present study, we extended this analysis to a broader range of object categories beyond faces. We do not see this as contradictory, but rather as a generalization of our previous findings to more diverse stimuli. Our claim is that the amygdala and hippocampus contribute to higher-level representations of object categories, consistent with a large body of literature.

Reber et al 2015 claims to have evidence for semantic category processing in HP. These authors analyzed the population vector summing all HP neurons and show that it carries some information about the superordinate category of their stimuli. This however was in fact an analysis of the noise in the firing rate (the analysis also held when restricting it to the unresponsive neurons), which has no use in coding object identity (thus no use for episodic memory). The present results suggest this might simply reflect implicit category information present in the feature space of the input to HP. Indeed the fig 4a of the present manuscript (revised) has some striking similarity with fig 3F in Reber et al. Thus one might speculate that the feature space represented by VTC input might be analyzed in parallel in two different ways in MTL and TP. This reference fails to show that neuronal responses to visual stimuli carry category information, and thus does not support the claim of the authors.

We thank the reviewer for these comments. Regarding Reber et al. (2015), we would like to clarify two points. First, while the analysis included neurons that did not reach conventional thresholds for visual responsiveness, this does not imply the signal was merely noise. As has been shown in multiple studies, subthreshold or weakly responsive neurons can still carry meaningful information when analyzed at the population level. In fact, Reber et al. demonstrated that the population activity—even from such neurons—was systematically related to superordinate object categories. Thus, rather than reflecting random noise, this population-level signal suggests the presence of structured, category-related information in the hippocampus.

Second, we agree with the reviewer that Fig. 3F in Reber et al. bears a notable resemblance to **Fig. 4a** of our manuscript. This similarity supports the idea that object representations are embedded in a lower-dimensional feature space, likely inherited from VTC inputs, and that this space provides the substrate for further categorical abstraction in MTL regions. We find the reviewer’s suggestion—that MTL and TP may engage in parallel but distinct modes of analysis over this input feature space—both compelling and consistent with our data.

Finally the papers from the Fried group (mainly with Quiroga as first author) described the concept cells which match very closely the properties of the HP neurons described in the present manuscript. While clearly describing how the properties of these neurons make them suitable for recording personal experiences and episodic memory, this group sometimes refers to them as semantic processing (see eg

the Nat NS Rev 2015 Quiroga et al). In the use of this word they point to the fact that the HP neurons represent something in the physical world- very like the real world entities I coined, Orban et al Front. Psychol 2014- and thus have a meaning. Used this way 'semantic processing' does not imply any relation to semantic memory or semantic categories.

Thus if the authors persist in using the word 'semantic-based' they should at least clarify what they mean by this word. However, given that it is now placed at the center of the manuscript (unlike in the Quiroga papers), it seems to be much wiser to avoid using this word and replace it by conceptual processing. As mentioned above, this refers to the concept cells described by the Fried group, which are well established, and matches the meaning of concept: 'a generic mental image abstracted from percepts', according to the Webster.

We thank the reviewer for the detailed clarification, and we largely agree. We have removed the term 'semantic' from the revised manuscript; please refer to our responses to the questions above. Furthermore, we would like to clarify that concept cells may encode a very narrow category—such as a specific identity or place—and can therefore be broadly construed as category-selective, encoding a semantic category.

Reply to comments from Reviewer 2

The authors have provided a revision to their manuscript in response to the first round of reviews. In general, it appears that the authors have sufficiently addressed many of the criticisms raised in the initial reviews, and have tempered some of their claims regarding the interpretation of the results. This is a valuable manuscript presenting a rare dataset, and will have of high interest for researchers interested in understanding visual object recognition and coding in the human brain.

Once again, we thank the reviewer for the expert and constructive comments.

Reply to comments from Reviewer 3

General Comment

I appreciate the authors' thoughtful and detailed responses to my previous comments, as well as the substantial revisions made to the manuscript. I particularly value the additional analysis regarding shared computational mechanisms for face and object coding in the VTC, as well as the enhanced mechanistic analysis provided in the latter half of the paper. However, there may still be room for further examination regarding the validation of the computational framework presented in the first part.

Overall, the study addresses an important and compelling question in systems neuroscience. The proposed computational framework, which links axis-based coding in the VTC with region-based semantic representation in the MTL during object recognition, is highly valuable and thought-provoking.

In the following, I will provide comments on each of the authors' previous responses. In doing so, I will also attempt to suggest possible ways to more rigorously evaluate the proposed computational framework. Italicized segments indicate the authors' responses, whereas standard (non-italicized) text contains my own remarks.

Once again, we thank the reviewer for the expert and constructive comments.

Major Points for Revision

1. Evidence for axis coding in the VTC

We thank the reviewer for the question and suggestions. First, we apologize for an error in estimating the R^2 for the partial least squares (PLS) model, which has now been corrected in Supplementary Fig. 2a. Please note that although the trend remains consistent, the mean R^2 is now 0.6, indicating a good fit of the model.

The correction of the R^2 values from 0.1 to 0.6 is indeed substantial, and I appreciate the authors' transparency in addressing this error. However, such a large change raises concerns about the original implementation of the model and how the correction was made. It would be helpful if the authors could briefly explain the nature of the error (e.g., whether it was a coding mistake, a misinterpretation of the

R² definition, or some other issue) and clarify how the corrected values were validated. Providing this information would enhance confidence in the revised analysis and its implications for the validity of the axis-coding model.

We thank the reviewer for raising this question, and we apologize for not explaining it thoroughly in our previous response. We implemented a cross-validation procedure to evaluate model performance and calculate statistical significance. The previously reported R^2 was based on the held-out test dataset (50%). However, to determine which DNN layer best models the neural responses—in other words, which set of features best fits the data—it is more reasonable to estimate R^2 using the full dataset, as the final axis-coding models were trained on all available data. This approach not only increases the amount of data but also includes the training set, which naturally results in a higher R^2 .

While we acknowledge that a large difference in performance between training and testing sets may suggest overfitting, this did not affect our DNN layer selection. Both the testing dataset (previous **Supplementary Fig. 2a**) and the full dataset (current **Supplementary Fig. 2a**) consistently indicated that layer pool5 yielded the best model fit. It is also worth noting that pool5 did not have the highest dimensionality among the layers, making it less likely to be the most overfitted.

We have clarified this in the revised manuscript:

“ R^2 was calculated using the full dataset; however, R^2 calculated on the test dataset showed a similar pattern, suggesting that layer pool5 best fit the neural responses.”

Additionally, the R^2 values reported for the pool5 layer appear to be highly similar across regions (Supplementary Fig. 2a), which seems to contradict the patterns shown in panel b of the same figure and the associated main text description suggesting regional differences. It would be beneficial for the authors to clarify whether this apparent discrepancy is due to a difference in what is being measured (e.g., number of significant channels vs. model fit quality) or whether the similarity in R^2 curves across regions has a specific interpretational significance.

We thank the reviewer for pointing this out. This apparent discrepancy likely arises from differences in what is being measured. Specifically, **Supplementary Fig. 2a** shows R^2 values for all visually

responsive channels, calculated using the full dataset. In contrast, **Supplementary Fig. 2b** and **Fig. 3d** show the strength of axis coding, quantified as the correlation coefficient between the observed and predicted responses in the test dataset. The predicted responses were generated using models trained only on the training dataset (see **Methods** for details). It is likely that the models fit the data equally well across channels, but the inherent predictability varied—reflecting differences in data consistency. In summary, R^2 values reflect how well the model explains neural responses and are used to compare relative model fits across feature layers, whereas the strength of axis coding reflects generalization performance.

Second, to determine whether axis coding or category-based coding better explains neural responses in the VTC, we compared the R^2 from the PLS model with that derived from an ANOVA applied to neural responses using categorical labels for the category-based coding model. Indeed, we found that across all visually responsive channels, the axis model explained significantly higher variance compared to the category model in both the FG (axis: 0.63 ± 0.038 [mean \pm SD]; category: 0.17 ± 0.087 ; two-tailed paired t -test: $t(200) = 91.94$, $P < 10^{-20}$) and ITG (axis: 0.62 ± 0.030 ; category: 0.12 ± 0.046 ; $t(73) = 63.34$, $P < 10^{-20}$; Supplementary Fig. 2d). We have included the following results in the revised manuscript:

The direct comparison between the axis and category models using variance explained clearly supports the superiority of axis-based coding in both FG and ITG regions. This is a strong and well-motivated addition. While the statistical differences are robust and the results are clearly presented, I have a few concerns regarding the interpretation of this comparison that merit clarification.

The PLS model for axis-based coding is based on continuous DNN features that inherently contain high-dimensional and fine-grained visual information, whereas the ANOVA model relies on discrete categorical labels. Given this asymmetry in the richness of input features, the superior performance of the PLS model may be expected. Could the authors comment on how they controlled for the differences in input dimensionality and information content between the two models?

We thank the reviewer for pointing this out, and we agree that differences in model fitting can be influenced by differences in input dimensionality. To control for this, we additionally calculated R^2 using

a one-dimensional feature (the first PLS component), ensuring that both the PLS model and the ANOVA model had one-dimensional input. Although, as expected, R^2 was reduced when we restricted the PLS input dimensionality, the PLS model still showed a better fit even under this constraint for both the FG (axis: 0.22 ± 0.06 [mean \pm SD]; category: 0.17 ± 0.087 ; two-tailed paired t -test: $t(200) = 54.84$, $P < 10^{-20}$) and ITG (axis: 0.20 ± 0.04 ; category: 0.12 ± 0.046 ; $t(140) = 56.25$, $P < 10^{-20}$). We have included the following results in the revised manuscript:

“Notably, across all visually responsive channels, the axis model explained significantly more variance compared to the category model in both the FG (axis: 0.63 ± 0.038 [mean \pm SD]; category: 0.17 ± 0.087 ; two-tailed paired t -test: $t(200) = 91.94$, $P < 10^{-20}$) and ITG (axis: 0.62 ± 0.030 ; category: 0.12 ± 0.046 ; $t(73) = 63.34$, $P < 10^{-20}$; **Supplementary Fig. 2d**; similar results were obtained when controlling for the input dimensionality of the axis model and the number of object categories used in the category model).”

Relatedly, further clarification is warranted regarding the construction of the category-based model. The ANOVA model appears to rely on ImageNet-derived category labels comprising approximately 50 relatively fine-grained object classes (e.g., “young mammal,” “rodent,” “frog,” “reptile”), many of which are closely related. Such fine-grained labeling, combined with potentially limited trial numbers per class, raises the possibility that the model’s R^2 values may be underestimated. It would be helpful if the authors could clarify the rationale behind this categorical structure and, additionally, evaluate whether alternative grouping schemes—such as clustering based on superordinate semantic categories—would affect the comparison with the axis-based model. Such analyses would strengthen the validity and generalizability of the conclusions. Previous studies have shown that higher layers of CNNs represent semantic category-level information (e.g., Bau et al., 2017; Khaligh-Razavi & Kriegeskorte, 2014). Given this, and considering that the activity of individual VTC channels is well explained by deep CNN features, the relatively low R^2 values obtained from the ANOVA-based category model are somewhat unexpected. It would therefore be valuable to discuss possible reasons for this discrepancy.

We thank the reviewer for the suggestion, and we agree with the reviewer. As the reviewer mentioned, we initially used the ImageNet-derived category labels. In the revised manuscript, we further repeated our analysis using broader categories as in **Fig. 4a** (animal, insect, man-made objects, equipment,

vehicle, food, plants, and natural objects) for the ANOVA-based category model. The axis model still explained significantly more variance than the category model in both the FG (axis: 0.63 ± 0.038 [mean \pm SD]; category: 0.052 ± 0.056 ; two-tailed paired t -test: $t(200) = 182.75$, $P < 10^{-20}$) and ITG (axis: 0.62 ± 0.030 ; category: 0.024 ± 0.019 ; $t(140) = 213.25$, $P < 10^{-20}$). Therefore, our results cannot simply be attributed to the number of categories used in the analysis. It is worth noting that the ANOVA model using broader categories yielded even lower R^2 values. Furthermore, the results remained significant when we restricted the input dimensionality of the PLS model (see above; both P s < 0.0001). This result has been included in the revised manuscript (please see above).

Furthermore, we agree that the relatively low R^2 values from the ANOVA-based category model may initially seem surprising, given prior findings that higher layers of DNNs encode category-level information. One likely reason for this discrepancy is that our category model treats categories as discrete labels and captures only between-category variance, whereas neural responses in the VTC are better characterized by continuous visual feature tuning that spans across category boundaries. In contrast, the axis model captures this continuous variation along meaningful feature dimensions, providing a more accurate account of the observed neural responses. Additionally, even though higher DNN layers are associated with categorical representations, they still retain substantial continuous visual structure (e.g., shape, texture), which may not be fully captured by discrete category models. These factors together may explain why the axis-based feature model accounts for more variance than the category-based model.

It is important to note that while the highest DNN layers may correspond more closely to semantic representations, the VTC may more closely align with intermediate DNN layers (see **Supplementary Fig. 2a** for details), which retain rich visual information without abstracting fully to semantic categories. This intermediate-level feature representation in both VTC and DNNs may explain why the axis model—derived from DNN features—is more successful than a coarse category model in capturing VTC activity. Thus, the relatively low R^2 values of the category model likely reflect both the limitations of discrete category labels and the nature of VTC coding itself.

Overall, the comparison between the PLS and ANOVA models is a valuable addition. However, I encourage the authors to justify more carefully the conclusion that axis-based coding outperforms

category-based coding, from a principled and biologically grounded perspective—particularly given that the way category coding is modeled is a critical factor in this comparison. In this regard, the authors might also consider incorporating alternative models that take into account lexical or semantic distances—such as those derived from word embeddings—as they have explored in later sections of the manuscript. This could provide a more graded and neurobiologically plausible account of semantic representation in the VTC and MTL.

We thank the reviewer for the thoughtful comments and suggestions. We agree that incorporating alternative models that account for lexical or semantic distances would be valuable for understanding which features best explain VTC responses. As suggested by the reviewer, we extracted word-label embeddings for the 50 included categories using the Global Vectors for Word Representation (GloVe)¹⁷ model. We then compared the correspondence between VTC neural responses and word embeddings versus visual features using RSA. We found that visually responsive VTC channels (**Revision Fig. 1a**) exhibited a higher, though not statistically significant, correspondence with visual features (Spearman's $\rho = 0.23$) compared to word embeddings (Spearman's $\rho = 0.13$; permutation test against the null distribution of differences: $P = 0.13$). This pattern held when considering the entire population of VTC channels (**Revision Fig. 1b**), suggesting that category-based coding does not better explain VTC responses at the population level. As expected, channels classified as axis-coding (**Revision Fig. 1c**) demonstrated a significantly higher correspondence with visual features (Spearman's $\rho = 0.39$) than with word embeddings (Spearman's $\rho = 0.22$; $P = 0.005$). Notably, the same pattern was observed even in category-selective channels (**Revision Fig. 1d**). Together, these results support the conclusion that visual models provide a better explanation of VTC neural responses than word-based category models.

Revision Fig. 1. Comparison of visual and lexical representations in the VTC. Representational similarity analysis (RSA; see **Methods**) was conducted to assess the correspondence between VTC

neural responses and visual embeddings, as well as between VTC neural responses and word embeddings. **(a)** Visually responsive channels. **(b)** All channels. **(c)** Axis-coding channels. **(d)** Category-selective channels.

It is worth noting that we could not fit a single regression model that includes both axis coding and category-based coding variables to directly compare their contributions. Additionally, we would like to clarify that while representational similarity analysis (RSA) can be used to construct image-by-image dissimilarity matrices (DMs) for visual features (i.e., by correlating visual feature vectors between each pair of images) and for neural responses (i.e., by correlating neural vectors between each pair of images), it cannot be used to construct DMs based on coding schemes. Therefore, we could not use this approach to test whether the DM derived from axis coding better aligns with neural responses in the VTC. In other words, RSA can be used to examine the similarity between coding schemes or between brain areas (e.g., Fig. 3h), but not to compare how well different coding schemes explain neural responses.

*The authors correctly point out the methodological challenges in directly comparing axis-based and category-based coding models within a unified regression framework, as well as the limitations of RSA in assessing model fit. However, as discussed in the preceding point, given the differing nature of the models (PLS vs. ANOVA), it remains unclear whether the higher R^2 values observed for the PLS model truly reflect superior explanatory power, or whether they may instead result from differences in model complexity or representational mismatch. While it is true that RSA is not designed to compare explanatory power directly, it could still serve as a complementary approach. Specifically, comparing dissimilarity matrices (DMs) derived from axis-based, category-based, and neural representations may offer useful insights into the nature of VTC representations (Kriegeskorte et al., *Front. Syst. Neurosci.* (2008). Khaligh-Razavi & Kriegeskorte, *PLoS Comput. Biol.* (2014), Jozwik et al., *Journal of Neurosci.* (2016)). In fact, the authors have already performed a similar analysis in Revision Fig. 6, which appears to yield valuable findings.*

We thank the reviewer for the clarification and we agree with the reviewer. In the revised manuscript, we have consolidated the comparison between the PLS and ANOVA models (see response above). Additionally, as in the previous **Revision Fig. 6**, we further compared axis-based and lexical/semantic-

based coding using RSA (as categorical labels used in the ANOVA model could not be used to construct a dissimilarity matrix). Please refer to our response to the question above.

Third, we thank the reviewer for acknowledging the cross-dataset generalizability and consistency of the axes as a major strength of our work. We also appreciate the reviewer's suggestion. In response, we have characterized the extracted feature axes in greater detail and compared them with known object feature dimensions. The following results have been included in the revised manuscript:

Thank you for the additional analyses and the detailed revise. The new results characterizing the neural axes along the natural–artificial, animacy, and spiky–stubby dimensions provide useful insight and clearly demonstrate that the extracted axes align with previously reported visual feature dimensions. The cross-dataset replication using the Microsoft COCO stimuli further supports the robustness of these dimensions.

We thank the reviewer again for the insightful suggestions.

2. Region-based feature coding in the MTL relative to alternative coding schemes

We thank the reviewer for raising this important question. Indeed, in our recent study 13, we have demonstrated the absence of axis coding in MTL neurons using the same set of object stimuli. Notably, in that work, we also showed that region-based coding in the MTL provides a more comprehensive mechanism for explaining category selectivity. Specifically, the region code does not rely on categorical membership of individual images, as long as the images share similar visual features. While images from the same category often cluster together in feature space, which can be captured by category-selective coding, images from different categories may also cluster due to shared visual features (e.g., structure, texture), which category coding alone cannot account for. The broader, feature-based representation observed in MTL neurons suggests that region-based coding serves as a more general and robust model for transforming the VTC feature space into MTL representations 13.

I understand that in their recent work, the authors report the absence of axis coding in MTL neurons and instead propose a region-based coding scheme in the MTL. This form of coding can be viewed as

intermediate between axis-based and label-based category coding. In terms of semantic abstraction, one might conceptualize a hierarchy from axis coding (low-level), to region-based coding (intermediate), to label-based category coding (high-level), with corresponding models being the PLS model, semantic embedding models, and the ANOVA model, respectively. While this may be a somewhat simplified interpretation or may not fully capture the authors' framework, it nonetheless offers a useful way to contextualize the different coding schemes and their associated analyses.

Given this, it would be informative to evaluate neural responses in each region using all three models. Although direct comparisons between regions may be challenging due to differences in measurement modalities (e.g., iEEG vs. single-unit recordings), I believe that within-region comparisons of explanatory power—along with representational similarity analyses (RSA) comparing dissimilarity matrices predicted by each model—are both meaningful and feasible.

We thank the reviewer for the suggestions. The reviewer has accurately captured our proposed computational framework, and we agree with the insightful conceptualization of “a hierarchy from axis coding (low-level), to region-based coding (intermediate), to label-based category coding (high-level), with the corresponding models being the PLS model, semantic embedding models, and the ANOVA model, respectively”. Furthermore, the reviewer is correct that between-region comparisons are challenging, whereas within-region comparisons using data from a single measurement modality are more feasible. Accordingly, we used iEEG responses for model comparisons in the VTC, and single-unit responses for model comparisons in the MTL.

Specifically, as shown in **Revision Fig. 1**, VTC neural responses were better explained by a visual model than by a lexical/semantic model. In contrast, as shown in our previous **Revision Fig. 6**, MTL region-coding neurons exhibited greater representational similarity with semantic representations than with visual representations (similar results were obtained using the GloVe model). These findings support the proposed transition in coding models from the VTC to the MTL. It is worth noting that the ANOVA model could not be evaluated using RSA, as the categorical labels it relies on cannot be used to construct a dissimilarity matrix. Furthermore, while explanatory power (R^2) could not be calculated for semantic features—since no specific predictive model was assumed—RSA is more appropriate in this case. Nonetheless, the R^2 -based analyses (see above) further support visual, rather than category-based, coding in the VTC.

3. Relationship between face-selective channels and axis coding in the VTC

We thank the reviewer for the suggestions. First, we would like to clarify that the current study focused on general object coding rather than face coding. We agree that it is an important question to investigate whether face selectivity in the human FFA corresponds to axis coding, as observed in non-human primate (NHP) face patches. However, this question falls slightly outside the scope of the current study, as we did not specifically include face images. We would also like to clarify that our results did not rely on face/object selectivity, and we did not argue for face/object-selective processing or domain specificity in face/object processing. Rather, the VTC, particularly the fusiform gyrus (FG), may exhibit similar computational principles for both faces (cf. 25) and non-face objects (our present study), consistent with previous macaque studies demonstrating common computational principles of axis coding for both faces¹⁵ and objects¹⁴.

Thank you for the clarification. I appreciate that the focus of the present study is on general object coding rather than face-selective processing. Nevertheless, I sincerely appreciate the additional analysis you performed, which I found highly interesting and informative.

We thank the reviewer again for this insightful suggestion.

Second, as the reviewer suggested, we conducted additional control analyses using faces. We recorded neural responses to both faces and objects from a subset of 11 patients (14 sessions), allowing us to further examine whether axis coding generalizes across object and face stimuli. We found that face-selective channels (10/57, 17.54%) were not significantly more likely to be axis-coding channels compared to the overall population (40/240, 16.67%) in the VTC (χ^2 -test: $P = 0.87$), suggesting that axis coding is not specific to face-selective channels. Additionally, we found that 29 out of 72 VTC channels (40.28%) that exhibited axis coding for object stimuli also exhibited axis coding for face stimuli. These results suggest that axis coding functions as a general coding principle for visual stimuli in the VTC, encompassing both faces and objects. Similarly, when examining the FG specifically, we found that face-selective channels (8/26, 30.77%) were not significantly more likely to be axis-coding channels compared to the overall population (36/135, 26.67%; χ^2 -test: $P = 0.67$). Moreover, 26 out of

57 FG channels (45.61%) that exhibited axis coding for object stimuli also exhibited axis coding for face stimuli.

Thank you for conducting the additional analyses using face stimuli. The results directly demonstrate that axis coding is not limited to face-selective channels, and instead may represent a more general coding principle for visual information in the VTC. The finding that a substantial proportion of axis-coding channels respond similarly to both object and face stimuli further supports this interpretation. I appreciate the effort to clarify this point, as it helps distinguish axis coding from domain-specific selectivity and highlights its broader relevance across stimulus categories.

We thank the reviewer again for the suggestion and comment.

Lastly, we further analyzed the relationship between category coding and axis coding for general objects and found that 95 out of 157 (60.5%) category-selective channels in the VTC also exhibited axis coding, suggesting that axis coding is a common representational format among category-selective channels in the VTC. Specifically in the FG, 50 out of 66 (75.76%) category-selective channels also exhibited axis coding.

This additional analysis meaningfully strengthens the conclusion that axis coding serves as a general representational format within the VTC, particularly among category-selective channels. The high proportion of overlap—especially in the FG—provides compelling support for the idea that axis-based representations are a core feature of category-selective processing in the human ventral visual stream.

We thank the reviewer again for the comment.

4. Relationship between regional coding and semantic coding

We thank the reviewer for this important question. First, we apologize for not clearly stating that semantic clustering emerges within the neural feature space constructed by visual axes (Fig. 1). Specifically, although the neural feature space is based on visual features, it ultimately organizes information semantically. Therefore, the neural feature space—and the region coding of MTL neurons within it—translates feature-based representations in the VTC into semantic-based representations in

the MTL. We have clarified this key point in the Abstract, Introduction, and in the legend of Fig. 1. Please also refer to our detailed responses to Reviewer 4's first two questions.

Third, we further examined whether abstract category representations or purely visual features better align with region coding in the MTL using representational similarity analysis (RSA). To this end, we constructed neural, visual, and semantic representational dissimilarity matrices (RDMs). Specifically, we built a category-by-category visual RDM using DNN features (Revision Fig. 6a, left), which we previously used to test axis coding. Similarly, we constructed a neural RDM using responses from all region-coding neurons in the MTL (Revision Fig. 6a, middle). To build the semantic RDM (Revision Fig. 6a, right), we applied the SGPT model 26, a large language model, to extract semantic embeddings from text descriptions of each image. These text descriptions were generated using the ALBEF model based on the object images 27. We found that MTL region-coding neurons exhibited greater representational similarity with semantic representations (Spearman's $\rho = 0.26$, $P = 0.0005$) than with visual representations (Spearman's $\rho = 0.16$, $P = 0.0009$), establishing a link between region coding and semantic representation in the MTL and supporting the idea that region coding serves as an intermediate mechanism for transforming visual features into semantic object representations.

In summary, given that region coding organizes representations semantically, it should be considered a semantic transformation rather than a mere extension of perceptual processing. The theoretical framework describing how region coding bridges perceptual and semantic representations has been made more explicit in the revised manuscript. Please refer to our substantially revised Abstract, Introduction, and Fig. 1 legend.

I appreciate the authors' clarification. In particular, the point that the neural feature space is constructed based on visual axes yet naturally gives rise to semantic clustering is both interesting and compelling. This perspective effectively highlights the role of region-based coding as a potential bridge between perceptual and semantic representations. I also commend the authors for reflecting this conceptual framework in the revised Abstract and Introduction. Nonetheless, I still find it somewhat unclear how regional coding in the MTL gives rise to semantic representations. While the example I previously gave regarding V1 might not have been ideal, the broader point remains: in visual processing, it is quite common for higher-level receptive fields to emerge from the integration or convergence of lower-level feature representations.

In this sense, while the transformation observed between the VTC and MTL may be interpreted as semantic in nature, it could also reflect a more general hierarchical integration process, similar to what is observed between intermediate visual areas such as V4 and IT. That is, the emergence of apparent semantic structure may not necessarily imply a dedicated semantic transformation, but could arise from the cumulative convergence of feature representations.

Therefore, interpreting this as a semantic transformation through “regional coding” requires caution, especially because the term “regional coding” itself can carry implicit semantic connotations. To avoid circular reasoning, it would be preferable to explicitly define “regional coding” in non-semantic terms and describe its relation to semantic structure as reflective or suggestive, rather than causative.

That said, it is also possible that the nature of convergence differs between these stages, and future work may help elucidate whether and how the VTC-to-MTL transformation departs from purely perceptual hierarchical integration, and to what extent it reflects uniquely semantic processing.

We thank the reviewer for this insightful and constructive feedback. We appreciate the reviewer’s recognition of our conceptual framework; and we agree that caution is needed when interpreting the transformation from VTC to MTL as “semantic”. We acknowledge that such convergence could be the result of a general hierarchical integration process, similar to those observed along the ventral visual stream (e.g., V4 to IT), rather than a transformation uniquely tied to semantics. We now highlight this alternative possibility in the revised **Discussion**, emphasizing that the apparent semantic structure in MTL may reflect the cumulative integration of lower-level visual features, and not necessarily a dedicated semantic operation:

“An alternative interpretation of the VTC-to-MTL transformation is that the observed representational changes reflect a general hierarchical integration process, rather than a dedicated transformation. Similar to how receptive fields in early visual areas converge to form higher-order visual representations in regions such as V4 and IT, the emergence of region-based coding in the MTL may arise from the cumulative integration of visual features represented in the VTC. In this view, the clustering of neural responses in MTL—sometimes aligning with object categories—could result from overlapping visual feature selectivity, rather than an explicit coding of semantic content. This interpretation is consistent with known principles of hierarchical organization in sensory systems and suggests that apparent semantic structure may emerge naturally from the integration of complex visual information, even in the

absence of explicit category representations. Our findings that region-based tuning in the MTL transcends strict category boundaries further support this view. Future work will be critical to determine whether the transformation from perceptual to conceptual representations in the MTL reflects purely integrative processes or additional mechanisms that encode abstract conceptual meaning.”

It is worth noting that our current data suggest that MTL region coding captures structured information that aligns more closely with semantic embeddings than with purely visual similarity, as supported by the RSA (please see above). While this correspondence is suggestive, we agree with the reviewer that further studies are needed to disentangle whether the VTC-to-MTL transformation reflects uniquely semantic processing or a broader convergence process.

The new analyses using representational similarity matrices (RDMs) based on visual features, neural data, and semantic embeddings are particularly helpful in grounding the interpretation. This corresponds precisely to the RSA (Representational Similarity Analysis)-based comparative analysis I proposed in Major Point 1. The authors' implementation of RDM comparisons between neural, visual, and semantic representations aligns well with the suggestion to use RSA as a complementary method for evaluating model fit and exploring the nature of representational structures. Given its utility, I would recommend applying this approach to other models as well, such as the axis-based and category-based models, to enable a more comprehensive comparison across representational frameworks.

We thank the reviewer again for the suggestions and comments. Please refer to our response above regarding the application of RSA to other models (**Revision Fig. 1**). We agree with the reviewer that this additional analysis enables a more comprehensive comparison of representational frameworks.

Minor Points

1. We have corrected it in the revised manuscript. In the figure legend, we now clarify that “neural response” can refer to iEEG high-gamma power (HGP), as in the present study, or to single-neuron activity, as in classical studies of axis coding in the VTC.

Thank you for clarifying in the revised legend.

We thank the reviewer again for the suggestion.

2. We thank the reviewer for pointing this out. We have corrected it in the revised manuscript.

Thank you for correcting the color assignment.

We thank the reviewer again for pointing this out.

3. We thank the reviewer for the suggestion and we agree with the reviewer. We have replicated the results using response peak latency, and consistent with the findings based on response onset latency, we found that response peak latency in the FG (419.5 ± 61.4 ms [median \pm SD]) was significantly shorter than in all other regions (IT: 460.5 ± 40.8 ms; PH: 472.0 ± 32.5 ms; AH: 474.5 ± 38.9 ms; amygdala: 462.5 ± 43.0 ms; all P s < 0.0001 ; Revision Fig. 7).

Thank you for addressing this point. The inclusion of response peak latency analyses (Revision Fig. 7) strengthens the claim that the observed latency differences reflect genuine processing hierarchies rather than being confounded by response amplitude. The consistency between onset and peak latency results supports the robustness of the hierarchical interpretation.

We thank the reviewer again for the suggestion.

4. We thank the reviewer for the suggestion and we have now provided boundary lines for the FG and ITG in Fig. 3d, with labels in the figure legend

Thank you for implementing the suggested changes. The inclusion of boundary lines and labels for FG and ITG in Fig. 3d significantly improves the interpretability of the figure.

We thank the reviewer again for the suggestion.

5. We thank the reviewer for raising this important question. We agree that the late-onset latency of axis coding in the AH may align with region coding. In our previous study, we also demonstrated that an elevated response at the border of the feature space can drive both axis coding and region coding 1.

However, it is important to note that we did not observe a significant number of iEEG channels exhibiting region coding in the AH, suggesting that axis coding may better account for the iEEG responses in this region. The lack of region-coding channels also prevented a direct comparison of the latencies between axis and region coding. We have acknowledged this in the revised manuscript:

Thank you for the thoughtful clarification. The authors have appropriately acknowledged the temporal similarity between axis and region coding, particularly in the AH, and have discussed the limitations in distinguishing between the two at the iEEG level. The explanation that few region-coding channels were detected in the AH helps justify the interpretation that the observed activity likely reflects axis coding. The added discussion in the revised manuscript adequately addresses the concern. I agree that future studies are warranted to dissociate these mechanisms more directly.

We thank the reviewer again for the suggestion.

6. We thank the reviewer for pointing this out, and we agree with reviewer that response latency could be influenced by response amplitude. To address this, we normalized the response of each channel to its maximum amplitude before calculating response latency and obtained consistent results: we observed a significant correlation between the onset latency of axis coding and the y-coordinate in MNI space within the FG (Revision Fig. 8a), even after controlling for individual differences in response latency (Revision Fig. 8b, as in Fig. 3g). We have included the following results in the revised manuscript:

Thank you for performing the amplitude normalization to address potential confounds in latency estimation. To further strengthen this point, it would be helpful to show whether the normalization indeed removed any pre-existing correlation between response amplitude and latency. Specifically, could the authors report the correlation between response amplitude and latency both before and after normalization? This would clarify whether the observed latency gradient is independent of response magnitude.

We thank the reviewer again for the suggestion. First, as expected, we found a pre-existing correlation between response amplitude (i.e., axis-coding strength) and latency before normalization ($r = -0.43$, $P = 9.77 \times 10^{-5}$). After normalization (i.e., response amplitude normalized to 1), this correlation was eliminated ($r = 0$, $P = 1$). Therefore, the observed latency gradient was independent of response magnitude.

7. We thank the reviewer for pointing this out and the suggestions. First, we have included the representational dissimilarity matrices (DMs) in Supplementary Fig. 5.

I appreciate the addition of Supplementary Fig. 5, which enhances the transparency of the RSA by showing the original dissimilarity matrices used for each ROI. The authors' discussion of the relatively low but statistically significant correlation values is well-reasoned. I agree that the distinct nature of the coding schemes—feature-based representations in the VTC versus categorical representations in the MTL—as well as the nonlinear nature of the transformation between them likely contribute to this pattern.

Interestingly, the axis-coding RDM in the FG (Supplementary Fig. 5a) exhibits a clearer block structure suggestive of category-level clustering than some of the category-coding RDMs in the MTL (Fig. 5c–e). While this might appear counterintuitive, it is consistent with the notion that visual feature spaces derived from naturalistic stimuli can preserve semantic regularities. On the other hand, category-coding channels may operate in a more sparse or selective fashion, reducing the resolution of similarity across all categories. Including a brief discussion of this apparent discrepancy would help clarify how these distinct coding schemes contribute to the transition from perceptual to semantic representations. A brief discussion of this apparent discrepancy would help readers reconcile the functional roles of these distinct coding schemes.

We thank the reviewer for this insightful observation. We agree that the axis-coding RDM in the FG exhibits a clearer block structure than the category-coding RDMs in the MTL, and we appreciate the opportunity to elaborate on this point. We have included the following discussion in the revised manuscript:

“Interestingly, we observed that the axis-coding RDM in the FG exhibited a more prominent block structure than some of the category-coding RDMs in the MTL (Supplementary Fig. 5). Although this may seem counterintuitive, it is consistent with the idea that feature-based representations derived from naturalistic visual input can preserve underlying semantic regularities. In contrast, category-coding signals—particularly in MTL regions—may operate in a more selective or sparse manner, prioritizing categorical boundaries over continuous similarity structure. This distinction suggests that axis coding in perceptual regions such as the FG may support rich, graded encoding of visual similarity, while category coding in the MTL may support more abstract, discrete representations that facilitate memory and decision making. Together, these complementary coding schemes likely contribute to the transformation from perceptual to conceptual representations along the ventral visual pathway.”

8. We thank the reviewer for the suggestion. We have revised Fig. 4a to improve the visualization of the object representational space. Specifically, we grouped the 50 object categories into 6 broader groups and used color coding to illustrate the semantic boundaries. We also added semantic labels for individual categories at their median coordinates to provide fine-grained category information. We believe this revised format not only shows the detailed distribution of object categories within the space but also better highlights the semantic structure and boundaries.

I appreciate the improved visualization in the revised Fig. 4a. Grouping object categories into broader semantic clusters and color-coding them has greatly enhanced readability. The addition of median-category labels provides helpful granularity and clarifies how semantic boundaries emerge in a feature-based representational space. This updated format effectively illustrates the key concept that although visual features vary continuously, categorical structure becomes apparent in the neural coding scheme.

However, one element that may have been lost in the revised figure is the direct visualization of object-level visual features. To better convey the continuity of visual appearance across category boundaries, it may be helpful to provide a few example stimuli located near those boundaries. This addition could enhance the figure by more explicitly demonstrating how semantic structure emerges atop continuous visual similarity, thereby reinforcing the central claim of perceptual-to-semantic transformation.

We thank the reviewer for the suggestion. We agree and have included example object images located near category boundaries to better illustrate how semantic structure emerges atop continuous visual similarity. A new supplementary figure has been added to the revised manuscript:

Supplementary Fig. 6. VTC neural feature space with example images. The feature space is the same as that shown in **Fig. 4a**.

9. We thank the reviewer for pointing this out. The response peak in the permuted density map was due to the heterogeneous (non-uniform) distribution of objects in the VTC neural feature space. We have

acknowledged this methodological caveat in the revised manuscript and confirmed that our procedure reliably identifies coding regions even under heterogeneous distributions:

Thank you for addressing the concern regarding the residual response peak in the permuted density map. Your explanation—attributing the effect to a heterogeneous distribution of object representations in the VTC neural feature space—and your acknowledgment of this caveat in the revised manuscript are appreciated.

We thank the reviewer again for this suggestion.

10. We thank the reviewer for the suggestions. Our goal here is to provide a group-level summary to present a comprehensive encoding profile of region-coding neurons across MTL subregions. In other words, the purpose of this analysis is to characterize the coding properties of MTL neurons within the VTC-defined feature space, rather than to compare the coding properties of MTL and VTC directly. Furthermore, the feature space in this analysis was constructed using VTC responses, so it is not appropriate to use this space to study VTC responses themselves. We have further elaborated on how these results support the concept of region-based coding:

Thank you for your clarification, while I now better understand that the goal of Fig. 4e and 4f is to characterize the encoding properties of region-coding neurons within the VTC-defined feature space—rather than to directly compare MTL and VTC—the current metrics (i.e., number of categories and number of images per tuning region) may not fully capture the key claim that these neurons selectively respond to visual features that cut across semantic category boundaries.

For instance, although the average tuning region contains relatively few categories and images, this alone does not clarify whether the encoded regions correspond to narrow semantic domains or to non-semantic clusters of visual similarity. The interpretation that region-coding neurons are sensitive to continuous visual features rather than categorical labels would be more directly supported if the preferred images spanned multiple categories despite visual coherence, or if the image set showed low semantic but high visual similarity.

To more robustly support this interpretation, I suggest including additional analyses or visualizations—such as examples of selected image clusters per neuron overlaid with category labels, or quantifications of within-cluster semantic entropy or visual similarity. These additions would help clarify whether region-based tuning indeed transcends semantic category boundaries, thus strengthening the central claim of region-based feature representation in MTL neurons.

We thank the reviewer for this excellent suggestion. To further illustrate region-based feature representations in MTL neurons, we visualized object images that fell within the tuning regions of example region-based neurons (upper panels). Additionally, we generated word clouds to qualitatively depict the relative frequency of object categories represented by each neuron (bottom panels). These illustrations show that most object images encoded by Cell #586 (**Fig. 4b**) exhibit nested structures, while those represented by Cell #630 (**Fig. 4c**) feature radical shapes and small sizes. Notably, the clustered images spanned a variety of semantic categories. Furthermore, the number of images from each category within the tuning region varied, and many images from the same category were located outside the tuning region—suggesting that the clustering was driven more by visual features than by semantic meaning. Together, these results indicate that region-based tuning indeed transcends semantic category boundaries.

(b, c) Two example MTL neurons that encoded a region containing visually similar object images in the VTC neural feature space. The left panel shows the neuronal responses to 500 objects (50 object categories; see legend on the left). Trials are aligned to stimulus onset (gray line) and are grouped by individual object category. On each box, the central mark is the median, the edges of the box are the 25th and 75th percentiles, the whiskers extend to the most extreme data points the algorithm considers to be not outliers, and the circles denote the outliers. The middle panel shows the projection of firing rates onto the feature space, with each color representing a different object category (see legend on the left). The size of the dot indicates the firing rate. The coding regions are delineated with encompassed object images and categorical labels in the insets, with text size proportional to the number of encoded stimuli within each category. The right panel shows the estimate of the spike density in the feature space. By comparing observed (upper) vs. permuted (lower) responses, we could identify a region (black contour in the middle panel) where the observed neuronal response was significantly higher in the feature space. This region was defined as the tuning region of a neuron.

11. We thank the reviewer for the comments and suggestions. We have now provided the exact percentage of detected region-coding channels in the text:

Thank you for this thorough and thoughtful revision. The authors now provide a clear explanation for the absence of region coding in MTL iEEG recordings, including both the spatial pooling of signals and possible differences in coding detectability between single-unit and mesoscopic recordings. Including the detection rate (3.31%) was also helpful for interpreting the result.

We thank the reviewer again for the suggestion.

12. We thank the reviewer for the suggestion. We have updated Fig. 5g to enable a more intuitive comparison between the ImageNet and COCO datasets. We have also noted that the same scale is used as in Fig. 4g-i.

Thank you for addressing this point.

We thank the reviewer again for the suggestion.

13. We thank the reviewer for the suggestions. First, we would like to clarify that the RSA in Fig. 3h was conducted across channels—that is, we computed image-by-image correlations using all channels from a brain area, and then correlated the resulting image-by-image dissimilarity matrices between brain areas (see Methods for details). In contrast, the PLV analysis in Fig. 6 was conducted between each pair of channels across brain areas. Therefore, we could not directly correlate the representational similarity results from Fig. 3h with the PLV results. However, we qualitatively compared the results from Fig. 3h with those from Fig. 6

Thank you for the clarification. The finding that FG–AH pairs exhibit strong PLV in the low-frequency bands (theta and alpha), in line with their high representational similarity (as shown in Fig. 3h), provides converging evidence for a functional connection between these regions. While I understand the technical limitations that prevent direct correlation between PLV and RSA, your qualitative comparison meaningfully supports the overall framework.

We thank the reviewer again for the suggestion and comments.

14. We thank the reviewer for the suggestions. First, we have included the PLV results for each MTL subregion (PH, AH, and amygdala) in the revised Fig. 6a, b. We apologize for omitting the PH in the previous manuscript; however, the PH did show consistent results. We have included the following results in the revised manuscript (we have also included the PH in Fig. 6 and Supplementary Fig. 7):

Thank you for incorporating the subregion-specific PLV analyses in the revised Fig. 6a and 6b. The dissociation of frequency-specific PLV patterns across subregions—particularly theta-band differences for AH, alpha for PH, and delta for the amygdala—provides important insights into the functional heterogeneity within the MTL. Considering prior studies linking low-frequency oscillatory coupling between these regions to language and memory functions, this is a particularly thought-provoking finding that offers new avenues for understanding the division of labor within the MTL.

Additionally, I would like to point out that all three MTL subregions—AH, PH, and the amygdala—show consistently higher PLV with axis-coding VTC channels across multiple frequency bands, particularly in the theta and alpha ranges (Fig. 6b and Revision Fig. 9b). This suggests that the increased phase synchronization is not limited to a specific MTL subregion but may reflect a more global network-level coordination pattern driven by axis-based coding in the VTC. It would be helpful if the authors could elaborate on whether this distributed PLV enhancement reflects a general broadcast mechanism for axis-coded information, or if it may still support region-specific coding transitions when combined with representational similarity patterns (e.g., Fig. 3h).

We thank the reviewer for the insightful observation and suggestions. We agree that the increased phase synchronization is not limited to a specific MTL subregion, but may instead reflect a broader, network-level coordination pattern driven by axis-based coding in the VTC. Accordingly, this distributed PLV enhancement may represent a general broadcast mechanism for axis-coding information. We have expanded our prior discussion as follows:

“We observed comparable region coding across MTL subregions (PH, AH, and amygdala). While these regions are associated with distinct cognitive and affective functions, extensive literature has shown that they share commonalities in the visual processing of faces and objects^{7,9,10,12,13,15,18}. Our previous

studies directly comparing face¹ and object⁷ coding in the amygdala and hippocampus also found qualitatively similar results between these regions, making our present findings highly consistent with the literature. This similarity may stem from a common input, such as the FG and/or ITG, as supported by the similar functional connectivity of the amygdala and hippocampus with the VTC (Fig. 6, Fig. 7). While there were differences in information flow between VTC and MTL subregions (Fig. 3h), as well as dissociations in frequency-specific PLV patterns across MTL subregions (Fig. 6a, b), it is important to note that the increase in phase synchronization for axis-coding channels was not confined to any single MTL subregion (Fig. 6b). Instead, this widespread PLV enhancement may reflect a more global, network-level coordination pattern driven by axis-based coding in the VTC. These findings suggest that axis-based coding in the VTC may serve as a unifying mechanism for broadcasting perceptual information across the MTL, facilitating integrated memory representations. Rather than operating in isolation, MTL subregions may be dynamically coordinated by shared input from feature-coding regions, allowing for distributed yet coherent processing of visual information.”

15. We thank the reviewer for the insightful observation and suggestions. We agree with the reviewer’s interpretations regarding Fig. 6. We have included the following discussion in the revised manuscript: “The FG and ITG exhibited stronger feedforward Granger causality (GC) to the AH in the low-frequency range, suggesting that low-frequency oscillations may serve as a mechanism for transmitting visual information from the VTC to the MTL during object perception. Interestingly, this effect was more pronounced in the spectral GC analysis (Fig. 6d) than in the time-resolved GC analysis (Fig. 6e). This discrepancy may reflect differences in the sensitivity of these methods—spectral GC is better suited for detecting sustained frequency-specific interactions, whereas time-resolved GC emphasizes transient, temporally localized effects. Additionally, the feedforward influence may be more specific to the AH, with weaker effects in other MTL subregions such as the PH or amygdala. These findings highlight the importance of combining multiple analytical approaches and considering subregional specificity within the MTL when examining directed functional connectivity in the human brain.”

Thank you for providing the subregion-specific breakdown of the GC analysis. I find the discussion particularly thoughtful and appropriately cautious in interpreting the frequency- and region-specific patterns. The distinction between spectral and time-resolved GC is especially helpful, and the

identification of stronger feedforward GC to the AH in the low-frequency band aligns well with the observed representational similarity. I agree that these converging lines of evidence strengthen the interpretation that the AH plays a distinct role in mediating visual–semantic integration between the VTC and MTL.

We thank the reviewer again for the suggestions and comments.

16. We thank the reviewer for the suggestions and agree with the reviewer’s insights. In the revised manuscript, we have analyzed spike–field phase consistency in the theta frequency range (Supplementary Fig. 9a-d). While we found that MTL region-coding neurons also fired spikes that were phase-locked to theta oscillations (4–8 Hz) in the VTC (Supplementary Fig. 9d)—supporting the critical role of theta oscillations in MTL processing, as noted by the reviewer—the pairwise phase coherence (PPC) of the MTL spike–VTC iEEG pairs was not significantly greater for in-region stimuli compared to out-region stimuli in the theta range (Supplementary Fig. 9d). This effect was only significant in the gamma-frequency range (Fig. 7e), suggesting that gamma oscillations may play a dominant role in MTL neurons during feature coding of visual objects. We have clarified this point in the revised manuscript and further discussed how this finding aligns with the existing literature.

Thank you for conducting the additional analyses of spike–field coupling in the theta frequency range and for presenting the detailed results in Supplementary Fig. 9. I appreciate the effort to address the potential role of lower frequency oscillations in VTC–MTL interactions. The finding that MTL region-coding neurons exhibit theta-phase locking to VTC activity, although not significantly modulated by stimulus region, is nonetheless informative. It supports the general role of theta oscillations in MTL coordination, while also highlighting the functional specificity of gamma-band coupling in visual object processing. Overall, I find your additions and discussion to be carefully reasoned and informative.

We thank the reviewer again for the suggestions and comments.

17. We thank the reviewer for the insightful questions and comments. First, we would like to clarify that the time window (0.1–0.6 s relative to stimulus onset) used for the iEEG analysis was determined based

on the temporal profile of the response (Fig. 2e). To ensure the robustness of our findings, we replicated the main analyses using a longer time window (0–1 s relative to stimulus onset) and observed consistent results. As the reviewer noted, the MTL spike–VTC field coupling (Fig. 7d) occurs at a late latency (~400 ms), which aligns with the spiking profile of MTL neurons observed in our previous studies 1,13,31–33. Accordingly, we applied a different time window (0.25–1.25 s relative to stimulus onset) when analyzing single-neuron activity in the current study. Therefore, the time windows applied to both the iEEG and single-neuron analyses were optimized based on their respective temporal response profiles. We have included the following discussion in the revised manuscript:

Thank you for the detailed and thoughtful response. I appreciate the authors' clarification regarding the rationale behind the use of distinct time windows for the iEEG and single-neuron analyses, and their effort to ensure robustness across temporal windows. The alignment of the late spike–LFP coupling with previous findings on MTL neuronal latencies is convincing and appropriately contextualized.

I also find the authors' interpretation—that this late coupling reflects higher-level integrative processes such as memory retrieval or semantic association—well justified. The proposed link between the late-onset axis coding in the AH and the delayed spike–field synchronization is compelling, even if a direct comparison is not feasible due to differences in signal source.

Overall, the revised discussion substantially improves the interpretability and coherence of the findings

We thank the reviewer again for the suggestions and comments.

18. We thank the reviewer for the suggestions. In Fig. 7a-c, we presented the spike–field coupling between MTL region-coding neurons and VTC channels, separately for the FG and ITG. In the revised manuscript, we further included time-resolved PPC analyses for the FG and ITG (see the newly added Supplementary Fig. 9e). Notably, FG axis-coding channels exhibited stronger synchronization with MTL region-coding neurons compared to ITG channels, indicating that the observed effects were primarily driven by FG channels. This finding aligns with the RSA results in Fig. 3h, which showed greater representational similarity between the FG and MTL. We have incorporated this result into the revised manuscript:

The new visualization of in-region stimuli along the most-preferred axis of phase-locked VTC channels (Fig. 7f, g) is informative and nicely illustrates the proposed functional link between VTC feature encoding and MTL region coding. The clustering of responses along a specific tuning axis is particularly compelling. That said, the example in Fig. 7g (ITG channel) appears to show stimulus distributions that may be more category-dependent, as stimuli from the same object category seem to cluster together along the axis. This raises the possibility that the axis-coding dimension in this case may reflect categorical distinctions more than continuous visual features. It may be worth discussing this point in more detail, and with greater caution, especially given the potential differences in coding strategies across VTC subregions. A more explicit analysis of the visual properties encoded by these axes (e.g., shape, category, semantics) could help clarify their representational content.

Additionally, the term "upstream VTC regions" might be potentially misleading, depending on the intended direction of processing; a more neutral term such as "VTC feature-coding regions" might improve clarity.

We thank the reviewer again for the helpful suggestions and comments. We agree that the example in **Fig. 7g** (from an ITG channel) shows stimulus clustering that appears more category-dependent, suggesting that in some cases, axis-coding dimensions may partially reflect categorical distinctions. This is not unexpected, for two reasons: (1) region coding inherently encompasses and explains category selectivity ⁷ and there are coding regions that contain stimuli primarily from a single category; and (2) object images from the same category often share similar visual features, which are preserved along the coding axes, thereby leading to their clustering along those axes. However, this observation highlights the diversity of representational content across VTC subregions and the possibility that axis coding can span both lower-level visual features and higher-level categorical structure. Therefore, we have included the following discussion in the revised manuscript:

“Axis coding

While our axis-coding framework primarily captures continuous visual feature dimensions, we note that in some cases, these axes may align more closely with categorical distinctions. For example, the stimulus distribution shown in **Fig. 7g** suggests that certain categories cluster along the preferred axis of the ITG channel, raising the possibility that axis coding may, in some instances, reflect categorically meaningful groupings. This could result from shared visual features within categories (e.g., similar

shapes or parts) or from the progressive transformation of perceptual representations into more abstract, category-like formats along the VTC hierarchy. On the other hand, it is worth noting that there was parametric variation along given axes within a single category, suggesting that axis coding preserves fine-grained visual differences even among exemplars of the same category. For instance, stimuli from the same object class may appear distributed along the axis in a gradient-like fashion, reflecting systematic changes in shape, size, or configuration (Fig. 3a, b). This supports the notion that axis coding does not merely represent categorical labels but captures a continuous feature space that underlies both within- and between-category structure. Together, these observations highlight the dual nature of axis coding in the VTC: while it may incidentally align with categorical groupings due to feature regularities, it fundamentally reflects graded visual dimensions that enable nuanced object discrimination. Future work could apply feature decomposition analysis to further elucidate the representational content of these axes and to examine the diversity of this content across VTC subregions.”

In addition, we conducted explicit analyses of the encoded axes (**Supplementary Fig. 3; Supplementary Fig. 4**). Specifically, to characterize the feature axes encoded by axis-coding channels, we performed PCA on the visual features of our stimuli, deriving orthogonal feature dimensions (principal components, PCs). The first three PCs corresponded to well-established object dimensions: natural–artificial, spiky–stubby, and animate–inanimate (**Supplementary Fig. 3a**), consistent with previous studies⁴. We next examined how the neural tuning axes aligned with these feature dimensions using Pearson correlation. Across axis-coding areas (FG, ITG, and AH), most tuning axes were significantly correlated with the first three PCs (**Supplementary Fig. 3b**). Notably, the first PC (natural–artificial) showed the strongest and most frequent alignment (FG: 60.00%; ITG: 56.52%; AH: 75.00%), followed by the third PC (animacy; FG: 28.24%; ITG: 21.74%; AH: 25.00%). In contrast to findings in monkey IT cortex⁴, human axis-coding channels showed weaker and less frequent alignment with the second PC (spiky–stubby), particularly in AH (0%). These results were replicated in a separate dataset (**Supplementary Fig. 4**), further supporting the robustness of these feature-to-axis mappings. Future work should further clarify the representational content of these axes, particularly in terms of continuous versus categorical coding, and examine its diversity across VTC subregions.

Lastly, we agree with the reviewer and have replaced “upstream VTC areas” with “VTC feature-coding areas” in the revised manuscript.

Reply to comments from Reviewer 4

I would like to express my gratitude to the authors for their comprehensive rewrite of both the Abstract and Introduction, as well as for updating Figure 1 and its legend in response to my request. These changes have significantly enhanced the clarity of the paper. Consequently, I believe this paper should be accepted for publication.

Once again, we thank the reviewer for the expert and constructive comments.

References

- 1 Cao, R. *et al.* Feature-based encoding of face identity by single neurons in the human amygdala and hippocampus. *Nature Human Behaviour* (2025).
- 2 Cao, R. *et al.* A neural computational framework for face processing in the human temporal lobe. *Current Biology* **35**, 1765-1778.e1766 (2025). <https://doi.org/https://doi.org/10.1016/j.cub.2025.02.063>
- 3 Kriegeskorte, N. *et al.* Matching Categorical Object Representations in Inferior Temporal Cortex of Man and Monkey. *Neuron* **60**, 1126-1141 (2008). <https://doi.org/http://dx.doi.org/10.1016/j.neuron.2008.10.043>
- 4 Bao, P., She, L., McGill, M. & Tsao, D. Y. A map of object space in primate inferotemporal cortex. *Nature* **583**, 103-108 (2020). <https://doi.org/10.1038/s41586-020-2350-5>
- 5 Lin, T.-Y. *et al.* 740-755 (Springer International Publishing).
- 6 Manns, J. R., Hopkins, R. O. & Squire, L. R. Semantic Memory and the Human Hippocampus. *Neuron* **38**, 127-133 (2003). [https://doi.org/https://doi.org/10.1016/S0896-6273\(03\)00146-6](https://doi.org/https://doi.org/10.1016/S0896-6273(03)00146-6)
- 7 Cao, R. *et al.* A neuronal code for object representation and memory in the human amygdala and hippocampus. *Nature Communications* **16**, 1510 (2025). <https://doi.org/10.1038/s41467-025-56793-y>
- 8 Kravitz, D. J., Saleem, K. S., Baker, C. I., Ungerleider, L. G. & Mishkin, M. The ventral visual pathway: an expanded neural framework for the processing of object quality. *Trends in cognitive sciences* **17**, 26-49 (2013). <https://doi.org/10.1016/j.tics.2012.10.011>
- 9 Kreiman, G., Koch, C. & Fried, I. Category-specific visual responses of single neurons in the human medial temporal lobe. *Nat Neurosci* **3**, 946-953 (2000).
- 10 Quian Quiroga, R., Reddy, L., Kreiman, G., Koch, C. & Fried, I. Invariant visual representation by single neurons in the human brain. *Nature* **435**, 1102-1107 (2005). https://doi.org/http://www.nature.com/nature/journal/v435/n7045/supinfo/nature03687_S1.html
- 11 Quian Quiroga, R. Concept cells: the building blocks of declarative memory functions. *Nature Reviews Neuroscience* **13**, 587 (2012). <https://doi.org/10.1038/nrn3251>
- 12 Rutishauser, U. *et al.* Representation of retrieval confidence by single neurons in the human medial temporal lobe. *Nat Neurosci* **18**, 1041-1050 (2015).
- 13 Reber, T. P. *et al.* Representation of abstract semantic knowledge in populations of human single neurons in the medial temporal lobe. *PLOS Biology* **17**, e3000290 (2019). <https://doi.org/10.1371/journal.pbio.3000290>
- 14 Rutishauser, U., Reddy, L., Mormann, F. & Sarnthein, J. The Architecture of Human Memory: Insights from Human Single-Neuron Recordings. *The Journal of Neuroscience* **41**, 883 (2021). <https://doi.org/10.1523/JNEUROSCI.1648-20.2020>
- 15 Wang, S., Mamelak, A. N., Adolphs, R. & Rutishauser, U. Encoding of Target Detection during Visual Search by Single Neurons in the Human Brain. *Current Biology* **28**, 2058-2069.e2054 (2018). <https://doi.org/https://doi.org/10.1016/j.cub.2018.04.092>
- 16 Quian Quiroga, R., Kreiman, G., Koch, C. & Fried, I. Sparse but not ‘Grandmother-cell’ coding in the medial temporal lobe. *Trends in Cognitive Sciences* **12**, 87-91 (2008). <https://doi.org/https://doi.org/10.1016/j.tics.2007.12.003>
- 17 Pennington, J., Socher, R. & Manning, C. 1532-1543 (Association for Computational Linguistics).

- 18 Rey, H. G. *et al.* Encoding of long-term associations through neural unitization in the human medial temporal lobe. *Nature Communications* **9**, 4372 (2018). <https://doi.org/10.1038/s41467-018-06870-2>

Reply to comments from Reviewer 1

the authors have answered all my comments and questions.

the manuscript is now in excellent shape.

I noted one typo: a verb (something like 'viewed') is missing in the second line of the section: 'neural responses to visual objects in the VTC and MTL'

Once again, we thank the reviewer for the expert and constructive comments. We have also corrected this typo in the revised manuscript.

Reply to comments from Reviewer 3

The authors have carefully addressed the previous comments, and the additional analyses have meaningfully advanced the evaluation of the proposed computational framework. Nevertheless, the responses have also revealed certain important operational issues concerning model-fitting metrics, and a few critical points remain unclear. Additional comments are therefore provided below.

Once again, we thank the reviewer for the expert and constructive comments.

Major Points:

We thank the reviewer for raising this question, and we apologize for not explaining it thoroughly in our previous response. We implemented a cross-validation procedure to evaluate model performance and calculate statistical significance. The previously reported R2 was based on the held-out test dataset (50%). However, to determine which DNN layer best models the neural responses—in other words, which set of features best fits the data—it is more reasonable to estimate R2 using the full dataset, as the final axis-coding models were trained on all available data. This approach not only increases the amount of data but also includes the training set, which naturally results in a higher R2.

While we acknowledge that a large difference in performance between training and testing sets may suggest overfitting, this did not affect our DNN layer selection. Both the testing dataset (previous Supplementary Fig. 2a) and the full dataset (current Supplementary Fig. 2a) consistently indicated that layer pool5 yielded the best model fit. It is also worth noting that pool5 did not have the highest dimensionality among the layers, making it less likely to be the most overfitted.

We thank the reviewer for pointing this out. This apparent discrepancy likely arises from differences in what is being measured. Specifically, Supplementary Fig. 2a shows R2 values for all visually responsive channels, calculated using the full dataset. In contrast, Supplementary Fig. 2b and Fig. 3d show the strength of axis coding, quantified as the correlation coefficient between the observed and predicted responses in the test dataset. The predicted responses were generated using models trained only on the training dataset (see Methods for details). It is likely that the models fit the data equally well across channels, but the inherent predictability varied—reflecting differences in data consistency. In summary,

R² values reflect how well the model explains neural responses and are used to compare relative model fits across feature layers, whereas the strength of axis coding reflects generalization performance.

Thank you for the clarification regarding the R² analysis. However, the current approach still raises concerns regarding potential overfitting and the interpretability of model comparisons.

As described, the R² values were estimated using models trained and tested on the full dataset, which likely inflates the R² values, particularly for the more flexible PLS models. Notably, much lower R² values (~0.1) were previously reported when evaluated on held-out data, in contrast to ~0.6 on the full dataset. This discrepancy suggests substantial overfitting and raises questions about the appropriateness of using full-dataset R² as an indicator of model generalization.

It is understood that the authors intentionally distinguish between model fit (as assessed by full-dataset R²) and generalization (as assessed by test-set correlation coefficients), and this clarification is appreciated. However, the fundamental problem is that they still use the fitting-based R² for comparisons even in situations where generalization performance should be evaluated. In particular, Supplementary Fig. 2d directly compares PLS and ANOVA models based on their respective R² values. Given the greater flexibility of the PLS model, such comparisons may inadvertently favor it over the more constrained ANOVA model, irrespective of actual generalization performance.

To ensure a fair and interpretable comparison of model performance, it would be preferable to evaluate both models on held-out test data or, at the very least, to report cross-validated R² values. Presenting both test-set and full-dataset R² values in parallel would enhance transparency, provide insight into the degree of overfitting, and better align with standard practices in the literature (e.g., Yamins et al., PNAS, 2014; Elmoznino et al., PLOS Comput Biol, 2024).

Importantly, the authors' additional RSA analyses provide multifaceted evidence that axis-based models derived from DNN features outperform the semantic model (Revision Fig. 1). This is a valuable contribution that is not undermined by the above concerns. Although, the absence of category (ANOVA) model in the RSA analyses appears to be a limitation. Moreover, as I will discuss later, the claim that category models cannot be applied may not be entirely accurate.

We thank the reviewer for the additional comments. First, as suggested, we have now included R² values from both the test dataset and the full dataset in the revised **Supplementary Fig. 2a** to enhance reporting

transparency. Notably, both the held-out test dataset and the full dataset consistently indicated that layer pool5 yielded the best model fit.

Supplementary Fig. 2. (a) Goodness-of-fit (R^2) of the partial least squares (PLS) regression with deep neural network (DNN) features for each DNN layer. The layer with the highest R^2 was selected for further analysis. Error bars denote \pm SEM across channels. R^2 was calculated using both the full dataset (solid line) and a held-out test dataset (dotted line; see **Methods**). Both datasets indicated that layer pool5 provided the best fit to the neural responses.

Second, we have now included a comparison with the ANOVA model using a held-out test dataset in the revised **Supplementary Fig. 2d**. This ensures that the comparison is transparent while acknowledging the methodological differences between the two models. We have clarified this point explicitly in the revised **Supplementary Fig. 2d** legend. It is important to note that ANOVA does not involve training/testing procedures or cross-validation; its R^2 is computed directly from the full dataset and thus does not provide an estimate of generalization performance.

Supplementary Fig. 2. (d) R^2 for the axis-coding PLS model and category-coding ANOVA model. Each circle represents a visually responsive channel, and error bars denote \pm SEM across channels. Asterisks indicate a significant difference between models using a two-tailed paired t -test. #: $P < 0.1$, and ****: $P < 0.0001$. (left) R^2 calculated using the full dataset. (right) R^2 calculated using a held-out test dataset. It is important to note that ANOVA does not involve training/testing procedures or cross-validation; its R^2 was computed directly from the full dataset.

Lastly, we thank the reviewer for recognizing the contribution of our RSA analyses. Regarding the category (ANOVA) model, it is important to note that ANOVA is categorical by design, and categorical labels are not well suited for computing dissimilarity values in RSA (please refer to our response below for more details). Because RSA requires continuous variables to calculate correlations, we did not test the ANOVA model in our previous analysis (but we explored this analysis in this revision). We appreciate the opportunity to clarify this conceptual distinction, and we have further fleshed out this analysis below.

We thank the reviewer for pointing this out, and we agree that differences in model fitting can be influenced by differences in input dimensionality. To control for this, we additionally calculated R^2 using a one-dimensional feature (the first PLS component), ensuring that both the PLS model and the ANOVA model had one-dimensional input. Although, as expected, R^2 was reduced when we restricted the PLS input dimensionality, the PLS model still showed a better fit even under this constraint for both the FG (axis: 0.22 ± 0.06 [mean \pm SD]; category: 0.17 ± 0.087 ; two-tailed paired t -test: $t(200) = 54.84$, $P < 10^{-20}$) and ITG (axis: 0.20 ± 0.04 ; category: 0.12 ± 0.046 ; $t(140) = 56.25$, $P < 10^{-20}$). We have included the following results in the revised manuscript:

Thank you for performing the additional analysis using the first PLS component. This successfully controls for input dimensionality and shows that the PLS model still outperforms the ANOVA model under this constraint, which is informative. However, the comparison remains based on full-dataset R^2 values, which are sensitive to overfitting. Therefore, even when dimensionality is matched, such comparisons would be more informative and interpretable if based on held-out or cross-validated R^2 values. It is particularly important to minimize the influence of potential overfitting when evaluating model performance. This is especially true in the present case, where, after controlling for other factors, the R^2 values of different models become similar—highlighting the need for careful interpretation. Without this, it remains difficult to assess whether the observed advantage of the PLS model reflects genuine generalization performance or simply better fit to the training data. This issue still warrants further consideration.

In addition to model comparisons, a more systematic use of RSA analyses applied to both the VTC and MTL could help clarify the respective modes of visual object coding in these regions of the human temporal lobe. (This point is further elaborated later in the review.)

We thank the reviewer for the additional comments. We have now calculated R^2 using the first PLS component with a held-out test dataset. The PLS model still showed a better fit for the held-out test dataset in both the FG (axis: 0.27 ± 0.042 [mean \pm SD]; category: 0.17 ± 0.087 ; two-tailed paired t -test: $t(127) = 9.56$, $P = 1.24 \times 10^{-16}$) and ITG (axis: 0.25 ± 0.036 ; category: 0.12 ± 0.046 ; $t(45) = 10.97$, $P = 2.66 \times 10^{-14}$). This control analysis further strengthens the conclusion that axis-based models provide a more accurate account of neural responses than category models in the human VTC.

It is important to note that while the highest DNN layers may correspond more closely to semantic representations, the VTC may more closely align with intermediate DNN layers (see Supplementary Fig. 2a for details), which retain rich visual information without abstracting fully to semantic categories. This intermediate-level feature representation in both VTC and DNNs may explain why the axis model—derived from DNN features—is more successful than a coarse category model in capturing VTC activity. Thus, the relatively low R^2 values of the category model likely reflect both the limitations of discrete category labels and the nature of VTC coding itself.

According to Supplementary Fig. 2a, the layer pool5 appears to provide the best fit in terms of R^2 . If this is the case, pool5—which is located immediately before the fully connected (fc) layer—would be more appropriately described as the “penultimate layer” or a “high-level semantic layer” rather than an “intermediate layer”. Thus, the current description may be somewhat inaccurate.

Additionally, while it may not be entirely clear whether this point belongs here, I would like to note that in Supplementary Fig. 2a (R^2), there is a clear difference between res4b22, res5, and fc, whereas in Supplementary Fig. 2b (Pearson’s r), these layers show minimal differences. This likely reflects fundamental differences between the two metrics—for example, their sensitivity to bias and scaling. Although the authors have already explained the conceptual differences between these two measures earlier in the manuscript, a brief clarification of how these differences manifest in Supplementary Fig. 2a and 2b would be appreciated and may help readers better interpret the results. Furthermore, it would be helpful if the authors could explicitly state which ResNet architecture (e.g., ResNet-50, ResNet-101, etc.) was used in the analysis, as this information is currently not clearly specified.

We thank the reviewer for the additional comments and suggestions. First, we agree with the reviewer’s point regarding the description of layer pool5, and we apologize for any confusion caused.

Second, we agree with the reviewer, and we have added the following clarification to the legend of **Supplementary Fig. 2**:

“It is worth noting that panel (a) shows R^2 values for all visually responsive channels, calculated using either the full dataset or a held-out test dataset, whereas panel (b) depicts the strength of axis coding, quantified as the correlation between observed and predicted responses in the held-out test dataset. The predicted responses in (b) were generated using models trained only on the training dataset (see **Methods** for details). The apparent discrepancy—where layers such as res4b22, res5c, and fc differ in R^2 but show little difference in Pearson’s r —likely reflects fundamental differences between these metrics. R^2 is sensitive to both bias and scaling, capturing how well the model explains variance in the dataset, whereas the correlation coefficient emphasizes generalization performance and is less sensitive to absolute scaling differences. In summary, R^2 values are most informative for comparing relative model fits across feature layers, while correlation coefficients (i.e., the strength of axis coding) reflect the models’ ability to generalize to unseen data.”

Lastly, in the revised manuscript, we have clarified that our model is ResNet-101.

We thank the reviewer for the thoughtful comments and suggestions. We agree that incorporating alternative models that account for lexical or semantic distances would be valuable for understanding which features best explain VTC responses. As suggested by the reviewer, we extracted word-label embeddings for the 50 included categories using the Global Vectors for Word Representation (GloVe) 17 model. We then compared the correspondence between VTC neural responses and word embeddings versus visual features using RSA. We found that visually responsive VTC channels (Revision Fig. 1a) exhibited a higher, though not statistically significant, correspondence with visual features (Spearman's $\rho = 0.23$) compared to word embeddings (Spearman's $\rho = 0.13$; permutation test against the null distribution of differences: $P = 0.13$). This pattern held when considering the entire population of VTC channels (Revision Fig. 1b), suggesting that category-based coding does not better explain VTC responses at the population level. As expected, channels classified as axis-coding (Revision Fig. 1c) demonstrated a significantly higher correspondence with visual features (Spearman's $\rho = 0.39$) than with word embeddings (Spearman's $\rho = 0.22$; $P = 0.005$). Notably, the same pattern was observed even in category-selective channels (Revision Fig. 1d). Together, these results support the conclusion that visual models provide a better explanation of VTC neural responses than word-based category models.

We thank the reviewer for the clarification and we agree with the reviewer. In the revised manuscript, we have consolidated the comparison between the PLS and ANOVA models (see response above). Additionally, as in the previous Revision Fig. 6, we further compared axis-based and lexical/semantic-based coding using RSA (as categorical labels used in the ANOVA model could not be used to construct a dissimilarity matrix). Please refer to our response to the question above.

Thank you for incorporating the analysis based on word embeddings and comparing it with visual feature models using RSA. This is a valuable addition that further clarifies the nature of neural coding in the VTC. Applying such analyses to rare and valuable human neurophysiological data is especially meaningful, particularly given that similar analyses in non-human primates might face limitations in interpretability or generalizability.

It is notable that even category-selective channels aligned better with visual features than with word embeddings, supporting the idea that category selectivity in the VTC may emerge from continuous tuning to visual features. It may be helpful to relate this explicitly to the results from the ANOVA-based category model, which assumes a discrete structure. Clarifying this relationship would help unify the findings across the modeling approaches.

I also found it somewhat difficult to fully understand the statement that an ANOVA model “could not be used to construct a dissimilarity matrix.” Initially, I thought this might reflect the limitation that each stimulus was presented only once, which would prevent reliable estimation of stimulus-level responses. However, looking at Supplementary Fig. 5 (where a neural dissimilarity matrix appears to have been constructed for the MTL), this may not be the case. If trial-level responses for each stimulus can be obtained, it should be possible to construct a category-based idealized dissimilarity matrix under the ANOVA model—for example, one in which within-category pairs are coded as 0 and between-category pairs as 1. If stimulus-level analyses are indeed infeasible in the VTC, one potential workaround would be to construct such matrices based on coarser categories, as in the R^2 analyses with broader category groupings. In fact, systematically comparing different category granularities could itself be informative.

More broadly, if both R^2 -based model fitting and RSA were systematically applied to the VTC and MTL using visual features (e.g., from DNNs), word embeddings (e.g., GloVe), and category-based models (at various levels of granularity), it may help clarify the computational basis of axis versus region coding, and shed further light on how visual representations are transformed into more abstract, conceptual formats along the human temporal lobe.

We thank the reviewer for the additional comments. First, we would like to clarify that the category-selective channels were identified using the ANOVA-based category model (which is why we examined these channels), and we thus further linked this ANOVA-based model to visual feature and semantic coding.

Second, while it is true that, in principle, one could construct an idealized dissimilarity matrix using categorical labels, such a matrix would consist of binary values (0 for within-category and 1 for between-category). This structure does not provide the continuous variability needed for computing meaningful correlations in RSA, as it would yield a matrix with 0s along the diagonal blocks and 1s elsewhere. In other words, the categorical nature of ANOVA prevents it from capturing graded

similarities or differences across stimuli, which are central to RSA analyses. For this reason, we did not apply the ANOVA model within the RSA framework in our previous revision. We appreciate the opportunity to clarify this conceptual limitation.

However, here, we explored this analysis by correlating the idealized RDM from the ANOVA model (**Revision Fig. 1a**) with VTC (**Revision Fig. 1b**) and MTL (**Revision Fig. 1c**) channels, and compared these results with correlations obtained using visual and lexical representations. We found that in the VTC, visual features explained neural responses better than lexical/word features or categorical labels (permutation test: visual versus ANOVA category: $P < 0.0001$; word versus ANOVA category: $P = 0.21$). In the MTL, lexical/word features provided a better explanation of neural responses (visual versus ANOVA category: $P = 0.55$; word versus ANOVA category: $P = 0.042$). These results are consistent with our proposed computational pathway.

Revision Fig. 1. Comparison of visual, lexical, and categorical representations in the VTC and MTL. **(a)** Idealized RDM constructed from ANOVA categorical labels. **(b)** VTC channels. **(c)** MTL channels. Black arrow: observed. Gray distribution: permuted/null.

Lastly, we agree with the reviewer that it is important to examine the coding properties of different features (visual, semantic/word embeddings, and ANOVA categorical) along the ventral visual pathway (VTC and MTL) using multiple metrics. Given the focus of the present study, we plan to explore these questions in future work. In fact, we have already been conducting systematic analyses addressing these questions. For example, we have compared visual and ANOVA categorical models in the MTL (Wang Y, Cao R, Wang S. *Encoding of Visual Objects in the Human Medial Temporal Lobe*. The Journal of Neuroscience. 2024;44(16):e2135232024.), and we are currently comparing visual and semantic models in the VTC and MTL using both RSA and PLS regression on the same iEEG dataset with additional

patients (Wang Y, Brunner P, Willie JT, Cao R, Wang S. *Neural computations of visual, semantic, and memorability features in the human brain*. PsyArXiv). We believe that these dedicated analyses merit a separate publication.

We thank the reviewer for the suggestions. The reviewer has accurately captured our proposed computational framework, and we agree with the insightful conceptualization of “a hierarchy from axis coding (low-level), to region-based coding (intermediate), to label-based category coding (high-level), with the corresponding models being the PLS model, semantic embedding models, and the ANOVA model, respectively”. Furthermore, the reviewer is correct that between-region comparisons are challenging, whereas within-region comparisons using data from a single measurement modality are more feasible. Accordingly, we used iEEG responses for model comparisons in the VTC, and single-unit responses for model comparisons in the MTL. Specifically, as shown in Revision Fig. 1, VTC neural responses were better explained by a visual model than by a lexical/semantic model. In contrast, as shown in our previous Revision Fig. 6, MTL region-coding neurons exhibited greater representational similarity with semantic representations than with visual representations (similar results were obtained using the GloVe model). These findings support the proposed transition in coding models from the VTC to the MTL. It is worth noting that the ANOVA model could not be evaluated using RSA, as the categorical labels it relies on cannot be used to construct a dissimilarity matrix. Furthermore, while explanatory power (R^2) could not be calculated for semantic features—since no specific predictive model was assumed—RSA is more appropriate in this case. Nonetheless, the R^2 based analyses (see above) further support visual, rather than category-based, coding in the VTC.

Thank you for the clear and thoughtful response. I appreciate the structured articulation of the proposed hierarchy, from axis to region-based to label-based coding, and the associated use of PLS, semantic embedding, and ANOVA models. This framework offers a helpful conceptual lens through which to interpret the observed transition from VTC to MTL.

Regarding the point that the ANOVA model could not be used for RSA, it would be helpful to clarify the nature of this limitation. As mentioned in the previous section, if stimulus-level neural responses are available, it should be possible to construct an idealized dissimilarity matrix. I wonder whether this

approach was considered, and if so, whether any specific challenges prevented its application in this case.

We thank the reviewer again for the helpful suggestions and comments. Please refer to our response to the question above.

We thank the reviewer for this insightful and constructive feedback. We appreciate the reviewer's recognition of our conceptual framework; and we agree that caution is needed when interpreting the transformation from VTC to MTL as "semantic". We acknowledge that such convergence could be the result of a general hierarchical integration process, similar to those observed along the ventral visual stream (e.g., V4 to IT), rather than a transformation uniquely tied to semantics. We now highlight this alternative possibility in the revised Discussion, emphasizing that the apparent semantic structure in MTL may reflect the cumulative integration of lower-level visual features, and not necessarily a dedicated semantic operation: "An alternative interpretation of the VTC-to-MTL transformation is that the observed representational changes reflect a general hierarchical integration process, rather than a dedicated transformation. Similar to how receptive fields in early visual areas converge to form higher-order visual representations in regions such as V4 and IT, the emergence of region-based coding in the MTL may arise from the cumulative integration of visual features represented in the VTC. In this view, the clustering of neural responses in MTL—sometimes aligning with object categories—could result from overlapping visual feature selectivity, rather than an explicit coding of semantic content. This interpretation is consistent with known principles of hierarchical organization in sensory systems and suggests that apparent semantic structure may emerge naturally from the integration of complex visual information, even in the absence of explicit category representations. Our findings that region-based tuning in the MTL transcends strict category boundaries further support this view. Future work will be critical to determine whether the transformation from perceptual to conceptual representations in the MTL reflects purely integrative processes or additional mechanisms that encode abstract conceptual meaning." It is worth noting that our current data suggest that MTL region coding captures structured information that aligns more closely with semantic embeddings than with purely visual similarity, as supported by the RSA (please see above). While this correspondence is suggestive, we agree with the

reviewer that further studies are needed to disentangle whether the VTC-to-MTL transformation reflects uniquely semantic processing or a broader convergence process.

Thank you for explicitly addressing the alternative interpretation that the transformation from VTC to MTL may reflect a general hierarchical integration process rather than dedicated semantic coding. This addition makes the Discussion (with Supplementary) more balanced and conceptually rich. In particular, I appreciate the clarification that apparent semantic structure may emerge naturally from the accumulation of visual feature selectivity, as seen in other parts of the ventral stream. The use of both RSA and R^2 -based analyses provides complementary evidence and helps clarify the extent to which MTL coding reflects semantic versus perceptual structure. This issue will indeed be an important direction for future work.

We thank the reviewer again for the insightful and constructive suggestions.

Minor points:

We thank the reviewer again for the suggestion. First, as expected, we found a pre-existing correlation between response amplitude (i.e., axis-coding strength) and latency before normalization ($r = -0.43$, $P = 9.77 \times 10^{-5}$). After normalization (i.e., response amplitude normalized to 1), this correlation was eliminated ($r = 0$, $P = 1$). Therefore, the observed latency gradient was independent of response magnitude.

Thank you for including this follow-up analysis. This strengthens the interpretation that latency differences reflect meaningful functional distinctions beyond mere signal strength.

We thank the reviewer again for the suggestion.

We thank the reviewer for the suggestion. We agree and have included example object images located near category boundaries to better illustrate how semantic structure emerges atop continuous visual similarity. A new supplementary figure has been added to the revised manuscript:

The addition of example images near category boundaries in the VTC feature space is very helpful. I found this figure particularly insightful and evocative. Since both Fig. 4a and Supplementary Fig. 6 depict essentially the same underlying feature space, this parallel is a strength, but it may also be perceived as slightly redundant. To make the figures more complementary, one option could be to color-code Supplementary Fig. 6 by the six broader super categories (e.g., animal, equipment, etc.), rather than by all 50 fine-grained categories. In this way, Fig. 4a would continue to emphasize semantic structure and category labels, while Supplementary Fig. 6 would highlight visual continuity and example stimuli, helping to more clearly differentiate the respective roles of the two figures.

We thank the reviewer again for the thoughtful suggestion. We would like to clarify that **Supplementary Fig. 6** is already color-coded by the broader super categories. The apparent fine-grained differences in color within each broader color category are used to distinguish the fine-grained categories nested within them. We have made this clearer in the figure legend to avoid any confusion:

“Colors indicate the broader super categories (e.g., animals, equipment, etc.). Variations within each broader color category represent the nested fine-grained categories.”

We thank the reviewer for this excellent suggestion. To further illustrate region-based feature representations in MTL neurons, we visualized object images that fell within the tuning regions of example region-based neurons (upper panels). Additionally, we generated word clouds to qualitatively depict the relative frequency of object categories represented by each neuron (bottom panels). These illustrations show that most object images encoded by Cell #586 (Fig. 4b) exhibit nested structures, while those represented by Cell #630 (Fig. 4c) feature radical shapes and small sizes. Notably, the clustered images spanned a variety of semantic categories. Furthermore, the number of images from each category within the tuning region varied, and many images from the same category were located outside the tuning region—suggesting that the clustering was driven more by visual features than by semantic meaning. Together, these results indicate that region-based tuning indeed transcends semantic category boundaries.

The addition of Fig. 4b and 4c, along with the word clouds, is highly effective in illustrating the specific nature of region-based tuning in MTL neurons. The descriptions such as “nested structure” and

“radical shape” offer compelling insight into how tuning appears to rely on visual features rather than semantic categories.

Interestingly, these visual characteristics might themselves reflect higher-order abstract concepts, which makes the findings particularly intriguing. It could be helpful to relate these qualitative examples to the quantitative analyses suggested above involving semantic categorization at multiple levels, as this would help ground the interpretation of tuning properties more firmly in the data.

We thank the reviewer again for the suggestion. We agree with the reviewer, and please refer to our response to the question above about semantic categorization at multiple levels.

We thank the reviewer for the insightful observation and suggestions. We agree that the increased phase synchronization is not limited to a specific MTL subregion, but may instead reflect a broader, network-level coordination pattern driven by axis-based coding in the VTC. Accordingly, this distributed PLV enhancement may represent a general broadcast mechanism for axis-coding information. We have expanded our prior discussion as follows:

Thank you for the thoughtful and well-integrated revision. The expanded discussion clarifies how widespread PLV enhancement across the MTL may reflect a common broadcast mechanism originating from axis-based coding in the VTC.

We thank the reviewer again for the suggestion.

We thank the reviewer again for the helpful suggestions and comments. We agree that the example in Fig. 7g (from an ITG channel) shows stimulus clustering that appears more category-dependent, suggesting that in some cases, axis-coding dimensions may partially reflect categorical distinctions. This is not unexpected, for two reasons: (1) region coding inherently encompasses and explains category selectivity 7 and there are coding regions that contain stimuli primarily from a single category; and (2) object images from the same category often share similar visual features, which are preserved along the coding axes, thereby leading to their clustering along those axes. However, this observation highlights the diversity of representational content across VTC subregions and the possibility that axis coding can

span both lower-level visual features and higher-level categorical structure. Therefore, we have included the following discussion in the revised manuscript:

The authors have provided a thorough and compelling discussion of the dual aspects of axis coding, capturing both continuous visual features and, in some instances, category-related structure. This interpretation is further supported by the analyses presented in Supplementary Fig. 3 and Fig. 7g, which suggest that axis-coding may capture not only continuous visual features but also aspects of categorical meaning.

To further strengthen this claim, it may be valuable to explore whether the encoded axes align with semantic dimensions using more explicit model-based approaches. For example, PLS regression could be used to predict axis directions from semantic or visual features, and semantic embeddings could serve as a basis for constructing representational dissimilarity matrices (RDMs) for RSA comparisons. Additionally, applying ANOVA models along the axis direction could help quantify the extent to which categorical information is captured. These approaches may help clarify the representational nature of axis coding across the VTC.

We thank the reviewer again for this valuable suggestion. Indeed, these are important directions for further analysis. In fact, we have examined these questions in a separate manuscript (Wang Y, Brunner P, Willie JT, Cao R, Wang S. *Neural computations of visual, semantic, and memorability features in the human brain*. PsyArXiv), using the same dataset (with a few additional patients). In that work, we applied PLS regression to construct axis-coding models for visual and semantic features and used semantic embeddings to build RDMs for RSA comparisons. We conducted a detailed characterization and comparison of visual and semantic models along the ventral visual pathway, including both the VTC and MTL. It is also worth noting that **Supplementary Fig. 3** of the current manuscript illustrates the alignment of axis-coding channels with model-based visual feature dimensions.

Reply to comments from Reviewer 3

I sincerely thank the authors for their persistent and careful efforts in addressing the concerns I raised. Their additional analyses have enhanced the transparency of the work and provided a reasonable interpretation of the valuable dataset. As a result, the overall quality and completeness of the manuscript have significantly improved.

Although there may be a difference in perspective regarding the assumptions underlying the RSA analyses, it is recognized that RSA was originally introduced here as one of multiple approaches to evaluate the plausibility of axis- and region-based coding schemes. In this regard, the conclusions of the present study are considered to be sufficiently supported by the current set of analyses.

Once again, we thank the reviewer for the expert and constructive comments.

Further comments are provided below.

We thank the reviewer for the additional comments. First, as suggested, we have now included R² values from both the test dataset and the full dataset in the revised Supplementary Fig. 2a to enhance reporting transparency. Notably, both the held-out test dataset and the full dataset consistently indicated that layer pool5 yielded the best model fit. Supplementary Fig. 2. (a) Goodness-of-fit (R²) of the partial least squares (PLS) regression with deep neural network (DNN) features for each DNN layer. The layer with the highest R² was selected for further analysis. Error bars denote \pm SEM across channels. R² was calculated using both the full dataset (solid line) and a held-out test dataset (dotted line; see Methods). Both datasets indicated that layer pool5 provided the best fit to the neural responses.

Thank you for performing the additional analysis. It is reassuring to see that the superiority of layer pool5 is consistently supported. However, when accounting for potential overfitting, its advantage appears to be less pronounced. The systematic patterns of layer dependence and regional differences observed in the test dataset—particularly given the small error bars—suggest that these effects are unlikely to be due to noise or random variability, but may instead reflect meaningful structure.

While the conclusion that pool5 offers the best overall fit appears robust, the relatively high R² values observed for intermediate layers (e.g., res3b3) in the test dataset may indicate that, in certain regions,

intermediate layers provide a better explanation of neural responses. Furthermore, the larger discrepancy between the full and test datasets at higher layers suggests that features from upper layers, while fitting the training data well, may have limited generalizability to novel data. This observation is particularly intriguing and may warrant further discussion. If space permits, it could be helpful to briefly note these points in the Supplementary Discussion. Of course, the final decision is up to the authors.

We thank the reviewer for the careful evaluation and insightful comments regarding the layer-dependent effects and potential overfitting patterns. We appreciate the observation that, while layer pool5 consistently provides the best overall fit, intermediate layers (e.g., res3b3) may better explain neural responses in certain brain regions, and that higher-layer features may show reduced generalizability to novel data. We agree that these findings highlight meaningful structure rather than random variability. In response, we have added a brief discussion in **Supplementary Discussion** to acknowledge these important points:

“When selecting the DNN layer for visual feature extraction (**Supplementary Fig. 2a**), we found that layer pool5 consistently provided the best overall fit to neural data across brain areas. However, intermediate layers (e.g., res3b3) also achieved relatively high R^2 values in the test dataset, particularly in certain brain areas. This pattern suggests that features from mid-level layers may, in some cases, better capture the representational structure of neural responses. Moreover, the larger discrepancy between the full and test datasets observed for higher layers indicates that these deeper representations, though highly predictive during training, may exhibit reduced generalizability to novel data.”

Second, we have now included a comparison with the ANOVA model using a held-out test dataset in the revised Supplementary Fig. 2d. This ensures that the comparison is transparent while acknowledging the methodological differences between the two models. We have clarified this point explicitly in the revised Supplementary Fig. 2d legend. It is important to note that ANOVA does not involve training/testing procedures or cross-validation; its R^2 is computed directly from the full dataset and thus does not provide an estimate of generalization performance.

Thank you for the additional analysis. While I appreciate the clarification that ANOVA, in its standard implementation, does not involve training/testing or cross-validation procedures, I would like to note that, in principle, cross-validation can be applied to ANOVA models as well, since they are linear models with discrete predictors. The risk of overfitting is generally lower compared to more flexible models such as PLS, but the distinction remains methodological rather than fundamental.

Regardless of the issue of training/testing, I find it noteworthy that the R^2 values from the ANOVA model appear higher in the fusiform gyrus (FG) than in the inferior temporal gyrus (ITG). The lack of difference between ANOVA and PLS models in FG may reflect the possibility that the relationship between axis coding and categorical boundaries is not necessarily consistent across regions. This could suggest that the nature of feature representation—and its alignment with categorical structure—differs between FG and ITG.

We appreciate and agree with the reviewer's clarification that, in principle, cross-validation procedures can also be applied to ANOVA models, given their linear nature with discrete predictors. We also agree that while the risk of overfitting is generally lower for such models compared to more flexible approaches (e.g., PLS), the distinction is indeed methodological rather than fundamental.

Regarding the observed differences in R^2 values across brain regions, we concur that the higher ANOVA-based R^2 values in the fusiform gyrus (FG) relative to the inferior temporal gyrus (ITG) may indicate regional variations in how neural representations align with categorical structure. The comparable performance in PLS models between the FG and ITG suggests that the relationship between axis coding and categorical boundaries may differ across cortical regions, reflecting potential distinctions in representational organization and coding strategies between the FG and ITG.

Lastly, we thank the reviewer for recognizing the contribution of our RSA analyses. Regarding the category (ANOVA) model, it is important to note that ANOVA is categorical by design, and categorical labels are not well suited for computing dissimilarity values in RSA (please refer to our response below for more details). Because RSA requires continuous variables to calculate correlations, we did not test the ANOVA model in our previous analysis (but we explored this analysis in this revision). We appreciate

the opportunity to clarify this conceptual distinction, and we have further fleshed out this analysis below.

Thank you for the conceptual clarification and the additional analyses. However, I believe the argument that “ANOVA is categorical by design, and categorical labels are not well suited for computing dissimilarity values in RSA (please refer to our response below for more details). Because RSA requires continuous variables to calculate correlations” may not be logically robust. The core objective of representational RSA is to assess the correspondence between the representational structure proposed by a model—whether continuous or discrete—and the neural similarity structure.

What the authors may have intended to emphasize is the following: when categorical labels are used to construct an idealized RDM, all within-category pairs receive a value of 0 and all between-category pairs a value of 1. As a result, such RDMs exhibit minimal variability across stimulus pairs, which limits the sensitivity and resolution of correlation-based comparisons with neural RDMs. In this sense, discrete RDMs may be at a disadvantage for RSA, not because they are inherently incompatible, but because they are less capable of capturing fine-grained similarity patterns across stimuli. Therefore, it would be more appropriate to describe this as a limitation rather than an incompatibility. This is precisely why combining RSA with other approaches such as R^2 -based model fitting can be advantageous in capturing complementary aspects of the neural representational structure.

We thank the reviewer for the insightful conceptual clarification and for highlighting the nuanced distinction between categorical and continuous model representations in RSA. As the reviewer rightly points out, RSA can, in principle, accommodate both continuous and discrete model structures. The reviewer’s interpretation is precise, and we fully agree that the limited variability in idealized RDMs reduces the sensitivity of correlation-based RSA to detect fine-grained similarities in neural patterns.

We thank the reviewer for the additional comments. We have now calculated R^2 using the first PLS component with a held-out test dataset. The PLS model still showed a better fit for the held-out test dataset in both the FG (axis: 0.27 ± 0.042 [mean \pm SD]; category: 0.17 ± 0.087 ; two-tailed paired t -test: $t(127) = 9.56$, $P = 1.24 \times 10^{-16}$) and ITG (axis: 0.25 ± 0.036 ; category: 0.12 ± 0.046 ; $t(45) =$

10.97, $P = 2.66 \times 10^{-14}$). This control analysis further strengthens the conclusion that axis-based models provide a more accurate account of neural responses than category models in the human VTC.

Does this correspond to the right panel of Supplementary Fig. 2d? The legend and the statistical results reported here seem inconsistent, so I would appreciate it if you could double-check this. It may also be helpful to revise the main text accordingly to ensure clarity and consistency. The final decision is entirely up to the authors.

We thank the reviewer for pointing this out. The above results do not correspond to the right panel of **Supplementary Fig. 2d**, which shows the comparison using the full PLS model with a held-out test dataset. To clarify, the main text and the left panel of **Supplementary Fig. 2d** used the full PLS model with the full dataset; the right panel of **Supplementary Fig. 2d** used the full PLS model with a held-out test dataset; the results shown in our previous revision used the first PLS component with the full dataset; and the results described above used only the first PLS component (controlling for input dimensionality) with a held-out test dataset. Notably, all analyses converged on the same conclusion.

We thank the reviewer for the additional comments. First, we would like to clarify that the category-selective channels were identified using the ANOVA-based category model (which is why we examined these channels), and we thus further linked this ANOVA-based model to visual feature and semantic coding. Second, while it is true that, in principle, one could construct an idealized dissimilarity matrix using categorical labels, such a matrix would consist of binary values (0 for within-category and 1 for between-category). This structure does not provide the continuous variability needed for computing meaningful correlations in RSA, as it would yield a matrix with 0s along the diagonal blocks and 1s elsewhere. In other words, the categorical nature of ANOVA prevents it from capturing graded similarities or differences across stimuli, which are central to RSA analyses. For this reason, we did not apply the ANOVA model within the RSA framework in our previous revision. We appreciate the opportunity to clarify this conceptual limitation.

However, here, we explored this analysis by correlating the idealized RDM from the ANOVA model (Revision Fig. 1a) with VTC (Revision Fig. 1b) and MTL (Revision Fig. 1c) channels, and compared these results with correlations obtained using visual and lexical representations. We found that in the

VTC, visual features explained neural responses better than lexical/word features or categorical labels (permutation test: visual versus ANOVA category: $P < 0.0001$; word versus ANOVA category: $P = 0.21$). In the MTL, lexical/word features provided a better explanation of neural responses (visual versus ANOVA category: $P = 0.55$; word versus ANOVA category: $P = 0.042$). These results are consistent with our proposed computational pathway.

While I understand the authors' rationale for avoiding categorical (binary) RDMs in RSA, I believe the argument could benefit from further clarification. The authors note that "the categorical nature of ANOVA prevents it from capturing graded similarities or differences," which is true in terms of continuous variability. However, the core aim of RSA is to evaluate the correspondence between the structure of representational spaces—regardless of whether they are continuous or discrete—and neural similarity matrices. From this perspective, the discrete structure of category-based models does not inherently disqualify them from RSA analyses.

As I mentioned earlier, a more precise framing might be that binary category-based RDMs offer limited resolution, with all within-category pairs assigned 0 and all between-category pairs assigned 1. This coarse granularity could reduce sensitivity in correlational analyses, particularly when comparing models. Nonetheless, this should be regarded as a limitation in discriminability or resolution, rather than a categorical inapplicability.

In this regard, I appreciate the authors' effort to conduct RSA, which I believe adds important value to the manuscript. However, regarding Revision Fig. 1a: if the "Idealized RDM" was constructed based on ANOVA categorical labels, one would expect to see square-shaped diagonal blocks, each with a size corresponding to the number of stimuli within each category. In contrast, Fig. 1a currently shows nearly uniform values (presumably 1) across both within-category and between-category pairs, which does not appear to visually reflect the key structural features of an idealized category RDM. The revised results themselves are highly interesting, but since this point is not explicitly addressed in the current version of the manuscript, I leave it to the authors to decide whether and how to clarify or revise the presentation.

We thank the reviewer again for the additional comments and we fully agree with the reviewer. To clarify, in **Revision Fig. 1a**, there were indeed fifty 10-by-10 square-shaped diagonal blocks within the 500-by-500 RDM, corresponding to 50 object categories with 10 images per category. The within-category (0) and between-category (1) values were not identical (although the vast majority of values

were 1, giving the appearance of uniformity). Furthermore, it is worth noting that all results from revisions will be published alongside the manuscript.

We thank the reviewer again for this valuable suggestion. Indeed, these are important directions for further analysis. In fact, we have examined these questions in a separate manuscript (Wang Y, Brunner P, Willie JT, Cao R, Wang S. Neural computations of visual, semantic, and memorability features in the human brain. PsyArXiv), using the same dataset (with a few additional patients). In that work, we applied PLS regression to construct axis-coding models for visual and semantic features and used semantic embeddings to build RDMS for RSA comparisons. We conducted a detailed characterization and comparison of visual and semantic models along the ventral visual pathway, including both the VTC and MTL. It is also worth noting that Supplementary Fig. 3 of the current manuscript illustrates the alignment of axis-coding channels with model-based visual feature dimensions.

The authors' responses and additional analyses are much appreciated. Future work may benefit from further examining the psychological plausibility of modeling category representations using ANOVA or alternative frameworks, and from conducting a detailed comparison of visual, semantic, and category models along the ventral visual pathway. Overall, the study provides a clear and important contribution by applying careful analyses to a valuable dataset, offering new insights into the neuronal transformations within the ventral visual stream.

We thank the reviewer again for the thoughtful and constructive comments.